# Dangerous degree forecast of soil and water loss on highway slopes in mountainous areas using RUSLE model

**Yue Li[1,2], Shi Qi*[1,2], Bin Liang[1,2], Junming Ma[1,2], Baihan Cheng[1,2],Cong Ma[3], Yidan Qiu[3], and Qinyan Chen[3]**

[1] Key Laboratory of State Forestry Administration on Soil and Water Conservation, Beijing Forestry University, Beijing 100083, China

[2] Beijing Engineering Research Center of Soil and Water Conservation, Beijing Forestry University, Beijing 100083, China

[3] Yunnan Science Research Institute of Communication & Transportation, Kunming 650011, China

**Abstract**

Many high and steep slopes are formed by special topographic and geomorphic types and mining activities during the construction of mountain expressways. Severe soil erosion may occur under heavy rainfall conditions. Therefore, predicting soil and water loss on highway slopes is important in protecting infrastructure and human life. This work studies Xinhe Expressway, which is in the southern edge of Yunnan–Guizhou Plateau, as the research area. The revised universal soil loss equation is selected as the prediction model of the soil and water loss on the slopes. Moreover, geographic information system, remote sensing technology, field survey, runoff plot observation testing, cluster analysis, and cokriging are adopted. The partition of the prediction units of the soil and water loss on the expressway slope in the mountain area and the spatial distribution model of the linear highway rainfall are studied. In view of the particularity of the expressway slope in the mountain area, the model parameter factor is modified and the risk of soil and water loss along the mountain expressway is simulated and predicted under 20- and 1-year rainfall return periods. The results are as follows. (1) Considering natural watershed as the prediction unit of slope soil erosion can represent the actual situation of the soil and water loss of each slope. The spatial location of the soil erosion unit is realized. (2) An analysis of the actual observation data shows that the overall average absolute error of the monitoring area is 33.24 $t \cdot km^{-2} \cdot a^{-1}$, the overall average relative error is 33.96%, and the overall root mean square error is between 20.95 and 65.64, all of which are within acceptable limits. The Nash efficiency coefficient is 0.67, which shows that the prediction accuracy of the model satisfies the requirements. (3) Under the condition of 1-year rainfall, we find through risk classification that the percentage of prediction units with no risk of erosion is 78%. Results show that soil erosion risk is low and therefore does not affect road traffic safety. Under the 20–year rainfall condition, the percentage of units with high risk and extremely high risk is 7.11% and mainly distributed on the K109+500–K110+500 and K133–K139+800 sections. The prediction results can help adjust the layout of the water and soil conservation measures in these units.

**Key words:** Soil and water loss; highway slopes; mountainous areas; RUSLE; dangerous degree forecast

**Introduction**

China has been gradually accelerating the construction of highways in recent years, thereby improving the transportation network and driving rapid economic development. Especially with the implementation of the western development strategy of the country, advanced requirements that focus on gradually connecting coastal plains and inland mountains have been proposed for the construction of expressways. Many unstable high and steep slopes, such as natural, excavation, and fill slopes, are inevitably formed by the considerable filling and deep digging along expressways in mountain areas.

The slope is the most fragile part of an expressway in a mountain area. During the rainy season, soil erosion is easily caused by rainwash and leads to a worrisome extent of damage (Figure 1). At present, China's highway industry is still in a period of rapid development. By the end of 2014, the total mileages of highway network exceeds 4,400,000 kilometers, and the expressway's mileage is 112,000 kilometers (Yuan et al., 2017; Mori et al., 2017; Kateb et al., 2013; Zhou et al., 2016). According to statistics, with the development of highway construction in China, slope areas reach 200–300 million $m^2$ each year (Dong and Zeng 2003). In the next 20–30 years, expressways in China will measure more than 40,000 km. For every kilometer of a highway, the corresponding bare slope area formed measures 50,000–70,000 $m^2$. The annual amount of soil erosion is 9,000 $g/m^2$, which causes 450 t of soil loss every year (Chen 2010). The soil and water loss of roadbed slope is different from that of soil and water in woodland and farmland. Forestlands and farmlands are generally formed after years of evolution and belong to the native landscape. Most slopes are gentle and stable (Kateb et al., 2013). Traditional soil and water conservation research focus on slopes with 20% grade or below, but the roadbed slope of the highway is generally greater than 30% (Zhou 2010). Soil erosion on roadbed side slopes affects not only soil and water loss along the highway but also road operation safety (Gong and Yang 2016; Jiang et al., 2017). Therefore, soil erosion on the side slopes of mountain expressways must be studied to control soil erosion, improve the ecological environment of expressways, and realize sustainable land utilization (Wang et al., 2005; Yang and Wang 2006).

RUSLE is a set of mathematical equations that estimate average annual soil loss and sediment yield resulting from interrill and rill erosion (Renard et al., 1997; Foster et al., 1999; Zerihun et al., 2018; Toy et al., 2002). It is derived from the theory of erosion processes, more than 10,000 plot-years of data from natural rainfall plots, and numerous rainfall-simulation plots. RUSLE is an exceptionally well-validated and documented equation. A strength of RUSLE is that it was developed by a group of nationally-recognized scientists and soil conservationists who had considerable experience with erosional processes. (Soil and Water Conservation Society, 1993).

The use of revised universal soil loss equation (RUSLE) models as predictive tools for the quantitative estimation of soil erosion has been maturing for a long time (Panagos et al., 2018;

Cunha et al., 2017; Taye et al., 2017; Renard 1997). The range of application of these models
involves nearly every aspect of soil erosion. Moreover, many scholars have made many useful
explorations in modifying the model parameter values and improving the simulation accuracy:
Tresch et al. (1995) believed that the slope length ($L$) or slope steepness factor ($S$) is one of
the main factor for soil erosion prediction, and significantly influence the erosion values
calculated by the RUSLE. All existing S factors are derived only from gentle slope inclinations of
up to 32%. Many cultivated areas, particularly in Switzerland, are steeper than this critical value.
Eighteen plot measurements on transects along slopes ranging from 20–90% in steepness were
used in this study to qualitatively assess the most suitable $S$ factors for steep subalpine slopes.
Results showed that a first selection of an S factor was possible for slopes above the critical 25%
steepness (Tresch et al., 1995). Rick D (2001) found that using universal soil loss equation (USLE)
and RUSLE soil erosion models at regional landscape scales is limited by the difficulty of
obtaining an $LS$ factor grid suitable for use in geographic information system (GIS) applications.
Therefore, he described the modifications applied to the previous arc macro language (AML) code
to produce a RUSLE-based version of the $LS$ factor grid. These alterations included replacing the
USLE algorithms with their RUSLE counterparts and redefining assumptions on slope
characteristics. Finally, in areas of western USA where the models were tested, the RUSLE-based
AML program produced $LS$ values that were roughly comparable to those listed in the RUSLE
handbook guidelines (Rick et al., 2001). Silburn's (2011) research showed that estimating $K$ from
soil properties (derived from cultivated soils) provided a reasonable estimate of $K$ for the main
duplex soils at the study site as long as the correction for the undisturbed soil was used to derive $K$
from the measured data and to apply the USLE model (Silburn 2011). Wu (2014) adopted GIS and
Revised Universal Soil Loss Equation (RUSLE) method to analyze the risk pattern of soil erosion
in the affected road zone of Hangjinqu highway in Zhuji City, Zhejiang Province. Digital
Elevation Model (DEM) data, rainfall records, soil type data, remote sensing imaging, and a road
map of Hangjinqu highway were used for these GIS and RUSLE analyses (Wu et al., 2014). Chen
(2010) according to terrain characteristics of roadbed side slope and through concrete analysis of
terrain factor calculation method in Revised Universal Soil Loss Equation (RUSLE), the
compatible question of terrain factor computational method of roadbed side slope is appraised and
the revision method on the basis of measured data of soil erosion in subgrade side slope of
Hurongxi Expressway (from Enshi to Lichuan) in Hubei Province is proposed. The results indicate
that: (1) In RUSLE slope length factor can be calculated by formula of $L=\left(\lambda/22.1\right)^{m}$, but $m$
should not be checked by the original method for the highway subgrade side slope because its
gradient surpasses generally applicable scope of RUSLE; (2) L, slope length factor of highway
subgrade side slope can be calculated by formula $L=\left(\lambda/22.1\right)^{0.35}$ (Chen et al., 2010). Zhang (2016)
investigated the spatio-temporal distribution of soil erosion in ring expressway before and after
construction process, they used land use/cover map of Ningbo City in 2010, topographic map,

map of North Ring expressway and field survey data was collected to derive digital elevation model (DEM). Rainfall data was collected from local hydrological station. Based on the collected data, the spatial distribution of the factors in RUSLE model was calculated, and soil erosion maps of the north ring expressway were estimated. Then, the soil erosion amount was calculated at three different stages by using RUSLE model. The results shows that: Slight erosion was dominant during preconstruction period and natural recovery period, which accounted for 98.53% and 99.73%, respectively. During construction period, mild erosion and slight erosion was the largest, which accounted for 52.5% and 35.4%, respectively. In general, soil erosion during the construction period is mainly distributed in the temporary soil ground (Zhang et al., 2016).

However, methods used to fit the parameters affected the results, and minimizing the sum of the squares of errors in the soil losses provided better results than fitting an exponential equation did. Yang (2014) found that the $C$ factor value can be determined as a function of fractional bare soil and ground cover derived from MODIS data at regional or catchment scales. The method offers a meaningful estimate of the $C$ factor, thus indicating ground cover impact on soil loss and erosion hazard areas. The method is better than the commonly used techniques, which are based only on green vegetation (e.g., normalized difference vegetation index, NDVI). Thus, the study provided an appropriate approach to estimating the C factor in hillslope erosion modeling in New South Wales, Australia, using emerging fractional vegetation cover products. This approach simply and effectively mapped the spatial and temporal distribution of the RUSLE cover factor and hillslope erosion hazard in a large area. The methods and results described in this article are valuable for understanding the spatial and temporal dynamics of hillslope erosion and ground cover. According to a study by Kinnell PIA (2014), runoff production, which is spatially uniform, is often inappropriate under natural conditions, where infiltration is spatially variable. The use of an upslope slope length that varies with the ratio of the upslope runoff coefficient to the runoff coefficient for the area down to the downslope boundary of the segment in modifications of the RUSLE approach produces only minor variations in soil loss compared with those predicted using the standard RUSLE approach when the runoff is spatially variable and the number of segments increases. On the contrary, the USLE-M approach provides predictions of soil loss that are influenced strongly by runoff when runoff varies in space and time. Therefore, an increase in the runoff through a segment causes an increase in soil loss, whereas a decrease in the runoff through a segment or cell results in a decrease in soil loss.

In general, these studies are mainly limited to sloping fields (Tresch S and others 1995; Rick D and others 2001; Silburn 2011; Yang 2014; Kinnell 2014).The research on soil erosion on highway slopes is limited. Subgrade slope is a major part of soil erosion in construction and operation periods. Therefore, the soil erosion caused by this slope should be predicted. However, the research progress on soil and water loss of highway hardly meet the requirements of the practical work. (Xu et al., 2009; Bakr et al., 2012). So far, we still need to do a lot of work on the prediction of soil erosion in highway slopes. The situation in various regions in China shows that certain researchers have improved the RUSLE model and studied soil erosion that occurs in

certain areas. Water and soil erosion caused by engineering construction is an important form after agricultural cultivation and forestry deforestation, the amount of soil erosion produced by the embankment slope occupies a large proportion in the whole project. It is not only related to the feasibility and cost of the project, but also has aroused great interest and attention. Yang (2001) investigated the behavior of soil erosion on the slope of a railway embankment during construction by comparing artificial and natural rainfalls on the special Qinhuangdao–Shenyang line of passenger trains. The results showed that the main type of soil erosion in the study area was gully erosion, which caused more soil erosion than surface erosion did; in addition, the principal factor causing soil erosion on the slope was the amount of precipitation and the width of the embankment. Wang (2005) established several experimental standardized spots for soil loss collection on the side slopes of the Xiaogan–Xiang fan freeway under construction and installed an on-the-spot auto-recorder of rainfall. The data collected were used for the revision of the main parameters R (rainfall and runoff) and $K$ (erodibility of soil) of the USLE, which is widely applied to forecast soil loss quantity in plowlands and predict the soil loss quantities of different types of soil on side slopes disturbed by engineering treatment (Wang et al., 2005). It can not only be applied to the prediction of disturbed soil loss in expressway construction, but also improve the prediction accuracy. It provides scientific support for relevant units or personnel to take reasonable preventive measures.

According to the literature, the study of soil and water loss in highways has the following problems. (1) In using the RUSLE model, most of the research on the $C$ and $P$ factors was conducted by referring to previous research results and data accuracy is often poor. (2) Most studies on rainfall erosivity ($R$) factors are still limited to sloping fields, and the rainfall erosivity factors of expressway slopes in mountain areas have rarely been studied. (3) Slope soils in highways differ from the broad sense of arable soil; moreover, the slopes themselves are also varied. Thus, accurately predicting the soil loss of different types of subgrade slopes using the traditional K factor calculation method is difficult.

Therefore, the RUSLE equation is selected as the prediction model for the soil and water loss on slopes with GIS technology as support in view of the characteristics of the soil and water loss in mountain expressways. The soil erodibility factor ($K$), slope length factor ($LS$), and soil and water conservation measure factor ($P$) are revised to improve the method of dividing slope units. In determining the predictive parameters of the model, the rainfall is obtained by spatial interpolation. The use of this technique addresses the shortage of rainfall data in mountain areas, the difficulty of representing the rainfall data of an entire expressway with those from a single meteorological station, and the uneven spatial distribution and strong heterogeneity of rainfall in mountain areas. In this study, a suitable prediction model of soil and water loss is established, the parameters of the model are revised, and the risk of soil and water loss under different rainfall scenarios is simulated and predicted. This study not only scientifically predicts the amount of soil erosion caused by the highway construction in mountain areas, but also provides a scientific basis for the prevention and control of soil erosion, and the rational allocation of prevention and control

measures. Meanwhile, the safe operation of highways and the virtuous cycle of the ecological
environment should be ensured to promote the sustainable development of the local economy.
**1 Study area**
Xinhe Expressway is in the southern margin of the Yunnan–Guizhou Plateau, which is in
southeast Yunnan Province, Honghe Prefecture, and Hekou County. This highway was the first in
Yunnan to cross the border, thereby becoming an important communication channel between
China and Vietnam and obtaining important strategic and economic value. The highway is at
longitude 103° 33′ 45″–103° 58′ 32″ and latitude 22° 31′ 19″–22° 51′ 48″. The expressway
stretches roughly from northwest to southeast, and the total length is 56.30 km. The climate type
belongs to subtropical mountain, seasonal monsoon forest, and humid heat climate categories.
Between May and the middle of October, the area experiences wet season characterized by
abundant rainfall, concentrated precipitation, and increased rain at night, the variation on
precipitation is from 400 to 2000mm, and most of the regions are between 800 to 1800mm (Fei et
al., 2017; Zhang et al., 2017). During the rest of the year, the area undergoes dry season. The
starting point of Xinhe Expressway is in Hekou County, New Street (pile number K83+500) at an
altitude of 296 m. The end point is in the estuary of Areca Village (pile number K139+800) at an
altitude of 95 m. The mountains along both sides are 200–380 m above sea level (Figure 2). The
topography of the hilly area in the northern part of Xinhe Expressway is complicated. The slopes
on both sides rise and fall, and most of the valleys constitute "V"- and "U"- shaped sections. The
natural slopes on both sides are mostly below 30°. The southern part of the highway has a
relatively flat terrain and a gentle slope. The slope of most of the hills on both sides is less than
15°, and the overall height difference is smaller than 100 m. The vegetation in the southern part of
Xinhe Expressway includes tropical rainforests and tropical monsoon forests. Meanwhile, the
vegetation in the northern part of China is classified as a south subtropical monsoon evergreen
broad-leaved forest. In recent years, the original vegetation in this area has been reclaimed as
farmland and is now planted with rubber, banana, pineapple, and pomegranate, which are sporadic
tropical rainforest survivors. The project area along Xinhe Expressway is an economic forest belt
with a single vegetation type and mainly has rubber, forest, and other economic trees. The soil
types along the highway are rich and mainly red, leached cinnamon, gray forest, and gray
cinnamon soils.

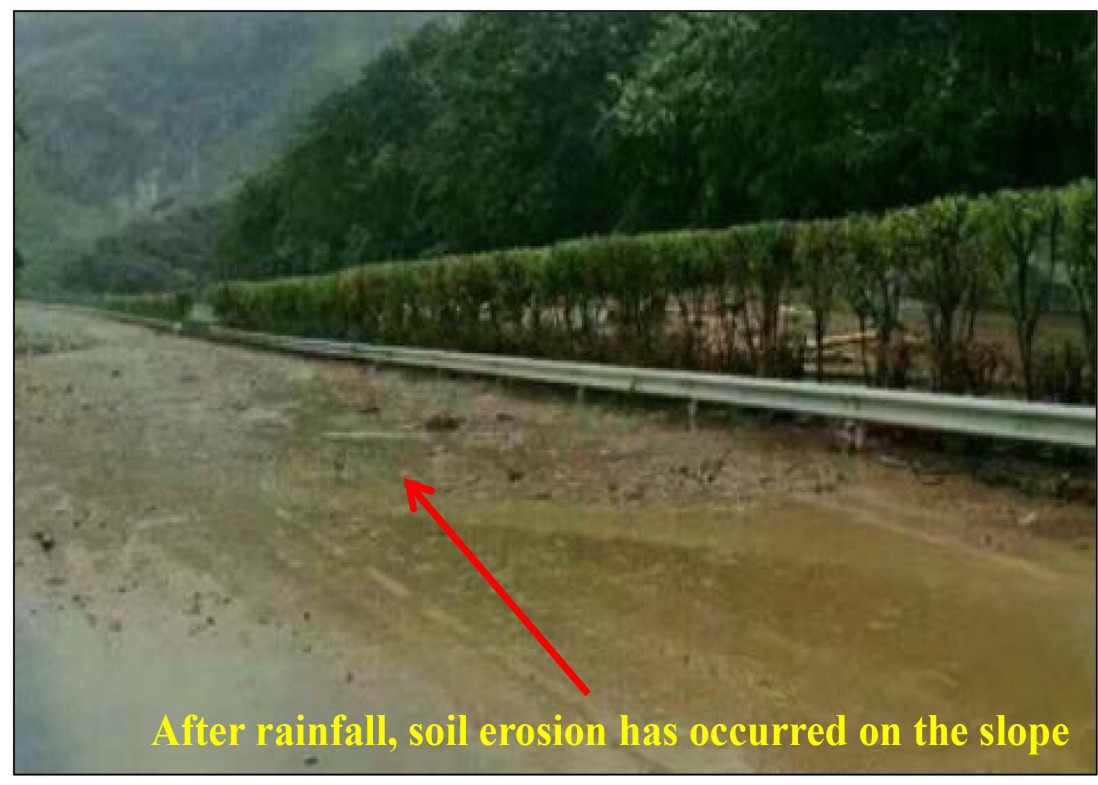

**Figure 1.** Soil erosion produced by rainwash on slope

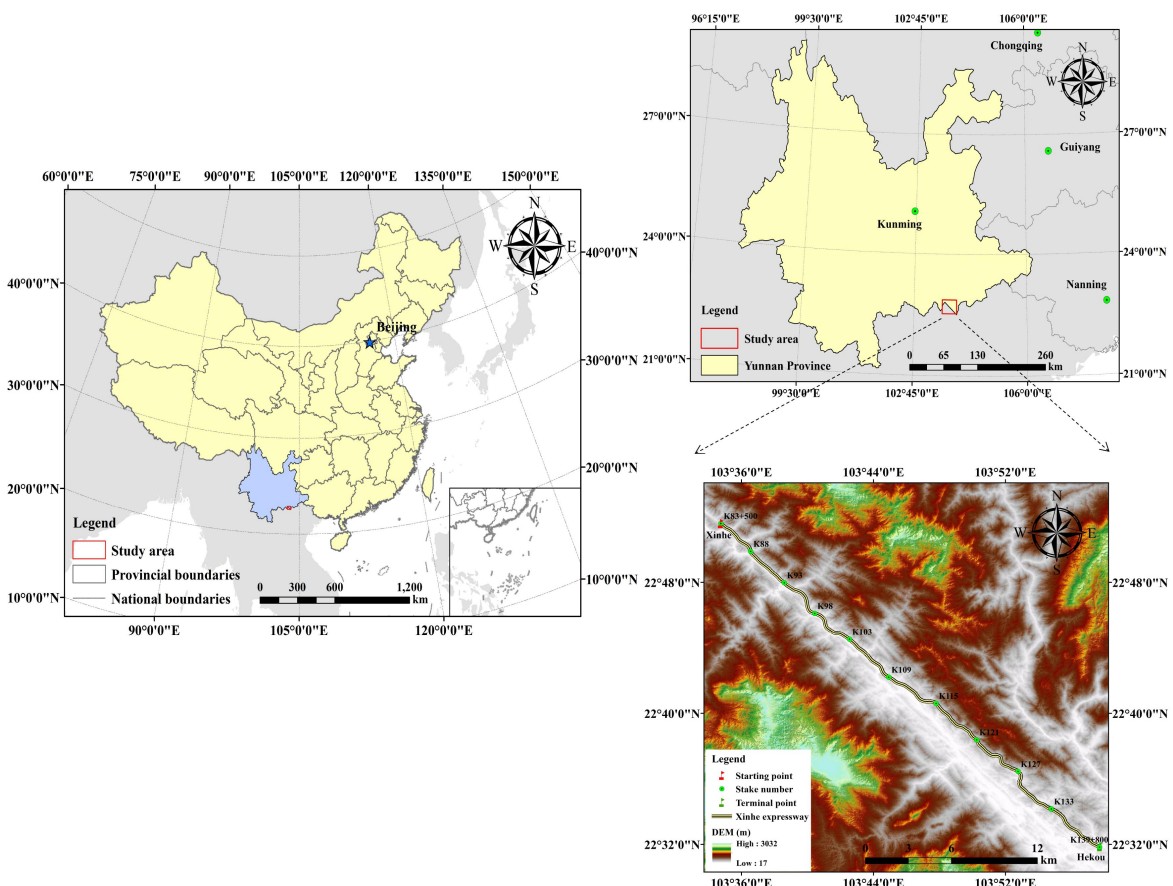

**Figure 2.** Overview of study region
**2 Materials and method**
**2.1 Data sources**
2.1.1 Meteorological data
Rainfall data from 2014 were obtained from Hekou Yao Autonomous County, Pingbian Miao
Autonomous County, Jinping Miao Yao Autonomous County, and the meteorological department
of Mengzi. The rainfall data type was in 5 min format. Meanwhile, two automatic weather stations
were established along Xinhe Expressway to gather weather data during the 2014 experiment.
Meteorological data was provided by the China Meteorological Data Network covered the period
of 1959–2015 (http://data.cma.cn/site/index.html).
2.1.2 Soil data
Soil type data were provided by Yunnan Traffic Planning and Design Institute. Soil texture
and organic matter data were obtained by field surveys, data sampling, and processing methods.
Soil samples were collected from every 1 km of the artificial and natural slopes on both sides of
the highway. Five mixed soil samples were obtained from one slope using the "S"-shaped
sampling method (Shu et al., 2017). Then, the method of coning and quartering was adopted
(Oyekunle et al., 2011), and half of the soil samples from the mixed soil samples were brought to
the laboratory for analysis. Finally, 186 soil samples were obtained. After the soil samples were
dried and sieved, we measured the soil texture and organic carbon content through specific gravity
speed measurement and potassium dichromate external heating, respectively.
2.1.3 Topographic data
A topographic map and design drawings of Xinhe Expressway were provided by the Traffic
Planning and Design Institute of Yunnan Province. The 1:2000 scale of the topographic map
coordinate system was based on the 2000 GeKaiMeng urban coordinate system, elevation system
for the 1985 national height datum, and the format for the CAD map DWG format.
2.1.4 Image data
The remote sensing images used in this study were derived from 8m hyperspectral images
produced by GF-1 satellite (http://www.rscloudmart.com/).

**2.2 Predicting model selection**
The RUSLE equation (Renard et al., 1997) was used to predict the soil and water loss on the
side slopes of Xinhe Expressway. The RUSLE equation considers natural and anthropogenic
factors that cause soil erosion to produce comprehensive results. Various parameters are easy to
calculate, and the calculation method is relatively mature. The RUSLE model is suitable for soil
erosion prediction in areas where the physical model is not needed. See Formula (1).
$$A = R \cdot K \cdot L \cdot S \cdot C \cdot P ,  \qquad (1)$$
where $A$ is the average soil loss per unit area by erosion (t/hm$^2$), $R$ is the rainfall erosivity factor
(MJ·mm / (hm$^2$·h)), $K$ is the soil erodibility factor (t·hm$^2$·h / (hm$^2$·MJ·mm)), $L$ is the slope length
factor, S is the steepness factor, $C$ is the cover and management practice factor, and $P$ is the
conservation support practice factor. The values of $L$, $S$, $C$, and $P$ are dimensionless.

**2.3 Prediction unit division and implementation**
Geological structures and rock and soil categories are complex because of considerable
changes in topography and physiognomy. The forms of slopes also vary. In general, according to
the relationship between slope and engineering, slopes can be natural or artificial. Artificial slope
formations can be subdivided into slope embankments and cutting slopes. This study used the
software ArcGIS to convert the topographic map of the highway design into a vectorization file
because the artificial and natural slopes of watershed catchments are the main components of soil
erosion prediction. On the basis of the extracted graphical units of the artificial and natural slope,
the natural and artificial slope was divided into a uniform prediction unit according to the aspect,
slope, land use, water conservation measures. The aspect, slope, land use, water conservation
measures, and other attributes of each prediction unit were consistent.

**3 Results and analysis**
3.1 Natural slope catchment area
The catchment unit of the slope was constructed using the structural plane tools of ArcGIS
combined with ridge and valley lines and artificial slope and highway boundaries (Zerihun et al.,
2018). After the completion of the catchment unit, the slope was divided according to soil type
data (Table 1). After the division and overlaying of the remote sensing image map, the land use
types and soil and water conservation measures were considered indicators through visual
interpretation and field survey results in further classifying the confluence units. Finally, the
partition units were amended using the vegetation coverage data obtained along Xinhe
Expressway. A total of 814 natural slope catchment prediction units were divided.

**Table 1.** Distribution of soil types along Xinhe Expressway

| A section of a expressway | Soil type |
| --- | --- |
| K83+500~K84+900 | latosolic red soil |
| K85+200~K93+200 | leached cinnamon soil |
| K93+200~K95+900 | gray forest soil |
| K96+900~K97+800 | gray cinnamon soil |
| K97+800~K100+500 | leached cinnamon soil |
| K100+500~K101+100 | gray cinnamon soil |
| K101+100~K104 | leached cinnamon soil |
| K104~K109+100 | gray cinnamon soil |
| K109+100~K139 | leached cinnamon soil |


The artificial slope was divided into roadbed and cutting slopes according to the design of
Xinhe Expressway into 1:1.5 and 1:1.0 slopes. After the preliminary division, the slope
measurements, data design, and field survey results were used as bases for the subsequent detailed
division of the artificial slope into cement frame protection and six arris brick revetments. Mccool
(1987) stated that slope length varies within a 10 m range and has only a small effect on results.
The specifications of each frame in the cement frame protection along Xinhe Expressway are the
same. The horizontal projection length of the cement frame is the slope length value of the
artificial slope. Therefore, the slope length of the artificial slope of each frame of the cement
revetment was considered the same and with a value of 0. According to investigations, the
vegetation coverage of artificial slopes with different plant species substantially varies. To achieve
an accurate prediction of unit division and improve prediction accuracy, the artificial slopes should
be continuously classified according to the plant species. A total of 422 artificial slope prediction
units were thus obtained. Then, the data of the 1236 slope prediction units were edited using GIS.
The results are shown in Figure 3.

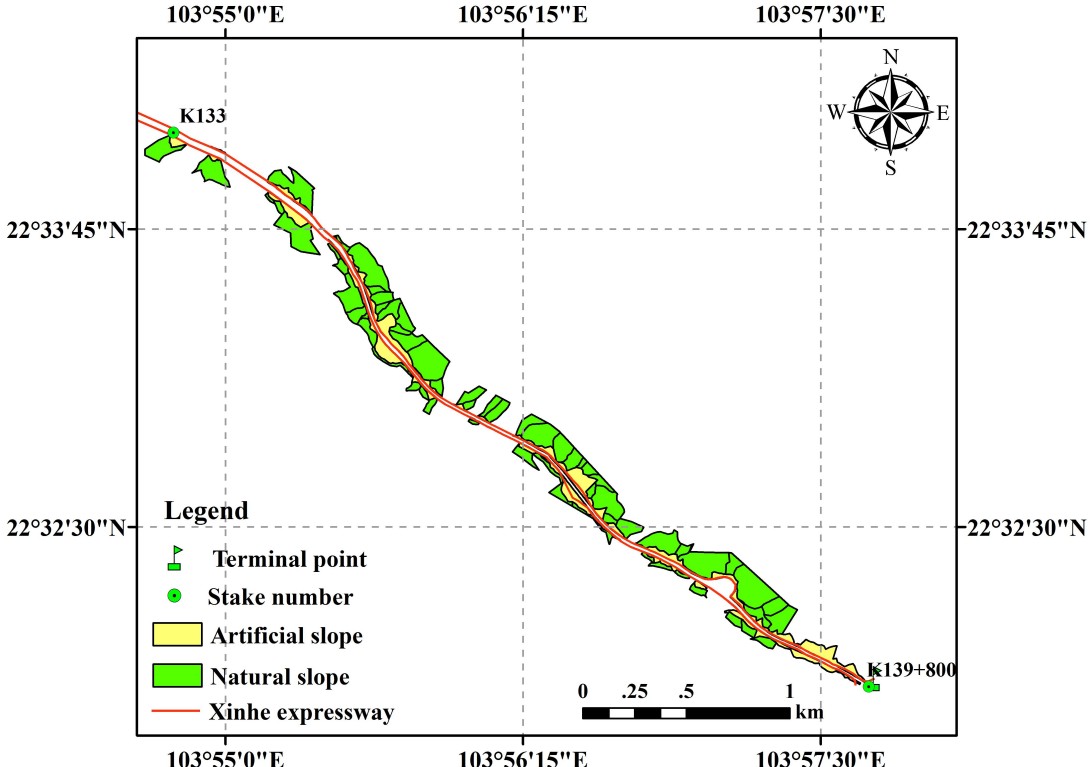

**Figure 3.** Division results of prediction units (A subset-6.8 km)

3.2 Determination of conventional parameter factor values of RUSLE model
3.2.1 Rainfall erosivity factor (*R*) (Panagos et al., 2017)

The formula of the *R* value (rainfall erosivity) was adopted (Wang et al., 1995; Liu et al.,

1999; Yang et al., 1999). This value was calculated using 30 min rainfall intensity as a measure, as
shown by Formulas (2) and (3).

$$R = 1.70 \cdot \left( P \cdot I_{30} / 100 \right) - 0.136 \qquad \left( I_{30} < 10\ mm/h \right), \qquad (2)$$

$$R = 2.35 \cdot \left( P \cdot I_{30} / 100 \right) - 0.523 \qquad \left( I_{30} \geq 10\ mm/h \right), \qquad (3)$$

where $R$ is the rainfall erosivity, $P$ is the sub-rainfall, and $I_{30}$ is the maximum 30 *min* rainfall
intensity.
The rainfall data were acquired from stationary ground meteorological stations. Thus, using
data from a single meteorological station to represent the rainfall data of a linear mountain
expressway was difficult. The $P$ and $I_{30}$ values along the highway were obtained by the method of
co-kriging calculations. The data included those derived from rainfall and 30 min rainfall data
from four meteorological stations in Hekou Yao Autonomous County, Pingbian Miao Autonomous
County, Jinping Miao Yao Autonomous County, and Mengzi City and those acquired from two
automatic weather stations along the highway. Then, the cross-validation method was used to
evaluate the accuracy of the interpolation results. The selection criteria were standard root mean
square error and the mean standard error. Detailed results are shown in Table 2. This work shows
only the interpolated results of secondary rainfall of two rainfall and 30 min rainfall intensity data,
as shown in Figure 4(a) and Figure 4(b).

**Table 2.** Interpolation error of $P$ and $I_{30}$ values

| The time of the second rainfall | $P$ | | $I_{30}$ | |
|---|---|---|---|---|
| | RMSS | MS | RMSS | MS |
| 2014.06.05 | 1.02 | -0.02 | 1.06 | -0.05 |
| 2014.06.07 | 1.04 | -0.02 | 1.01 | 0.02 |
| 2014.06.17 | 1.09 | 0.03 | 1.11 | 0.06 |
| 2014.06.28 | 1.11 | 0.07 | 1.05 | -0.03 |
| 2014.07.01 | 1.10 | 0.04 | 1.06 | -0.04 |
| 2014.07.13 | 1.03 | -0.02 | 1.01 | 0.02 |
| 2014.07.20 | 1.01 | 0.01 | 1.05 | 0.02 |
| 2014.08.02 | 1.03 | 0.03 | 0.94 | 0.02 |
| 2014.08.12 | 1.05 | -0.03 | 1.10 | 0.03 |
| 2014.08.26 | 1.03 | 0.01 | 0.97 | 0.03 |
| 2014.08.29 | 1.09 | -0.02 | 1.03 | -0.02 |
| 2014.09.02 | 1.07 | 0.03 | 1.05 | 0.02 |
| 2014.09.04 | 0.96 | -0.02 | 0.97 | -0.02 |
| 2014.09.17 | 1.07 | -0.03 | 1.09 | -0.03 |
| 2014.09.20 | 0.98 | 0.05 | 1.03 | 0.02 |
| 2014.10.05 | 1.02 | 0.03 | 1.04 | 0.03 |

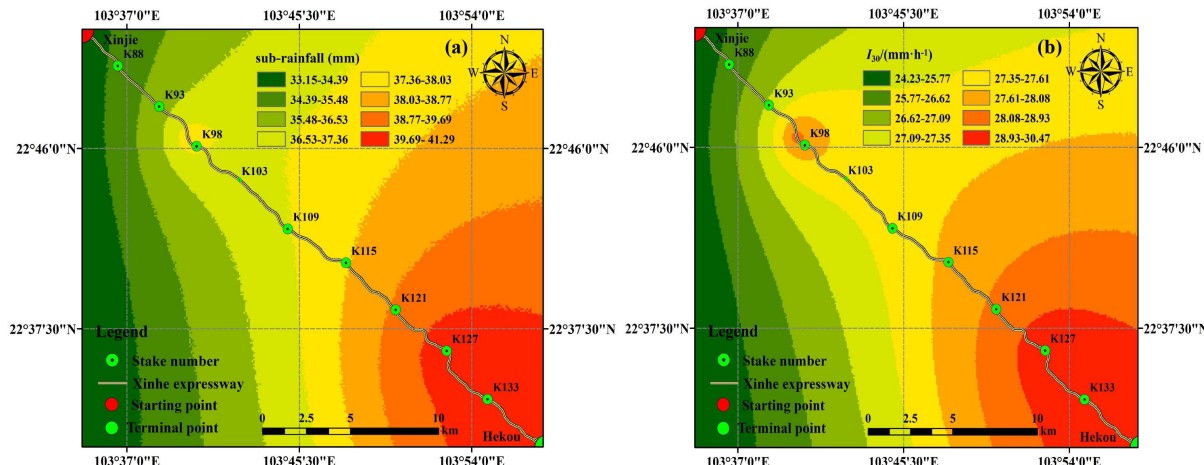

**Figure 4(a).** Interpolation results of secondary rainfall for June 5, 2014;
**Figure 4(b).** Interpolation results of $I_{30}$ for June 5, 2014

The secondary rainfall data of 16 rainfall instances along Xinhe Expressway were obtained
by interpolation because the internal rainfall and rainfall intensity of a single prediction unit are
the same. Therefore, the $R$ value was calculated using the average rainfall and rainfall intensity of
the unit. Only the spatial distribution map of the rainfall erosivity factors in certain sections (June
5, 2014) is shown due to space constraints (Figure 5 and Figure 5a).

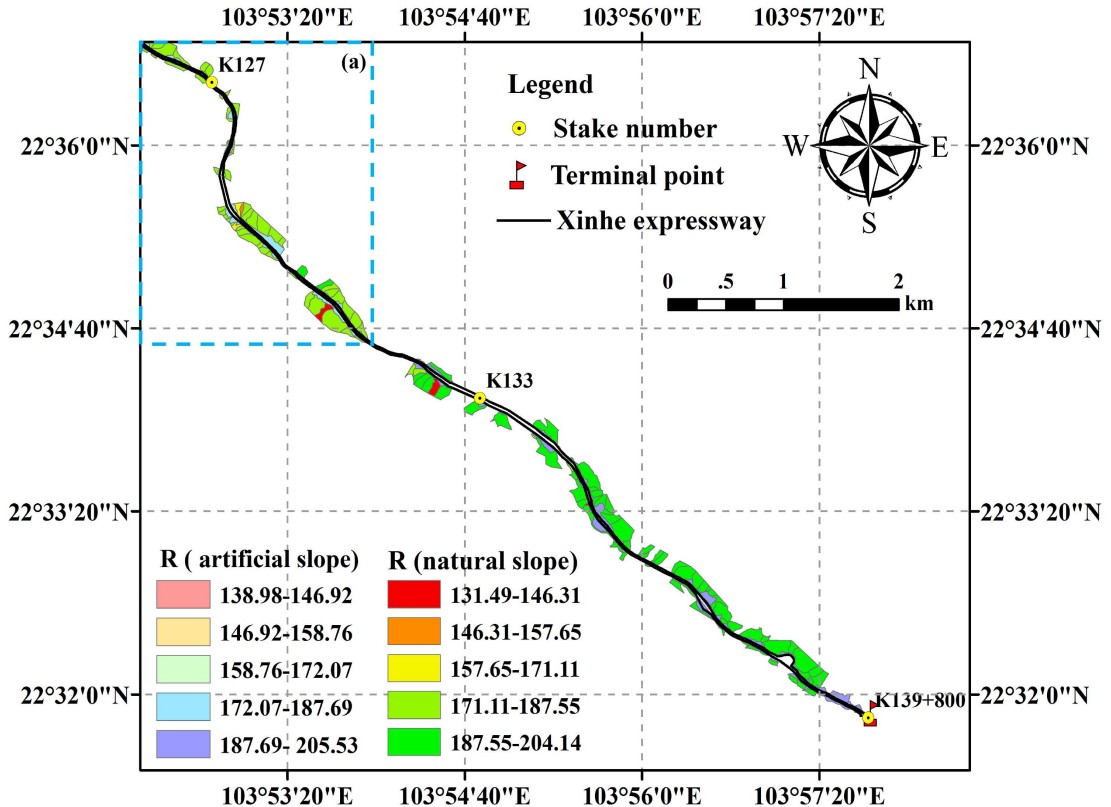

**Figure 5.** Spatial distribution map of rainfall erosivity factors (K127–K139+800)

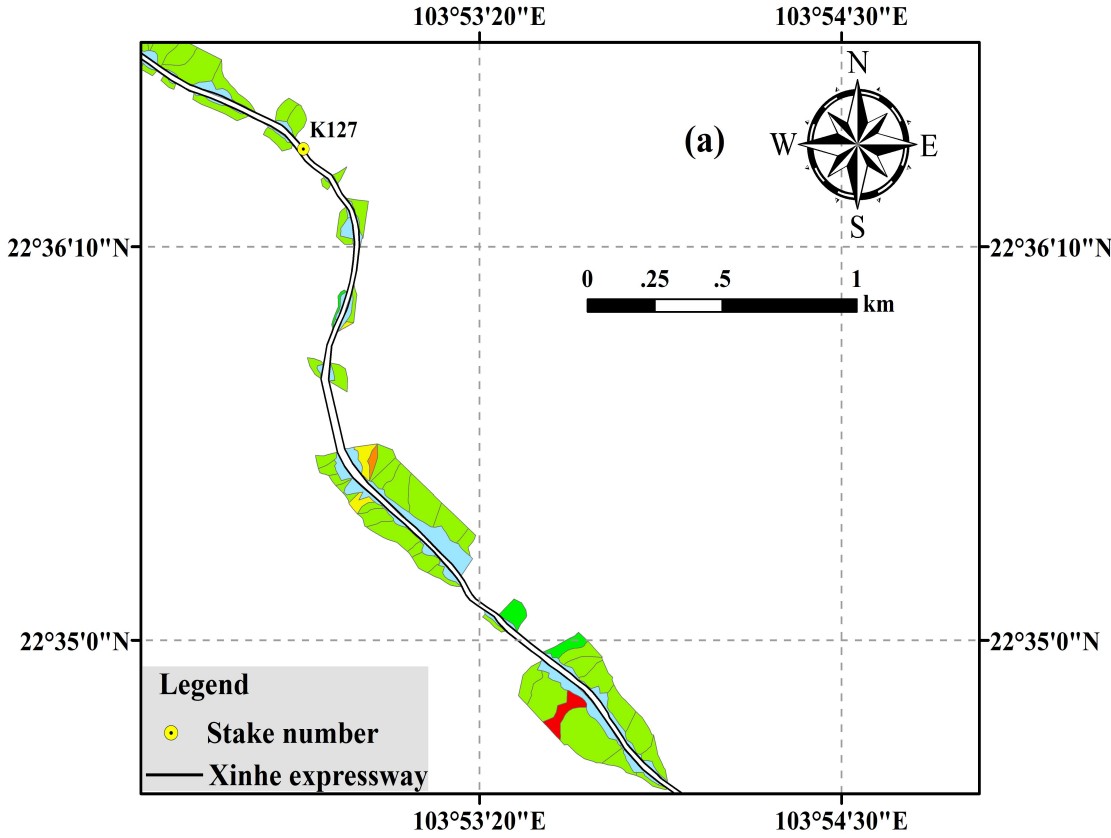

**Figure 5(a).** The subgraph of Figure 5 with zoomed sections.


3.2.2 Soil erodibility factor ($K$)

The soil data of the slope in each section were obtained by sampling on the basis of the

spatial distribution map of soil types in the study area and dividing the linear distribution of the
soil. The calculation method of the $K$ value was adopted by Formula 4 to obtain the soil erodibility
factor values of each slope (Sharply and Williams 1990 ), as shown in Tables 3 and 4 (see the
supplementary material/appendices).
$$K = 0.2 + 0.3e^{[0.0256SAN(1-SIL/100)]} \times \left(\frac{SIL}{CLA+SIL}\right)^{0.3} \times \left[1 - \frac{0.25C}{C+e^{3.72-2.95C}}\right] \times \left[1 - \frac{0.75N_1}{SN_1+e^{22.9SN_1-5.51}}\right] \quad (4)$$
In the formula, SAN, SIL, CLA, and C represent sand grains (0.05–2 mm), powder (0.002–0.05
mm), clay (<0.002 mm) and organic carbon content (%), and SN1=1-SAN/100, respectively.

3.2.3 Calculation of topographic factors in natural slope catchments
(1)  Slope length factor

According to the topographic map (1:2000 scale) and highway design of Xinhe Expressway, the

slope length and factor of slope catchment were calculated using DEM data with 0.5 m spatial
resolution generated by ArcGIS. The natural slope catchment slope was divided into less than 1°,
1°–3°, 3°–5°, and greater than or equal to 5° using the Reclassify tool in ArcGIS. The operation
formula adopted the L factor algorithm of Moore and Burch (1986), as shown by Formulas (5) and
364 (6).

$$L = \left( \frac{\lambda}{22.13} \right)^m \tag{5}$$

$$\lambda = flowacc \cdot cellsize \tag{6}$$

In the formula, $L$ is normalized to the amount of soil erosion along the slope length of 22.13 m, $\lambda$
is the slope length, *flowacc* is the total pixel number of water flowing into the pixel that is higher
than the pixel, and *cellsize* refers to the DEM resolution size. The value is 0.5 m, and $m$ is the $LS$
factor. See Formula (7).

$$m = \begin{cases} 0.2 & \theta < 1° \\ 0.3 & 1° \le \theta < 3° \\ 0.4 & 3° \le \theta < 5° \\ 0.5 & \theta \ge 5° \end{cases}, \tag{7}$$

where $\theta$ is the slope.
(2) Slope factor
The $S$ factor was calculated as follows. If the slope was less than 18°, then the formula of
McCool et al. (1987) was used. If the slope was greater than 18°, then the formula of Liu et al.
(1994) was adopted. See Formula (8).

$$S = \begin{cases} 10.8 \cdot sin\,\theta + 0.03 & \theta < 9° \\ 16.8 \cdot sin\,\theta - 0.05 & 9° \le \theta < 18° \\ 21.9 \cdot sin\,\theta - 0.96 & \theta \ge 18° \end{cases} \tag{8}$$

The DEM data were processed by ArcGIS to obtain slope data. The slope values of each
prediction unit were extracted using the Zonal statistics tool. Through the classification tool in
ArcGIS, the slope of the highway slope catchment of Xinhe was divided into less than 9°, 9°–18°,
and greater than or equal to 18°.
The S values of the slope catchments under the three slope grade conditions were calculated by
combining Formula (8) and ArcGIS techniques. The LS values of the slope prediction units are
shown in Figure 6.

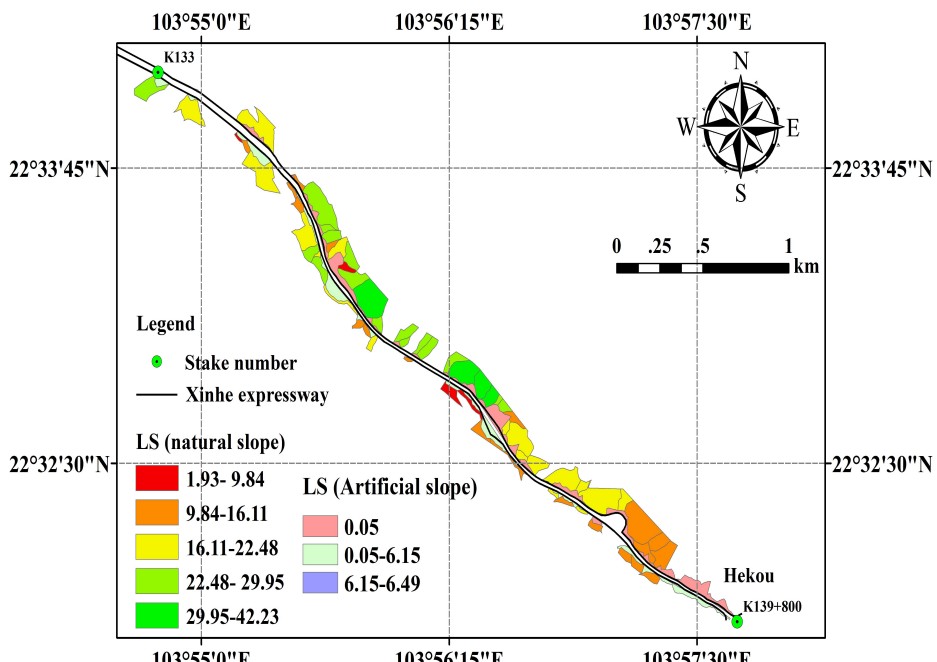

**Figure 6.** Spatial distribution map of topographic factors (K134–K139)

3.2.4 Calculation of topographic factors of artificial slopes

(1) Slope length factor

The method of Chen Zongwei (2010) was used to calculate the *LS* factor of the artificial slopes, and the calculation method for the topographic factors of the artificial slopes of Xinhe Expressway was modified. The slope length factor ($L_a$) was calculated using Formulas (5) and (6). The slope length index ($m_a$) was measured by a runoff plot experiment and then calculated by Formula (9).

$$m_a = log_{\frac{\lambda_1}{\lambda_2}} \frac{A_1}{A_2} \quad , \qquad (9)$$

where $A_1$ and $A_2$ are the soil erosion intensity values of two slopes when the slope lengths are $\lambda_1$ and $\lambda_2$, respectively. (The specifications of the two slopes were the same except for slope length.) The soil erosion amounts under 30 erosion rainfall conditions were monitored in the runoff field of Xiao Xinzhai of Mengzi City in 2014–2015 (Table 5). The $m_a$ value under each rainfall condition was calculated using Formula (9) according to the monitoring value of soil erosion amount. The average value of $m_a$ was 0.32, which was the $m_a$ value of the artificial slope length factor, as shown in Table 6.

**Table 5.** Amount of soil erosion of monitoring areas (t·km$^{-2}$·a$^{-1}$)

| The time of the second rainfall | 1 | 2 | 3 | 4 | 5 | 6 |
|---|---|---|---|---|---|---|
| 2014.06.05 | 4212 | 5158 | 5922 | 6423 | 12896 | 888 |
| 2014.06.07 | 1997 | 2447 | 2812 | 3089 | 6170 | 426 |

| | | | | | |
|---|---|---|---|---|---|
| 2014.06.17 | 867 | 1098 | 1227 | 1341 | 2664 | 185 |
| 2014.06.28 | 5700 | 7128 | 8107 | 8979 | 17915 | 1225 |
| 2014.07.01 | 477 | 608 | 686 | 748 | 1498 | 103 |
| 2014.07.13 | 1560 | 1915 | 2159 | 2374 | 4757 | 327 |
| 2014.07.20 | 3857 | 4878 | 5617 | 6183 | 12323 | 849 |
| 2014.08.02 | 5601 | 7048 | 7939 | 8600 | 17231 | 1194 |
| 2014.08.12 | 1955 | 2491 | 2881 | 3148 | 6294 | 435 |
| 2014.08.26 | 6211 | 7630 | 8750 | 9561 | 19196 | 1315 |
| 2014.08.29 | 1539 | 1889 | 2161 | 2356 | 4701 | 326 |
| 2014.09.02 | 611 | 758 | 868 | 950 | 1910 | 131 |
| 2014.09.04 | 1487 | 1893 | 2172 | 2372 | 4761 | 324 |
| 2014.09.17 | 1577 | 1954 | 2250 | 2451 | 4809 | 336 |
| 2014.09.20 | 1076 | 1329 | 1512 | 1633 | 3252 | 224 |
| 2014.10.05 | 749 | 925 | 1064 | 1172 | 2356 | 160 |
| 2015.07.04 | 5216 | 6377 | 7260 | 7877 | 15653 | 1090 |
| 2015.07.15 | 1575 | 1925 | 2192 | 2416 | 4775 | 334 |
| 2015.07.24 | 991 | 1250 | 1394 | 1522 | 3002 | 212 |
| 2015.07.28 | 4200 | 5188 | 5907 | 6544 | 13005 | 886 |
| 2015.08.13 | 829 | 1057 | 1189 | 1292 | 2567 | 177 |
| 2015.08.19 | 1010 | 1233 | 1390 | 1521 | 3016 | 208 |
| 2015.08.26 | 1682 | 2108 | 2415 | 2673 | 5263 | 364 |
| 2015.09.03 | 386 | 481 | 543 | 583 | 1169 | 81 |
| 2015.09.12 | 591 | 745 | 857 | 940 | 1868 | 129 |
| 2015.09.17 | 1172 | 1433 | 1632 | 1789 | 3555 | 245 |
| 2015.09.25 | 1369 | 1690 | 1906 | 2089 | 4152 | 287 |
| 2015.10.03 | 1188 | 1468 | 1671 | 1832 | 3664 | 252 |
| 2015.10.08 | 2908 | 3707 | 4220 | 4599 | 9196 | 625 |
| 2015.10.12 | 779 | 963 | 1111 | 1215 | 2339 | 164 |

**Table 6.** Calculation results of $m_a$

| The time of the second rainfall | $m_{12}$ | $m_{13}$ | $m_{14}$ | $m_{23}$ | $m_{24}$ | $m_{34}$ |
|---|---|---|---|---|---|---|
| 2014.06.05 | 0.29 | 0.31 | 0.30 | 0.34 | 0.32 | 0.28 |
| 2014.06.07 | 0.29 | 0.31 | 0.31 | 0.34 | 0.34 | 0.33 |
| 2014.06.17 | 0.34 | 0.32 | 0.31 | 0.27 | 0.29 | 0.31 |

| | | | | | |
|---|---|---|---|---|---|
| 2014.06.28 | 0.32 | 0.32 | 0.33 | 0.32 | 0.33 | 0.36 |
| 2014.07.01 | 0.35 | 0.33 | 0.32 | 0.30 | 0.30 | 0.30 |
| 2014.07.13 | 0.30 | 0.30 | 0.30 | 0.30 | 0.31 | 0.33 |
| 2014.07.20 | 0.34 | 0.34 | 0.34 | 0.35 | 0.34 | 0.34 |
| 2014.08.02 | 0.33 | 0.32 | 0.31 | 0.29 | 0.29 | 0.28 |
| 2014.08.12 | 0.35 | 0.35 | 0.34 | 0.36 | 0.34 | 0.31 |
| 2014.08.26 | 0.30 | 0.31 | 0.31 | 0.34 | 0.33 | 0.31 |
| 2014.08.29 | 0.30 | 0.31 | 0.31 | 0.33 | 0.32 | 0.30 |
| 2014.09.02 | 0.31 | 0.32 | 0.32 | 0.34 | 0.33 | 0.32 |
| 2014.09.04 | 0.35 | 0.35 | 0.34 | 0.34 | 0.33 | 0.31 |
| 2014.09.17 | 0.31 | 0.32 | 0.32 | 0.35 | 0.33 | 0.30 |
| 2014.09.20 | 0.30 | 0.31 | 0.30 | 0.32 | 0.30 | 0.27 |
| 2014.10.05 | 0.30 | 0.32 | 0.32 | 0.35 | 0.34 | 0.34 |
| 2015.07.04 | 0.29 | 0.30 | 0.30 | 0.32 | 0.30 | 0.29 |
| 2015.07.15 | 0.29 | 0.30 | 0.31 | 0.32 | 0.33 | 0.34 |
| 2015.07.24 | 0.33 | 0.31 | 0.31 | 0.27 | 0.28 | 0.31 |
| 2015.07.28 | 0.31 | 0.31 | 0.32 | 0.32 | 0.33 | 0.36 |
| 2015.08.13 | 0.35 | 0.33 | 0.32 | 0.29 | 0.29 | 0.29 |
| 2015.08.19 | 0.29 | 0.29 | 0.30 | 0.30 | 0.30 | 0.32 |
| 2015.08.26 | 0.33 | 0.33 | 0.33 | 0.34 | 0.34 | 0.36 |
| 2015.09.03 | 0.32 | 0.31 | 0.30 | 0.30 | 0.28 | 0.25 |
| 2015.09.12 | 0.34 | 0.34 | 0.34 | 0.35 | 0.34 | 0.32 |
| 2015.09.17 | 0.29 | 0.30 | 0.30 | 0.32 | 0.32 | 0.32 |
| 2015.09.25 | 0.30 | 0.30 | 0.30 | 0.30 | 0.31 | 0.32 |
| 2015.10.03 | 0.31 | 0.31 | 0.31 | 0.32 | 0.32 | 0.32 |
| 2015.10.08 | 0.35 | 0.34 | 0.33 | 0.32 | 0.31 | 0.30 |
| 2015.10.12 | 0.31 | 0.32 | 0.32 | 0.35 | 0.33 | 0.31 |
| The average value of $m_a$ | | | 0.32 | | | |

*$m_{xy}$ represents the m value simultaneously solved by erosion intensity values for monitoring plots that are numbered x and y.*

(2) Slope factor

The calculation of slope factor was based on the research method of Chen Zongwei. Six runoff plots were established in the Xiao xinzhai runoff field of Mengzi City. The soil erosion intensity under slope conditions of 1:1.5, 1:1.0, and 9:100 was monitored. Then, the slope factor under the

slope condition was obtained using Formula (10).

$$S_\theta = \frac{A_\theta}{A},$$  (10)

where $S_\theta$ represents the slope factor when the slope is $\theta$, $A_\theta$ represents the soil erosion intensity
when the slope is $\theta$ (t/hm²), and $A$ represents the soil erosion intensity when the slope is 9%
(t/hm²). The three slope conditions (1:1.5, 1:1.0, and control slope 9:100) in the soil erosion
monitoring experiment, combined with Formula (10), were used to calculate the slope factor
values of the two slopes (1:1.5 and 1:1.0) under the 30 rainfall conditions. The average factors of
the slopes under the 1:1.5 and 1:1.0 slope conditions were 7.28 and 14.49, respectively (Table 7).

After the slope design drawings were digitized by ArcGIS, the slope and length values of each

artificial slope prediction unit were determined according to the design specifications. The slope
length value of each artificial slope prediction unit was the horizontal projection length of the
cement frame. The slope length of the six arris brick revetment was 0. Formulas (5), (6), (9), and
(10), combined with the slope length factor and $m_a$ and $S_\theta$ values, were used to calculate the value
of $LS$ of each artificial slope prediction unit.

**Table 7.** Calculation results of slope factor

| The time of the second rainfall | $S_{46}$ | $S_{56}$ |
|---|---|---|
| 2014.06.05 | 7.23 | 14.52 |
| 2014.06.07 | 7.25 | 14.47 |
| 2014.06.17 | 7.25 | 14.41 |
| 2014.06.28 | 7.33 | 14.62 |
| 2014.07.01 | 7.28 | 14.57 |
| 2014.07.13 | 7.27 | 14.57 |
| 2014.07.20 | 7.28 | 14.52 |
| 2014.08.02 | 7.20 | 14.43 |
| 2014.08.12 | 7.23 | 14.46 |
| 2014.08.26 | 7.27 | 14.60 |
| 2014.08.29 | 7.24 | 14.44 |
| 2014.09.02 | 7.25 | 14.56 |
| 2014.09.04 | 7.33 | 14.72 |
| 2014.09.17 | 7.30 | 14.32 |
| 2014.09.20 | 7.28 | 14.49 |

| | | |
|---|---|---|
| 2014.10.05 | 7.33 | 14.73 |
| 2015.07.04 | 7.23 | 14.36 |
| 2015.07.15 | 7.24 | 14.32 |
| 2015.07.24 | 7.17 | 14.15 |
| 2015.07.28 | 7.39 | 14.68 |
| 2015.08.13 | 7.28 | 14.47 |
| 2015.08.19 | 7.33 | 14.53 |
| 2015.08.26 | 7.35 | 14.47 |
| 2015.09.03 | 7.22 | 14.47 |
| 2015.09.12 | 7.28 | 14.47 |
| 2015.09.17 | 7.29 | 14.48 |
| 2015.09.25 | 7.28 | 14.47 |
| 2015.10.03 | 7.27 | 14.53 |
| 2015.10.08 | 7.36 | 14.71 |
| 2015.10.12 | 7.40 | 14.26 |
| Average | 7.28 | 14.49 |

*Note: Sxy represents the slope factor value simultaneously solved by erosion intensity values for monitoring plots*
*numbered x and y.*

3.2.5 Cover and management practice factor
The *C*-factor after topography is an important factor that controls soil loss risk. In the RUSLE
model, the *C*-factor has been used to reflect the effects of vegetation cover and management
practices on the soil erosion rate ((Vander-Knijff et al., 2000; Prasannakumar et al., 2011;
Alkharabsheh et al., 2013). It is defined as the loss ratio of soils from land cropped under specific
conditions to the corresponding loss from clean-tilled and continuous fallow (Wischmeier and
Smith, 1978). Due to the variety of land cover patterns with severe spatial and temporal variation,
mainly in the watershed scale, data sets from satellite remote sensing were used to assess the C-
factor (Vander-Knijff et al., 2000; Li et al., 2010; Chen et al., 2011; Alexakis et al., 2013). Taking
full advantage of the Normalized Difference Vegetation Index (NDVI) data, C is calculated
according to the equation of Gutman and Ignatov (1998). The formula is shown as (11). Then, the
vegetation coverage data were corrected by selecting a sample plot every 2 km along the study
area for investigation. The algorithm for calculating f is referred to Tan et al (2005). The formula
is shown as (11). Finally, accurate vegetation coverage data were obtained (Figure 7). The *C* factor
map of the soil erosion prediction unit in the slope catchment area is shown in Figure 8.
$$C = 1 - \frac{NDVI - NDVI_{min}}{NDVI_{max} - NDVI_{min}} \qquad (11)$$

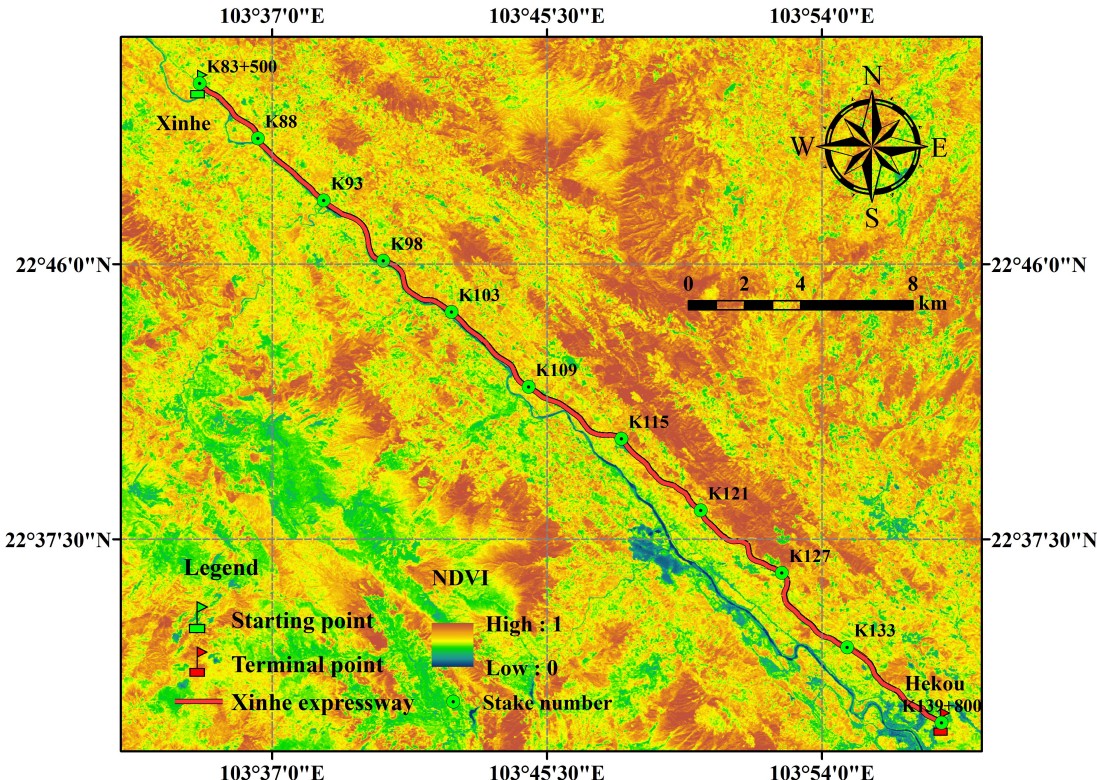

**Figure 7.** Vegetation coverage along Xinhe Expressway

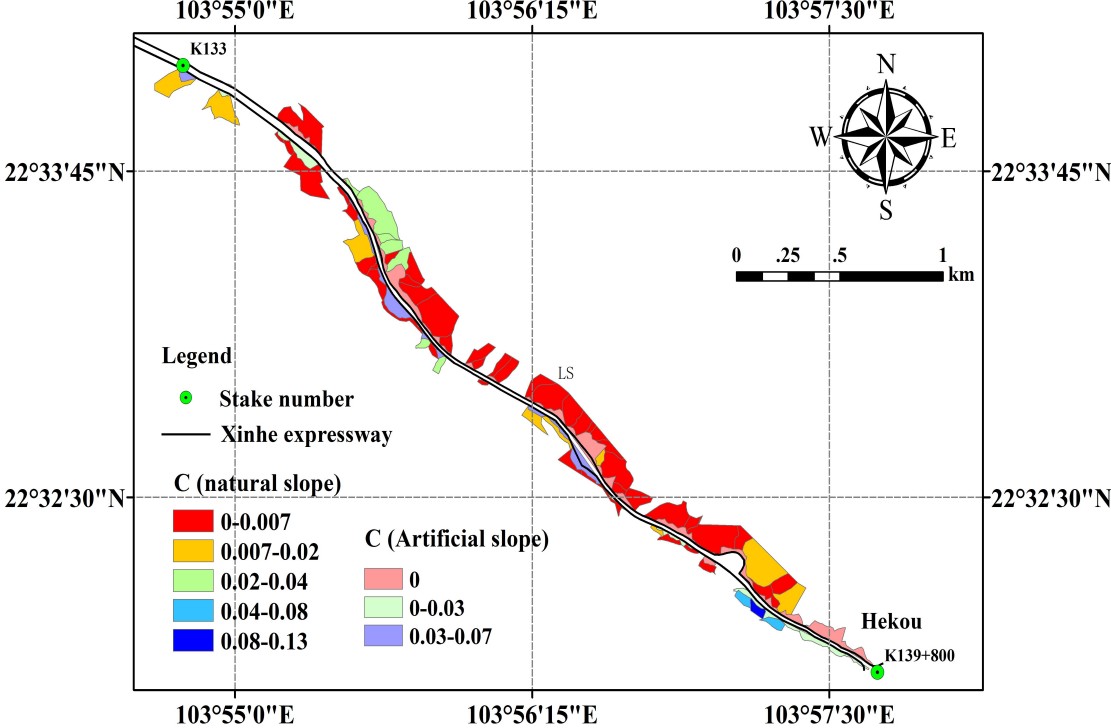

**Figure 8.** Spatial distribution map of cover and management practice factor

3.2.6 Factor of soil and water conservation measures

The land use types of the natural slope catchment area were mainly cultivated, forest,

construction, and difficult lands. Through field investigation and visual judgment, the water conservation measures of the farmland and forestland were identified to be mainly contour belt tillage, horizontal terrace and terrace, and artificial slope catchment area, including cement frame and six arris brick revetments. The $P$ values of the cement frame and the six arris brick revetments were determined by the area ratio method as 0.85 and 0.4, respectively. The $P$ values of the soil and water conservation measures are shown in Table 8.

**Table 8.** $P$ values of different slope types

| Slope type | Cement frame | Hexagonal brick | Contour strip tillage | Level bench/Terrace | Construction land | Difficult to use land | Other |
|---|---|---|---|---|---|---|---|
| $P$ | 0.85 | 0.4 | 0.55 | 0.03 | 0 | 0.2 | 1 |

**3.3 Validation of model simulation accuracy**

In this study, soil erosion in three monitoring areas under 16 erosive rainfall conditions was monitored in 2014. No rainfall occurred in the 24 h preceding each rainfall, and the disturbance of antecedent rainfall on soil erosion on the slopes was excluded. With an estimation of the historical soil and water loss of each slope prediction unit, the results were compared with data from three monitoring plots along the side slope of Xinhe Expressway, as shown in Figure 9-11.

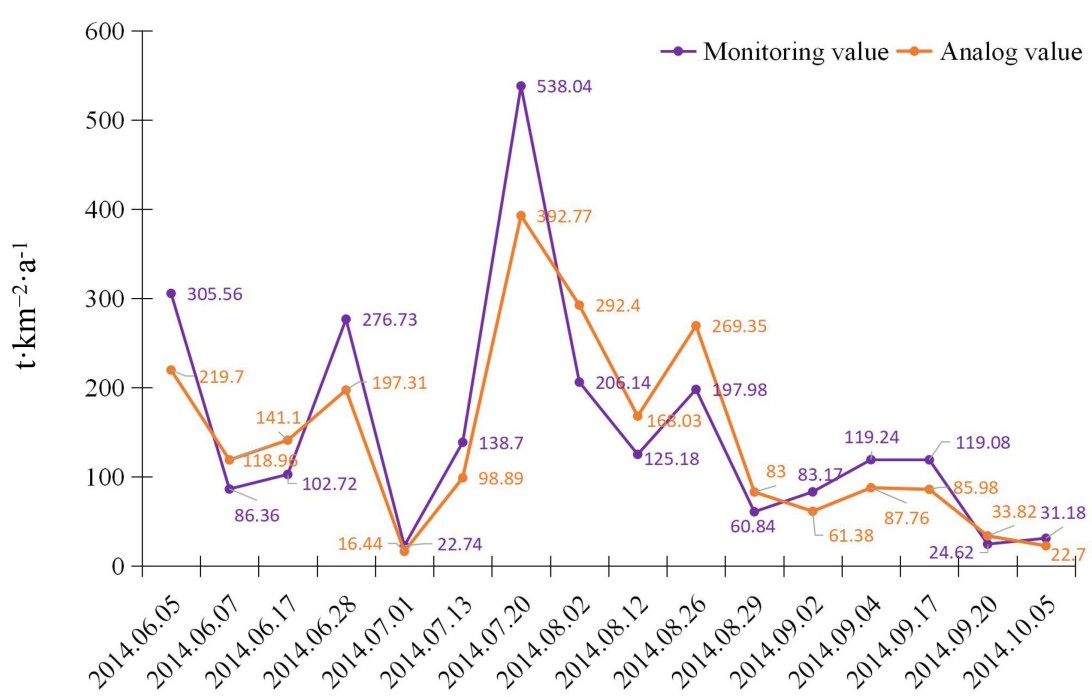

**Figure 9.** Comparison of model prediction and monitoring results ( K83+550)


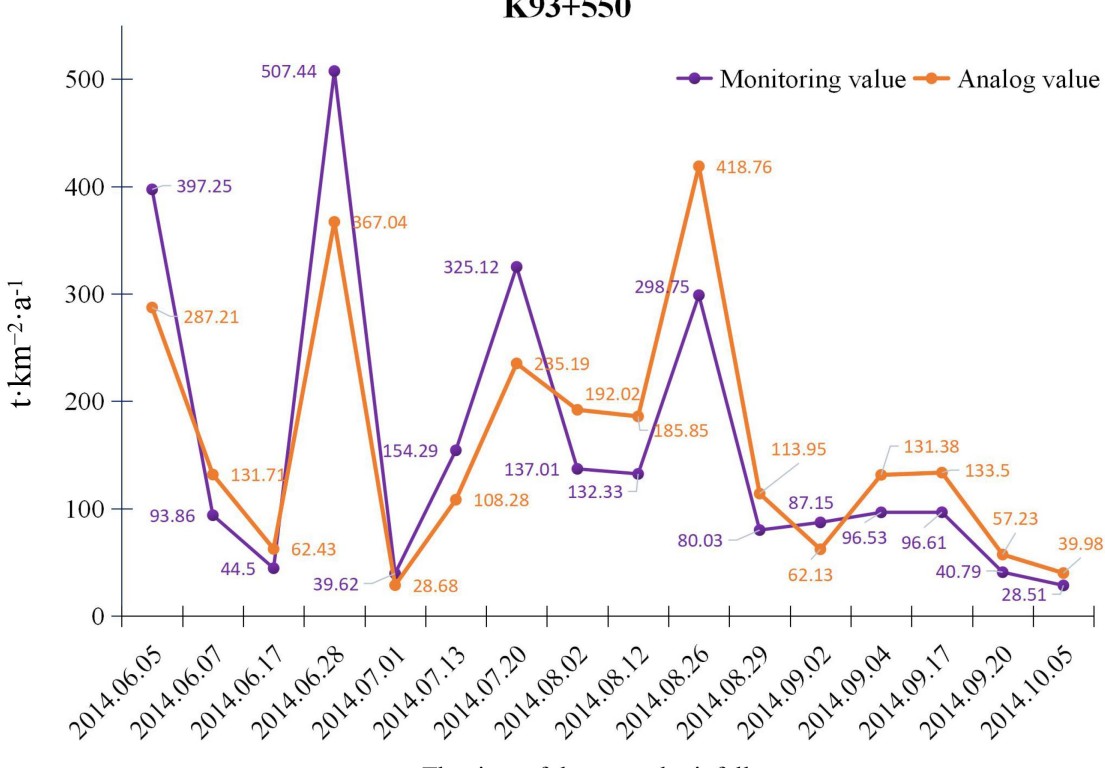

The time of the second rainfall

**Figure 10.** Comparison of model prediction and monitoring results ( K93+550)




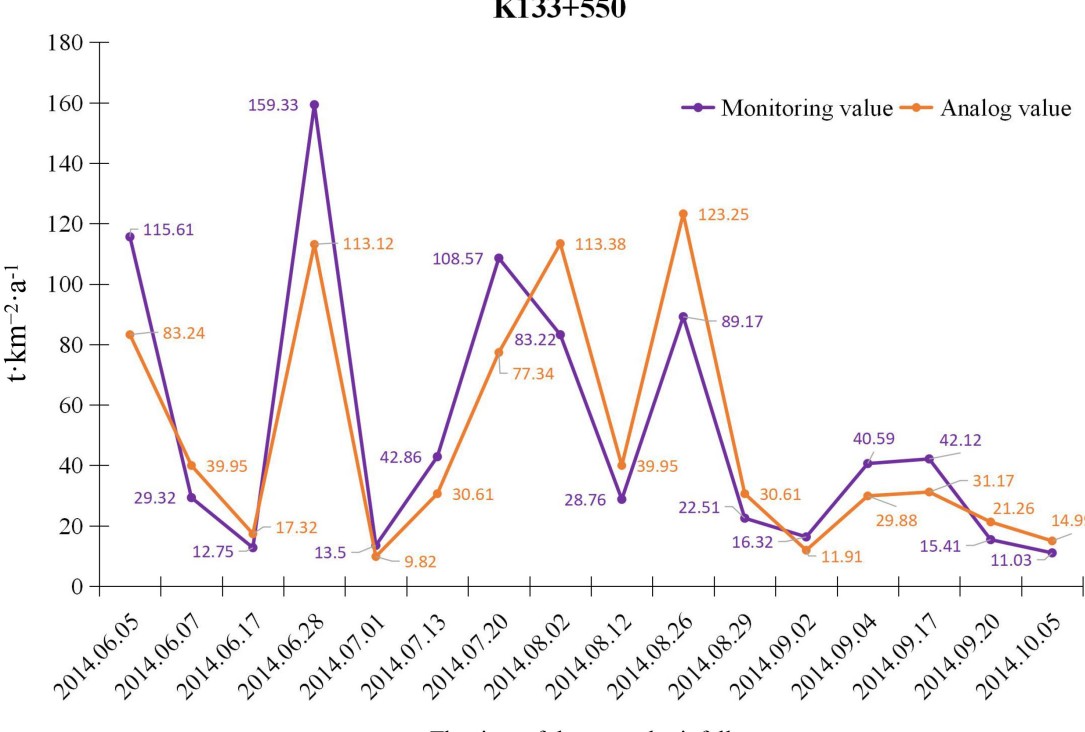

The time of the second rainfall

**Figure 11.** Comparison of model prediction and monitoring results ( K133+550)




480   The error analysis shows that under the 16 rainfall conditions, the absolute errors of the

481 three monitoring areas were 47.15, 52.52, and 16.27 t·km$^{-2}$·a$^{-1}$ and the overall average absolute

482 error was 38.65 t·km$^{-2}$·a$^{-1}$. The average relative errors were 31.80%, 35.49%, and 32.26%, and the

483 overall mean relative error was 31.18%. The root mean square errors were 59.44, 65.64, and 20.95,

484 all of which were within the acceptable range. The Nash efficiency coefficient of the model was

485 0.67, which was between 0 and 1 and thus shows that the model accuracy satisfied the

486 requirements. The calculation results are shown in Tables 10–12 (see the supplementary

487 material/appendices).

488  The analysis accuracy revealed that the northern and flat terrain of the southern region had a

489 small simulation error due to the high and low areas of the central region of the terrain, which

490 resulted in a slightly lower accuracy than that of the southern region. Under heavy rainfall

491 conditions, the absolute error value of the simulation was large. On the one hand, the result may

492 be caused by the artificial error in monitoring the sediment collection in the area. On the other

493 hand, the model itself may be defective.

495 **3.4 Application of early warning of soil erosion to mountain expressway**

496  The rainfall data and $I_{30}$ values in the 20 years covered by the study were obtained from the

497 meteorological departments of Mengzi, Pingbian, Jinping, and Hekou counties in Yunnan

498 Province. Rainfall and its intensity were interpolated using cokriging, which was introduced into

499 elevation and geographical position, as shown in Figures 12 and 13.

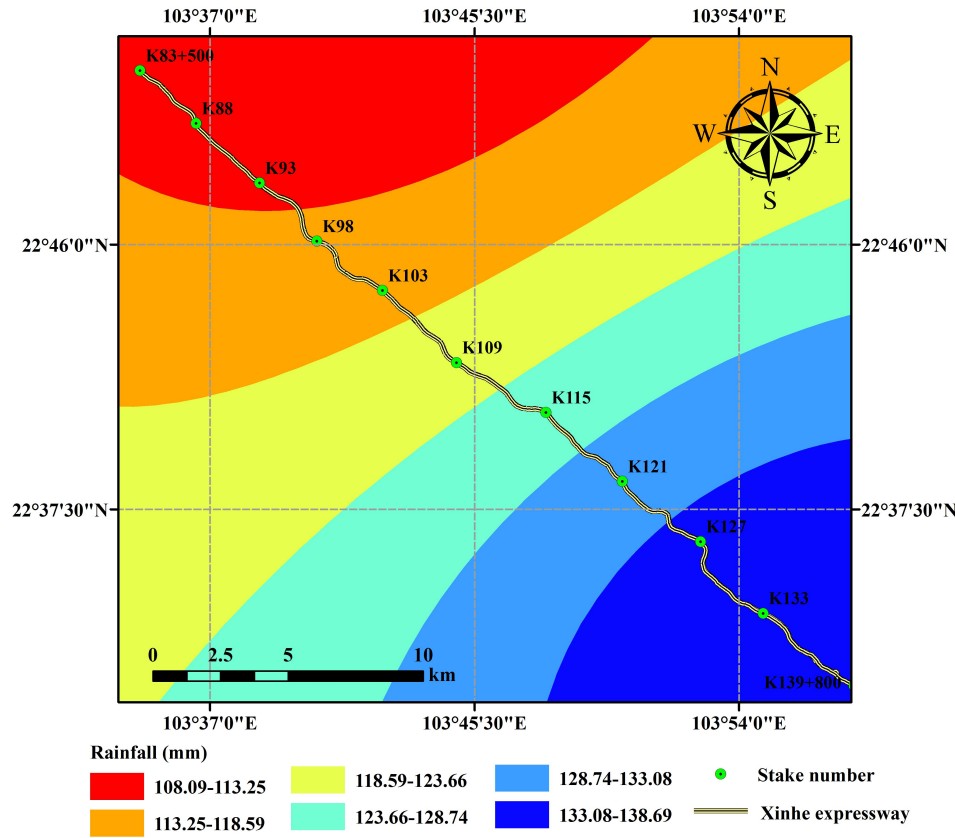

**Figure 12.** Rainfall interpolation results under 20-year return

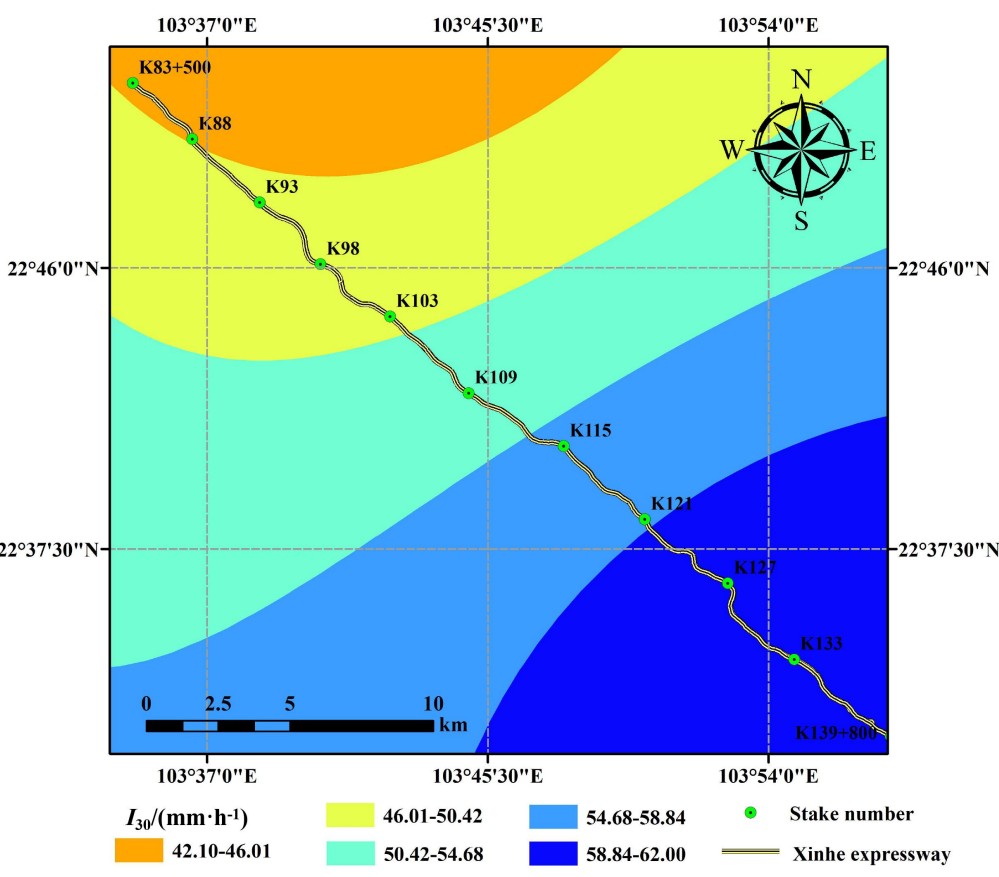

**Figure 13.** Rainfall intensity interpolation results under 20-year return


The total soil erosion amount of each prediction unit using 20-year rainfall data was obtained

by simulation according to the soil erosion intensity classification standard. The prediction results
were classified as "no risk," "slight risk," "moderate risk," "high risk," and "extremely high risk,"
as shown in Figure 14.

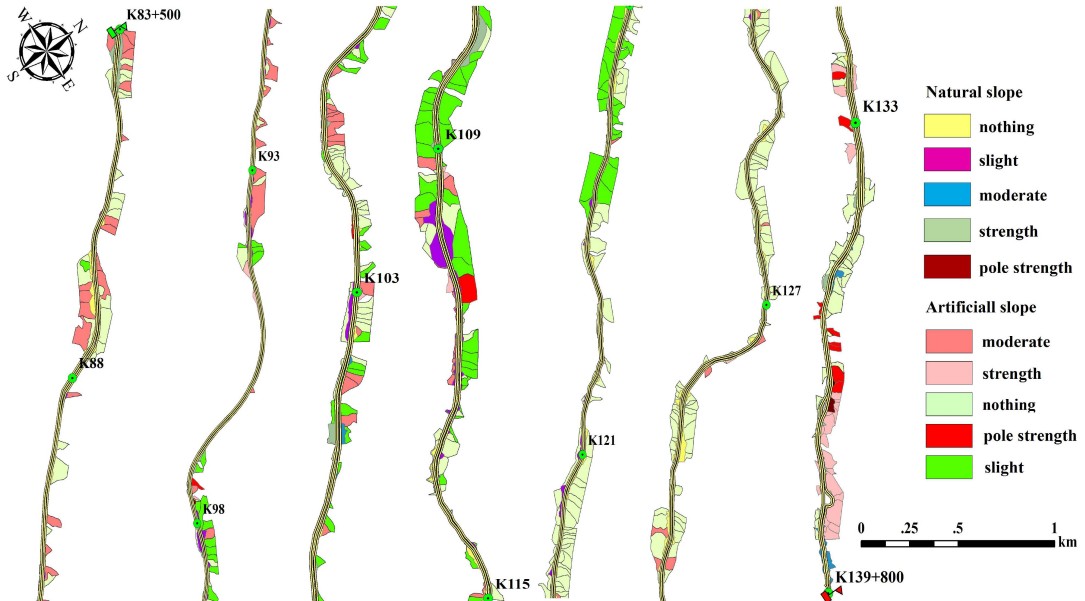


**Figure 14.** Risk analysis of soil and water loss under 20-year rainfall conditions


The grading results showed that the percentage of prediction units classified under low and

mild risk for soil and water loss was 88.60%. The risk of soil erosion was low in these areas. Thus,
road traffic safety was not affected. The percentage of prediction units classified as having
moderate risk was 4.29%. The risk of soil erosion in these areas was relatively low under the
general rainfall intensity. However, under high rainfall intensity, a certain scale of soil erosion
disaster could occur. The percentage of prediction units that were labeled high and extremely high
risk was 7.11%. The risk of soil erosion was great in these units. For example, from K134+500 to
K135+500 (1000 m), the average soil erosion amount on both sides of the slope under 20-year
rainfall amount reached $1757t \cdot km^{-2} \cdot a^{-1}$. Even if only a portion of the sediment was deposited on
the road, road safety would still be affected.

Similarly, the risk of soil erosion was analyzed according to the grading standard of soil and

water loss risk under the condition of 20-year rainfall by simulating the soil erosion amount of
each prediction unit under 1-year rainfall amount (Figure 15).

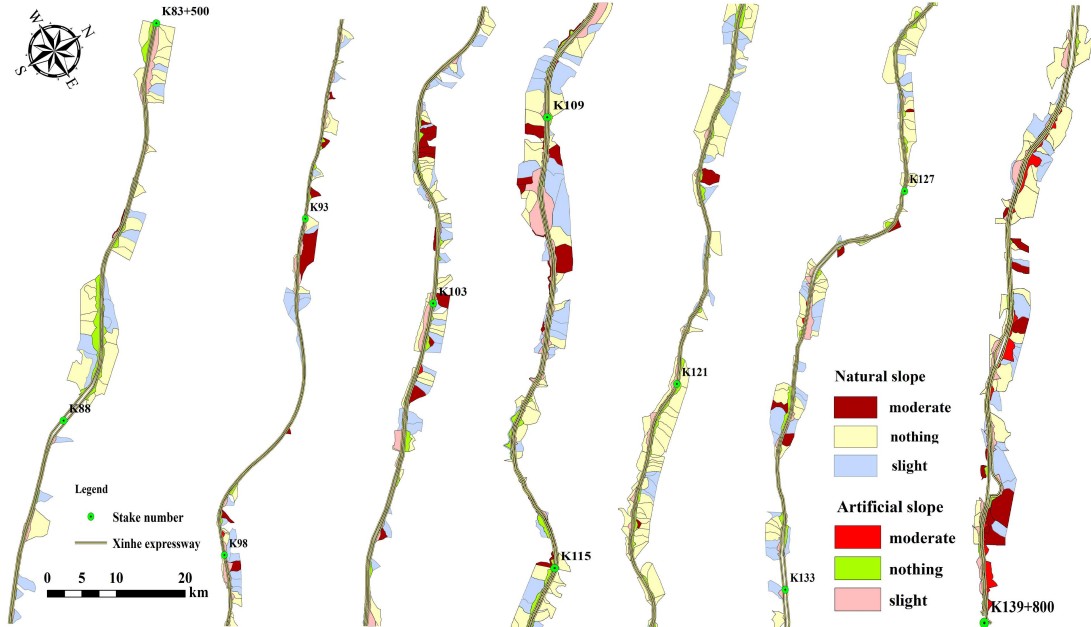

**Figure 15.** Risk analysis of soil and water loss under 1-year rainfall amount

The results indicated that the percentages of the prediction units with no and mild soil erosion risks were 78.00% and 17.92%, respectively. The risk of soil erosion was low in these areas. Thus, the safety of road traffic would not be affected. The percentage of prediction units at risk of mild soil erosion was 6.08%. The layout of soil and water conservation measures in these areas should therefore be rationally adjusted. Moreover, the comprehensive management of their slopes should be strengthened, and plant and engineering measures should be applied comprehensively to conserve soil and water in these regions. Inspections should be reinforced and motorists should be reminded to pay attention to traffic safety during the rainy season. Most of the artificial slopes covered by the study are made of six arris brick revetment, the amount of soil erosion is small, but the frame-type cement slope protection against soil erosion is sturdier than that in other areas. Slope protection measures should be rationally adjusted according to predicted results. We may consider slowing down the roadbed slope to keep the slope stable, then the ecological slope protection technology can be adopted. Such as the spraying and planting technology of bolt hanging net, it can build a layer of planting matrix which can grow and develop on the weathered rock slope, and can resist the porous and stable structure of the scouring. Finally, it can achieve the purpose of preventing and controlling soil erosion, beautifying the landscape environment of the road area and ensuring the safety of road traffic.

## 4   Discussion

Slope is the main part of soil and water loss caused by highways. So, it is very important to predict and early warning. The highway slope is divided into natural slope and engineering (artificial) slope. The RUSLE model is used to predict the soil erosion of natural slope, on the

premise of not considering the variation in rainfall erosivity, it is found that in the same type area, the method of model parameters acquisition are basically consistent through the literature analysis and comparison (Yang 1999; Yang 2002; Peng et al., 2007; Zhao et al., 2007; Chen et al., 2014; Zhu et al., 2016), after comparing the monitoring data onto runoff plots, it is found that the error between the predicted value and the monitoring value calculated by the RUSLE model is small (Yang 1999; Yang 2002; Li et al., 2004), it shows that the prediction results of the model are reliable. In the prediction of slope erosion of engineering (artificial), the previous study mainly considered the disturbance to the surface during the construction process (He 2004; Liu et al., 2011; He 2008; Hu, 2016; Zhang et al., 2016; Song et al., 2007), and do not consider the soil erosion resulting from the construction of the engineering slope; In the process of predicting soil and water loss in engineering slope by using RUSLE model, the correction of the conservation support practice factor (such as cement block, hexagonal brick, etc.) will often be ignored (Zhang 2011; Morschel et al., 2004; Correa and Cruz 2010);In addition, most of the cases using RUSLE model to predict soil erosion of highway slopes, in the use of remote sensing, it is usually based on grid data, but lack of consideration for catchment units (IsIam et al., 2018; Villarreal et al., 2016; Wu and Yan 2014; Chen et al., 2010).

Therefore, this study analysis the characteristics of soil erosion in the process of expressway construction, and improves the following aspects on the basis of previous research: (1) In order to be closer to the actual situation, we divide the highway slope into natural unit and artificial unit, and calculate the amount of soil loss from the slope surface to the pavement by the slope surface catchment unit, which is more in line with the actual situation, and this idea can be popularized; (2) We consider the spatial heterogeneity of linear engineering of expressway, then the rainfall factor is spatially interpolated, it has made up for the defects of rainfall data usually used by rainfall stations in previous studies. (3) We modify the parameters of the artificial slope by means of actual survey, runoff plot observation and other means, and the parameters of the artificial slope are corrected by referring to the form of the project and the materials used.

## 5   Conclusions

This study fully considered the differences between the model parameters of the artificial and natural slopes of mountain expressways. Each catchment area was considered a unit. The artificial and natural slope prediction units were then divided, thus producing 422 artificial slope prediction units and 814 natural slope catchment prediction units. The soil and water loss of each slope was predicted in real time, thus making the prediction of soil erosion accurate. The $R$ factor used the space interpolation method and the $P$ factor of the artificial slope was corrected by the area ratio method in determining the parameters of model prediction. The other factors were corrected by the experimental data. Error analysis of the actual observation data revealed that the overall average absolute error of each monitoring area was 38.65 $t \cdot km^{-2} \cdot a^{-1}$, the average relative error was 31.18%, the root mean square error was between 20.95 and 65.64, and the Nash efficiency coefficient was 0.67. The method of soil and water loss prediction adopted in this work generally has less error

and higher prediction accuracy than other models and can satisfy prediction requirements. The risk grades of the soil and water loss along the slope of Xinhe Expressway were divided into 20- and 1-year rainfall on the basis of the simulated prediction. The results showed that the percentage of slope areas with high and extremely high risks was 7.11%. These areas were mainly in the K109+500–K110+500 and K133–K139+800 sections. Therefore, relevant departments should strengthen disaster prevention and reduction efforts and corresponding water and soil conservation initiatives in these areas.

## 6  Acknowledgements

This research work was supported jointly by the Yunnan Provincial Communications Department Project [2012-272-(1)] and the Yunnan Provincial Science and Technology Commission Project (2014RA074).

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
