# Peer review of "Dangerous degree forecast of soil loss on highway slopes in"

_Natural Hazards and Earth System Sciences, 2017_

## Short Comment (SC1) · 28 Nov 2017

Figure 1. Please make an overview image to know where exactly this area is in China. An overview image helps non-local scientists to realize where the study area is.

Rainfall erosivity: Why you don't use the algorithms proposed in original RUSLE for calculating the R-factor since you have very high resolution rainfall data (Renard et al, 1997). You can also take into account the recent published Global Erosivity paper which includes also R-factor data produced in China with high resolution rainfall data (and also compare with yours): Panagos, P., Borrelli, P., Meusburger, K., Yu, B., Klik, A., Lim, K.J., Yang, J.E., Ni, J., Miao, C., Chattopadhyay, N., Sadeghi, S.H., Hazbavi,

Z., Zabihi, M., Larionov, G.A., Krasnov, S.F., Gorobets, A.V., Levi, Y., Erpul, G., Birkel, C., Hoyos, N., Naipal, V., Oliveira, P.T.S., Bonilla, C.A., Meddi, M., Nel, W., Al Dashti, H., Boni, M., Diodato, N., Van Oost, K., Nearing, M., Ballabio, C. Global rainfall erosivity assessment based on high-temporal resolution rainfall records (2017) Scientific Reports, 7 (1), art. no. 4175

Please correct the citation Panos et al 2015 and change to (by mistake you have copied the first names instead of last names): Panagos, P., Ballabio, C., Borrelli, P., Meusburger, K., Klik, A., Rousseva, S., Tadić, M.P., Michaelides, S., Hrabalíková, M., Olsen, P., Aalto, J., Lakatos, M., Rymszewicz, A., Dumitrescu, A., Beguería, S., Alewell, C. Rainfall erosivity in Europe (2015) Science of the Total Environment, 511, pp. 801-814.

---

## Author Comment (AC1) · 7 Dec 2017

Dear P. Panagos,

We are very pleased to learn from your letter about revision for our manuscript which entitled "Dangerous degree forecast of soil and water loss on highway slopes in mountainous areas using RUSLE model". We greatly appreciate your suggestion concerning improvement to this paper, and it is our honor to get your help to improve us! Thank you for your patience and advises. We have revised the manuscript according to the every single comment which made by the editor. Moreover, we have made some correction so that we hope meet with your approval. We are sending the revised manuscript according to the comments of the reviewer. We have marked the major changes in red in this revised version.(See the manuscript)

Thank you for your consideration! Sincerely yours, *Corresponding Author: Shi Qi P.S.

Reviewer's 1 comments: Comment 1: Please make an overview image to know where exactly this area is in China. An overview image helps non-local scientists to realize where the study area is.

Response 1: We greatly appreciate your valuable suggestion concerning improvement to this paper. We have followed your advise to adjusted it. Details are in following paragraph and MS.

Comments 2: Why you don't use the algorithms proposed in original RUSLE for calculating the R-factor since you have very high resolution rainfall data (Renard et al, 1997). You can also take into account the recent published Global Erosivity paper which includes also R-factor data produced in China with high resolution rainfall data (and also compare with yours): Panagos, P., Borrelli, P., Meusburger, K., Yu, B., Klik, A., Lim, K.J., Yang, J.E., Ni, J., Miao, C., Chattopadhyay, N., Sadeghi, S.H., Hazbavi, Z., Zabihi, M., Larionov, G.A., Krasnov, S.F., Gorobets, A.V., Levi, Y., Erpul, G., Birkel, C., Hoyos, N., Naipal, V., Oliveira, P.T.S., Bonilla, C.A., Meddi, M., Nel, W., Al Dashti, H., Boni, M., Diodato, N., Van Oost, K., Nearing, M., Ballabio, C. Global rainfall erosivity assessment based on high-temporal resolution rainfall records (2017) Scientific Reports, 7 (1), art. no. 4175

Response 2:Thank you for your patience and careful work! We are grateful to the reviewer for pointing out this comment. According to your comment, we explained it, details are in following paragraph.

Rainfall erosivity is an important parameter for predicting and evaluating soil erosion by using USLE/RUSLE model, it is also a common indicator of soil erosion under the effect of regional rainfall runoff. In the USLE/RUSLE model, the EI30 index should be

used to calculate the rainfall erosivity. However, in practical application, EI30 index has a high requirement for the observation and reorganization of rainfall data, which limits the popularization of the model to some extent. And because of the significant differences of natural and geographical factors, such as rainfall, terrain and vegetation in all parts of the world, the criteria for erosive rainfall are also different. Not only that, the raindrop characteristics of natural rainfall are closely related to geographical location and rain type, the structural form or coefficient of the formula for calculating the kinetic energy of rainfall is not the same.

A large number of studies have shown that: The formula selected in this paper has high stability and prediction accuracy in the southern region with rich precipitation, and also takes into account the rainfall characteristics of the study area.

References: Panagos, P., Borrelli, P., Meusburger, K., Yu, B., Klik, A., Lim, K.J., Yang, J.E., Ni, J., Miao, C., Chattopadhyay, N., Sadeghi, S.H., Hazbavi, Z., Zabihi, M., Larionov, G.A., Krasnov, S.F., Gorobets, A.V., Levi, Y., Erpul, G., Birkel, C., Hoyos, N., Naipal, V., Oliveira, P.T.S., Bonilla, C.A., Meddi, M., Nel, W., Al Dashti, H., Boni, M., Diodato, N., Van Oost, K., Nearing, M., Ballabio, C. Global rainfall erosivity assessment based on high-temporal resolution rainfall records. Scientific Reports, 7(1): 4175, 2017.

Wang, W. Z., Jiao, J. Y., Hao, X. P., Zhang, X. K., Lu, X. Q., Chen, F. Y., Wu, S. Y.: Study on Rainfall Erosivity in China. Journal of Soil and Water Conservation, (4):7-18, 1995 (in Chinese) Liu, W. Y.: Preliminary Study on R Index of Zhaotong Basin. Yunnan Forestry Science and Technology, (2):24-26, 1999 (in Chinese)

Yang, Z. S.: A Study on Erosive Force of Rainfall on Sloping Cultivated Land in the Northeast Mountain Region of Yunnan Province. SCIENTIA GEOGRAPHICA SINICA, 19(3):265-270, 1999 (in Chinese)

Comment 3: Please correct the citation Panos et al 2015 and change to (by mistake you have copied the first names instead of last names).

Response 3: Thank you very much for your reminding. At the same time, I feel very sorry for my mistakes and negligence. We have followed your advise to adjusted it. Details are in following paragraph and MS.

Panagos, P., Ballabio, C., Borrelli, P., Meusburger, K., Klik, A., Rousseva, S., Tadi′c, M.P., Michaelides, S., Hrabalíková, M., Olsen, P., Aalto, J., Lakatos, M., Rymszewicz, A., Dumitrescu, A., Beguería, S., Alewell, C.: Rainfall erosivity in Europe. Science of the Total Environment, 511:801-814, 2015.

Please also note the supplement to this comment:
https://www.nat-hazards-earth-syst-sci-discuss.net/nhess-2017-406/nhess-2017-406-AC1-supplement.pdf
* * *
**Fig. 1.**

---

## Short Comment (SC2) · 9 Dec 2017

In the introduction, In the third paragraph, in the literature review of the RUSLE model, the applicability of the model in the soil erosion of the expressway should be written, are there any related research results? What are the advantages and disadvantages of the comparison between the model and the traditional slope erosion model?

The value of C factor and P factor is almost referenced by previous research results, there are some defects in the accuracy of the data, but the author only modifies the P factor in solving the problem. How does the correction of C factor be reflected?

[Figure]

In the research area, since we are studying soil erosion, we should introduce several factors related to soil erosion, such as rainfall and soil types in the study area. As a result, only the extraction results can be retained on the extraction of natural slope catchment area, and the specific extraction method can be deleted.

———————————————————

---

## Author Comment (AC2) · 14 Dec 2017

Dear Yichao Tian, We are very pleased to learn from your letter about revision for our manuscript which entitled "Dangerous degree forecast of soil and water loss on highway slopes in mountainous areas using RUSLE model".

We greatly appreciate your suggestion concerning improvement to this paper, and it is our honor to get your help to improve us! Thank you for your patience and advises. We have revised the manuscript according to the every single comment which made by the editor. Moreover, we have made some correction so that we hope meet with your approval. We are sending the revised manuscript according to the comments of

the reviewer. We have marked the major changes in red in this revised version (See the manuscript).

Thank you for your consideration! Sincerely yours, *Corresponding Author: Shi Qi P.S.

Reviewer's #2 comments: Comment 1: In the introduction, in the third paragraph, in the literature review of the RUSLE model, the applicability of the model in the soil erosion of the expressway should be written, are there any related research results? What are the advantages and disadvantages of the comparison between the model and the traditional slope erosion model?

Response 1: Valuable suggestions! Thank you for your comments! According to your suggestion, we revised the manuscript's introduction carefully. Details are in following paragraph and MS.

We added the following section:

Wu (2014) adopted GIS and Revised Universal Soil Loss Equation (RUSLE) method to analyze the risk pattern of soil erosion in the affected road zone of Hangjinqu highway in Zhuji City, Zhejiang Province. Digital Elevation Model (DEM) data, rainfall records, soil type data, remote sensing imaging, and a road map of Hangjinqu highway were used for these GIS and RUSLE analyses. Chen (2010) according to terrain characteristics of roadbed side slope and through concrete analysis of terrain factor calculation method in Revised Universal Soil Loss Equation (RUSLE), the compatible question of terrain factor computational method of roadbed side slope is appraised and the revision method on the basis of measured data of soil erosion in subgrade side slope of Hurongxi Expressway (from Enshi to Lichuan) in Hubei Province is proposed. The results indicate that: (1) In RUSLE slope length factor can be calculated by formula of , but m should not be checked by the original method for the highway subgrade side slope because its gradient surpasses generally applicable scope of RUSLE; (2) L, slope length factor of highway subgrade side slope can be calculated by formula . Zhang (2016) investigated the spatio-temporal distribution of soil erosion in ring expressway before and after construction process, he used land use/cover map of Ningbo City in 2010, topographic map, map of North Ring expressway and field survey data was collected to derive digital elevation model (DEM). Rainfall data was collected from local hydrological station. Based on the collected data, the spatial distribution of the factors in RUSLE model was calculated, and soil erosion maps of the north ring expressway were estimated. Then, the soil erosion amount was calculated at three different stages by using RUSLE model. The results shows that: Slight erosion was dominant during preconstruction period and natural recovery period, which accounted for 98.53% and 99.73%, respectively. During construction period, mild erosion and slight erosion was the largest, which accounted for 52.5% and 35.4%, respectively. In general, soil erosion during the construction period is mainly distributed in the temporary soil ground.

The references added are as follows:

Wu, Y. L., Yan, L. J.: Impact of road on soil erosion risk pattern based on RUSLE and GIS: a case study of Hangjinqu highway, Zhuji section. ACTA ECOLOGICA SINICA, 34(19):5659-5669, 2014 (in Chinese).

Chen, Z. W., He, F., Wang, J. J.: Revises of Terrain Factor of Roadbed Side Slope in Universal Soil Loss Equation. HIGHWAY, (12):180-185, 2010 (in Chinese).

Zhang, T., Jin, D. G., Tong, G. C., Lin, J., Tang, P., Li, L. P.: Monitoring Soil Erosion in Linear Production and Construction Project Areas Based on RUSLE - A Case Study of North Ring Expressway in Ningbo City, Zhejiang Province. Bulletin of Soil and Water Conservation, 36(5):131-135, 2016 (in Chinese).

The advantages and disadvantages of the comparison between the model and the traditional slope erosion model: (1)USLE model: Advantage: TheÂămodelÂăexpressionÂăformÂăisÂăsimple, provided a model for world's soil erosion model

Disadvantage: Because it is an empirical model based on the calculation of soil erosion in small watershed, the physical process of soil erosion is not considered, in addition, it

is also found that this model is not suitable for planting, contourÂătillage. At the same time, it does not take into account the effect of surface runoff, which greatly reduces the effectiveness of the equation applied to the interception measures.

(2)RUSLE model: Advantage: The generality of the computational factor is strengthened. For the rainfall erosivity factor (R), the meteorological data in the wider region are used, and the accuracy is greatly improved. For soil erodibility factors (K), the RUSLE model further considers seasonal variations, such as freezing thawing, soil moisture and consolidation. For the topographic factors (LS), a new equation is added to the RUSLE to reflect the relative proportion of the rill erosion and the inter ditch erosion, and the complex changes of the slope are involved. For vegetation coverage factors (C), RUSLE considers their variation patterns and interactions in the whole year, as well as the effect of climatic conditions on the decomposition of mulch, and the effect of farming system. For the management measures of soil and water conservation factors (P), RUSLE not only considered the above hydrological and geomorphic characteristics in detail, but also analyzed various soil and water conservation measures in detail, such as the influence of equal tillage and strip farming on erosion. The application range of RUSLE is extended from the original two-dimensional field to three dimensional, which can simulate the spatial evolution characteristics of geomorphic landscape. In addition, the reliability of the forecast is also greatly strengthened.

Disadvantage: The application of RUSLE has the problems of difficulty in computing and inefficiency.

(3)WEPP model: Advantage: The WEPP model (slope version) is a new generation of computer models for prediction of soil erosion, it can predict the amount of soil loss and its dynamic change at any moment on the slope in the rainfall process, the application range is very wide, and it is the soil erosion model which describes the most related parameters of water erosion so far.

Disadvantage: The WEPP model needs more input parameters, and the acquisition

and correction are difficult. In particular, the determination of some important parameters can not be experimentally determined. The non measured "estimation" method must be adopted, and sometimes the default value of its modification is also needed. The WEPP model can only be used for the prediction of rill, interrill erosion and shallow gully erosion, and can not be used for trenches, river erosion and ditch erosion, it can not be used for gully erosion, stream erosion and ditch erosion, in particular, gravity erosion and sediment deposition and re-handling process in the slope and channel are ignored, or only for simplification. At present, the slope version has been widely used in other countries, and the results of the simulated forecast are better, while the basin version is still limited to the last stage subbasin.

(4)EUROSEM modelïijŽ Advantage: The biggest advantage of this model is dynamics, it is a dynamic formulation model. it can be simulated as small as a single field, and can be large to a small watershed. It has a good physical basis, and the erosion can be divided into rill erosion and interrill erosion.

Disadvantage: The accuracy of the model is limited by field or experimental conditions.

(5)ANSWERS model: Advantage: The degree of structurization of the model is high.

Disadvantage: The model's erosion module is largely empirical and only simulates the migration process of the total sediment.

Comments 2: The value of C factor and P factor is almost referenced by previous research results, there are some defects in the accuracy of the data, but the author only modifies the P factor in solving the problem. How does the correction of C factor be reflected?

Response 2: We are grateful to the reviewer for pointing out this comment. According to your comment, we will explain the problem, details are in following paragraph.

In this study, we modified the C factor. The NDVI was used to calculate the vegetation coverage. After that, the C factor was estimated. Then, the vegetation coverage

data were corrected by selecting a sample plot every 2 km along the study area for investigation. Finally, accurate vegetation coverage data were obtained.

Comment 3: In the research area, since we are studying soil erosion, we should introduce several factors related to soil erosion, such as rainfall and soil types in the study area.

Response 3:We completely agree with your comment! We have followed your advise to adjusted it. Details are in following paragraph and MS.

Xinhe expressway is located in Honghe Prefecture, Hekou County. The climate type belongs to the humid and hot climate of the subtropical monsoon forest. Due to the monsoon activity, the climatic characteristics of the study area are dry and wet seasons change clearly, among them, the wet season (also known as the rainy season) starts from May to mid October every year, the rest of the time for the dry season, the wet season has the characteristics of heavy rainfall, concentration of precipitation, etc. The difference in precipitation is about 400mm-2000mm, and most of the regions are between 800-1800mm.

The soil types along the Xinhe expressway are mainly red soil, leached cinnamon soil, gray forest soil and gray cinnamon soil. The plant division of the southern part of the Xinhe expressway is a tropical rainforest and a tropical seasonal rain forest, the vegetation division in the northern part of the region belongs to the south subtropical monsoon evergreen broad-leaved forest. In recent years, the original vegetation is mostly cultivated land, mainly planting rubber, bananas, pineapple, pomegranate and so on, and the tropical rainforest is fragmentary.

Comment 4: As a result, only the extraction results can be retained on the extraction of natural slope catchment area, and the specific extraction method can be deleted.

Response 4: Thank you for your comments! We greatly appreciate your valuable suggestion concerning improvement to this paper. We will explain the problem. Details are

in following paragraph.

In this study, the revised universal soil loss equation (RUSLE) is used as a prediction model for soil and water loss on slopes, combined with GIS and remote sensing technology. The methods of field survey and runoff observation are used, on the basis of fully considering the differences between the model parameters of the artificial slopes and the natural slopes of the expressway. The catchment area is considered a prediction unit. The prediction units of the artificial and natural slopes are classified, and the soil and water loss of each slope is predicted in real time, thus reflecting the soil and water loss of expressway slope accurately. The method of slope element division it is the focus of this study, so it needs to be in detail.

Please also note the supplement to this comment:
https://www.nat-hazards-earth-syst-sci-discuss.net/nhess-2017-406/nhess-2017-406-AC2-supplement.pdf

---

## Referee Comment (RC1) · Anonymous Referee #1 · 2 Mar 2018

This paper asks a novel and well-justified question about how to estimate soil loss on highway slopes. The authors state that most work to date on this area has been on natural slopes, and present some striking statistics about the area covered by highway slopes. However, the paper is extremely hard to read - it assumes a lot of prior knowledge about the RULSE model, does not define variables clearly and is written in long, dense and technical paragraphs. In its current draft state, it is hard to address the scientific quality of the paper in depth because it is hard to read and follow. For this reason, I would recommend major revisions to work on the communication of the paper before the content can be reviewed in detail.

[Figure]

I have listed suggested revisions below in the approximate order in which they appear in the paper. These are a mix of medium and minor level revisions.

General - The English language needs work in places - at times, the language is very dense and sometimes the incorrect tense is used. Please proofread or consult a proof-reader.

The abstract is very long, this should be shortened to around 300 words.

The introduction assumes a large amount of prior knowledge about the RULSE model and the parameters that go into it. Please give more background. Imagine the audience was a highway manager, please state more clearly what the implications of previous research are in practical terms.

Paragraph starting on line 57. Many of the sentences need a citation to existing peer review or grey literature.

Line 62. 50 - 70 thousand should be changed to 50,000 - 70,000 m2

General - please ensure numeric units are described consistently and using SI units.

General - please only use author's last name for citations. E.g., line 73 should be Tresch et al. (date), not Tresch S et al (date).

Line 80 and 81, please ensure variables such as LS and S are defined

Line 84. This sentence needs restructuring as when you say 'this study', it sounds as though you are talking about primary research.

Paragraph starting on line 73 is very long (in excess of a page), and should be broken down into shorter paragraphs.

Line 96. Please state what 'K' is.

Line 101 Please state what 'C' is.

Paragraph on line 122 needs more citations to existing literature. E.g., the sentence on

line 125 should have a citation. Line 142, Please define what C and P factors are. This is confusing to talk about multiple parameters in the introduction without describing what they are.

Section 2 - please add references throughout this paragraph. Many of the statements made should have a citation e.g., about the seasonal regime of the area.

Section 3 - please use the Harvard reference system to cite each dataset

Section 3.1.2 Is the 's-shaped sampling method' already established in the literature? If so, please cite the literature. If not, please give more detail on this method.

Section 3.1.4 Is the imagery pixel size 8 m x 8m? If so, please state it in this way.

Section 2.2 give a citation for the RULSE equation.

Section 3 general - the sub-section numbering switches from 3.X.X to 2.X.X - please check the numbering is in order

[The following sections no longer have line numbering, so I will try to give detail on where I am referring to]

Section 4.1 The sentence referring to ArcGIS software needs further explanation. Why was it necessary to vectorise the data? What does the vectorisation have to do with soil erosion prediction?

Section 4.1 sentence starting 'The natural and artificial slope catchment watershed..' - the final statement 'such as property' needs clarification

Section 4.1.1 please provide a citation for the ArcGIS tool

Section 4.1 general - some of this information feels like it belongs in the methodology rather than results

The results section of the paper is highly technical, and currently would be more suited to an engineering type journal. Please consider the title of the journal and the likely

audience. Try to tell a more logical story of why certain methods have been used, and think about what information is useful to the reader. Some information could possibly go in supplementary material.

Section 4.2.1 Please state what the interpolation calculations are

Table 1 and Table 3 Please make clearer what the section of expressway names actually refer to. I.e., how would I find 'K83+500∼K84+900' on a map?

General. When referring to equations, do not refer to the authors by name and then cite. Also, check the spelling of author names is consistent and do not use author first names. For example in section 4.2.3, a sentence reads: If the slope is less than $18°$, then the formula proposed by McCool et al (Mccol et al., 1987) was used. This should be rewritten as something like: If the slope is less than $18°$, then the formula of McCool et al. (1987) is used

Section 4.2.5 How was NDVI calculated? What data was used?

Section 4.3 How were the field measurements taken in 2014?

Section 4.3 Please state why the RMSE is within an acceptable range

End of Section 4.3 The idea that the model may be defective needs further discussion. What might the uncertainties be? Why are the difference between monitoring and analogue spatially variable?

Page 27. Can you provide any recommendations for how soil and water conservation measures could be rationally adjusted?

Discussion - please comment on how meaningful these methods and results might be in other locations? Is this a site-specific study or does it have wider relevance elsewhere?

Conclusion. Please write this in paragraphs, not numbered sections. Please define variables again.

---

## Referee Comment (RC2) · Anonymous Referee #2 · 5 Mar 2018

Highway is booming in China in recent twenty years. Hence how to control soil erosion from the subgrade slope is an important issue of the road safety. From this point, the topic is valuable. The manuscript is also readable although still many language errors exist. Nevertheless, the structure of the paper is poor. Instead of a research paper, the manuscript looks like a scientific report. Too many original data are shown in the manuscript but short of in-depth discussions. I strongly suggest the authors rewrite the manuscript. Some of errors or problems are shown as follows. (1) Abstract. ïČŸ Lines 29-30: The word "show" should be replaced with the word "shows". There are many grammar errors throughout the manuscript. I think the paper may be polished by a soil scientist whose native language is English. ïČŸ Lines 28-30: Is the error of

the erosion rate per year? I think the unit may be t.km-2. a-1. So are the units of the erosion rates appear through the manuscript. ïČŸ Lines 30-31: The unit of the root mean square error of the soil loss is same to the unit of soil loss. Here the unit is t.km-2. a-1. ïČŸ Lines 33-37: You said "the percentage of prediction units with no risk of erosion is 78% and that with mild soil erosion risk is 15.92%". Do you mean the high and extremely high risk is 1-78%-15.92%=6%? If my deduction is right, I think no large difference exists between this risk and that described in line 37, 7.11%. (2) Introduction ïČŸ Lines 60-61: "In the next 20–30 years, expressways in China will measure more than 40 thousand km". However I find in the network that "since 2016, the total length of the highway in China is about 131 thousand km". I do not know which length is correct? ïČŸ Line 131: ". . .the main type of soil erosion in the study area was gully erosion". Could USLE or RUSLE be applied in the gully erosion? Other minor errors are shown as follows: ïČŸ Lines 62-63: I am lost! Might you kindly let me you the area attacked by soil erosion in this sentence? ïČŸ Line 68: The word "affect" should be "affects" or "influences". ïČŸ Line 78: What's the meaning of the following sentence: ". . .have explored the process of using the RUSLE model"? ïČŸ Lines 81-82: I am lost. ïČŸ Line 85: 20%-90% should be revised as follows: 20-90%. ïČŸ Lines 125-126: I can not understand. ïČŸ Line 141: "According to studies at home and abroad". Please remember you are writing a paper for the international journal instead of a scientific report in Chinese! ïČŸ Line 158: "In this study, " may be added in front of the sentence "A suitable. . ." ïČŸ Lines 161-162: I am lost. (3) Study area ïČŸ Line 193: Figure 1 may be merged to Figure 4 in page 11. (4) Materials and method ïČŸ All of the subtitle of the part, including 3.1, 3.1.1-3.3.4, and 3.2, may be erased. Attention please, some of the subtitles in the manuscript are wrong. (5) Results and analysis ïČŸ From page 9 to page 22: The part looks like a scientific report instead of a research paper. Except the original experimental data, hardly any in-depth discussion exists. ïČŸ Page 23: The calculated results have not been compared with the results described in other references. Also I do not know why the errors emerge.

Some of other problems are shown as follows: ïČŸ Where is the line number? ïČŸ

Segment 4.1: The segment is a method, and it may be simplified and removed to Part 3 Materials and method. ïČŸ Segment 4.2.5 in page 21: Why were the landform factors calculated according to the natural slopes (4.2.3) and the artificial slope (4.2.4), but was the vegetation cover calculated only in one situation (4.2.5)? ïČŸ Line 4 of page 28: "A cement box should be added in the soil a year": Is this the only conservation practice we should adopt? References ïČŸ Page 29: "Liu, X. Y.: Study on the slope stability and its rheological influence in Mountain highway. Central South University, 2013": Is it a paper in Chinese journal or international journal, or an academic dissertation in Chinese? ïČŸ Page 30: "Bosco, C., De Rigo, D., Dewitte, O., Poesen, J., and ..." should be removed forward as the second reference. ïČŸ Page 30: "Wang, H. J., Yang, Y., andWang, W. J.: Prediction of Soil Loss Quantity...": Is it a paper in Chinese journal?

Please also note the supplement to this comment:
https://www.nat-hazards-earth-syst-sci-discuss.net/nhess-2017-406/nhess-2017-406-RC2-supplement.pdf

---

## Author Comment (AC3) · 15 Mar 2018

The comment was uploaded in the form of a supplement:
https://www.nat-hazards-earth-syst-sci-discuss.net/nhess-2017-406/nhess-2017-406-AC3-supplement.pdf

[Figure]

**Fig. 1.** Overview of study region

![After rainfall, soil erosion has occurred on the slope]

**Fig. 2.** Soil erosion produced by rainwash on slope

**Natural slope**

**Artificial slope**

Topography

Ridge line ✚ Valley line

The type
of slope

Subgrade slope

Cutting slope

slope

The slope is 1:1.5

The slope is 1:1.0

Agrotype

Gray cinnamon soil

Lateritic red soil

Yellow brown earth

Leached cinnamon soil

Gray forest soil

Slope
protection
measures

Cement frame
+
plant slope

Six arris brick revetment

Land use
types

Farmland

woodland

Construction land

Barren land

Plant
species

macrophanerophytes

shrub

grass and trees

shrub and herb

arbor-shrub-grass
compound land

Soil and water
conservation
measures

Horizontal step

Terraced fields

Contour strip cropping
etc.;

**814 prediction units**

**422 prediction units**

**Natural slope
catchment area**

**Artificial slope
catchment area**

**Fig. 3.** Prediction unit division

**Fig. 4.** Division results of prediction units (K133–K139)

**Fig. 5.** Interpolation results of secondary rainfall for June 5, 2014

**Fig. 6.** Interpolation results of I30 for June 5, 2014

**Legend**

⊙ **Stake number**

⚑ **Terminal point**

— **Xinhe expressway**

0    .5    1         2
                        km

**R ( artificial slope)**
- 138.98-146.92
- 146.92-158.76
- 158.76-172.07
- 172.07-187.69
- 187.69- 205.53

**R (natural slope)**
- 131.49-146.31
- 146.31-157.65
- 157.65-171.11
- 171.11-187.55
- 187.55-204.14

**Fig. 7.** Spatial distribution map of rainfall erosivity factors (K127–K139+800)

**Fig. 8.** Spatial distribution map of topographic factors (K134–K139)

**Fig. 9.** Vegetation coverage along Xinhe Expressway

**Fig. 10.** Spatial distribution map of cover and management practice factor

Rainfall (mm)

| | | |
|---|---|---|
| ■ 108.09-113.25 | ■ 118.59-123.66 | ■ 128.74-133.08 |
| ■ 113.25-118.59 | ■ 123.66-128.74 | ■ 133.08-138.69 |

● Stake number

Xinhe expressway

**Fig. 11.** Rainfall interpolation results under 20-year return

**K83+500**

**K88**

**K93**

**K98**

**K103**

**K109**

**K115**

**K121**

**K127**

**K133**

**K139+800**

103°37'0"E  103°45'30"E  103°54'0"E

22°46'0"N

22°37'30"N

N
W    E
S

0   2.5   5      10
km

$I_{30}$/(mm·h⁻¹)

| | | |
|---|---|---|
| ■ 42.10–46.01 | ■ 46.01–50.42 | ■ 54.68–58.84 |
| | ■ 50.42–54.68 | ■ 58.84–62.00 |

● Stake number

Xinhe expressway

**Fig. 12.** Rainfall intensity interpolation results under 20-year return

**Fig. 13.** Risk analysis of soil and water loss under 20-year rainfall conditions

**Fig. 14.** Risk analysis of soil and water loss under 1-year rainfall amount

**Supplement:**

**Response to Decision Letter**

**Dear Dr. Paolo Tarolli and referee,**

We are very pleased to learn from your letter about revision for our manuscript which entitled _"Dangerous degree forecast of soil and water loss on highway slopes in mountainous areas using RUSLE model"._

We greatly appreciate reviewer's thoughtful suggestions concerning improvement to our paper. These comments are all valuable and very helpful for revising and improving our paper, as well as the important guiding significance to our researches.

Thank you for your consideration!

Sincerely yours,

*Corresponding Author: Shi Qi

P.S.

=========================================================================

**Response to review's comments for nhess-2017-406-RC1**

**Reviewer's #1 comments:**

**Comment 1:** This paper asks a novel and well-justified question about how to estimate soil loss on highway slopes. The authors state that most work to date on this area has been on natural slopes, and present some striking statistics about the area covered by highway slopes. However, the paper is extremely hard to read-it assumes a lot of prior knowledge about the RULSE model, does not define variables clearly and is written in long, dense and technical paragraphs. In its current draft state, it is hard to address the scientific quality of the paper in depth because it is hard to read and follow. For this reason, I would recommend major revisions to work on the communication of the paper before the content can be reviewed in detail.

**Response 1:** First of all, we are very grateful for your affirmation of our research work. Most important of all, we greatly appreciate your valuable suggestion concerning improvement to this paper. We have made a detailed revision of the grammar and structure of the manuscript, hoping to meet your requirements. The revised place is marked with red and details are in manuscript.

**Comment 2:** I have listed suggested revisions below in the approximate order in which they appear in the paper. These are a mix of medium and minor level revisions.

**Response 2:** Thank you for your patience and careful work! We have made correction according to your comments. Details are in the manuscript.

**Comment 3:** General-The English language needs work in places-at times, the language is very dense and sometimes the incorrect tense is used. Please proofread or consult a proofreader.

**Response 3:** Thank you for your comments! We have followed your advice to adjust it. We have had the manuscript polish with a professional assistance in writing. Details are in the manuscript.

**Comment 4:** The abstract is very long, this should be shortened to around 300 words.

**Response 4:** Thank you for your valuable advice. According to your comment, we have followed your advice to adjust it. Details are in following paragraph and MS.

Many high and steep slopes are formed by special topographic and geomorphic types and mining activities during the construction of mountain expressways. Severe soil erosion may occur under heavy rainfall conditions. Therefore, predicting soil and water loss on highway slopes is important in protecting infrastructure and human life. This work studies Xinhe Expressway, which is in the southern edge of Yunnan–Guizhou Plateau, as the research area. The revised universal soil loss equation is selected as the prediction model of the soil and water loss on the slopes. Moreover, geographic information system, remote sensing technology, field survey, runoff plot observation testing, cluster analysis, and cokriging are adopted. The partition of the prediction units of the soil and water loss on the expressway slope in the mountain area and the spatial distribution model of the linear highway rainfall are studied. In view of the particularity of the expressway slope in the mountain area, the model parameter factor is modified and the risk of soil and water loss along the mountain expressway is simulated and predicted under 20- and 1-year rainfall return periods. The results are as follows. (1) Considering natural watershed as the prediction unit of slope soil erosion can represent the actual situation of the soil and water loss of each slope. The spatial location of the soil erosion unit is realized. (2) An analysis of the actual observation data shows that the overall average absolute error of the monitoring area is 33.24 $t\cdot km^{-2}\cdot a^{-1}$, the overall average relative error is 33.96%, and the overall root mean square error is between 20.95 and 65.64, all of which are within acceptable limits. The Nash efficiency coefficient is 0.67, which shows that the prediction accuracy of the model satisfies the requirements. (3) Under the condition of 1-year rainfall, we find through risk classification that the percentage of prediction units with no risk of erosion is 78%. Results show that soil erosion risk is low and therefore does not affect road traffic safety. Under the 20-year rainfall condition, the percentage of units with high risk and extremely high risk is 7.11% and mainly distributed on the K109+500–K110+500 and K133–K139+800 sections. The prediction results can help adjust the layout of the water and soil conservation measures in these units.

**Comment 5:** The introduction assumes a large amount of prior knowledge about the RULSE model and the parameters that go into it. Please give more background. Imagine the audience was a highway manager, please state more clearly what the implications of previous research are in practical terms.

**Response 5:** Valuable suggestions! Thank you for your comments. We have followed your advice to adjust it. Details are in following paragraph and manuscript.

Water and soil erosion caused by engineering construction is an important form after agricultural cultivation and forestry deforestation, the amount of soil erosion produced by the embankment slope occupies a large proportion in the whole project. It is not only related to the feasibility and cost of the project, but also has aroused great interest and attention. Yang (2001) investigated the behavior of soil erosion on the slope of a railway embankment during construction by comparing artificial and natural rainfalls on the special Qinhuangdao–Shenyang line of passenger trains. The results showed that the main type of soil erosion in the study area was gully erosion, which caused more soil erosion than surface erosion did; in addition, the principal factor causing soil erosion on the slope was the amount of precipitation and the width of the embankment. Wang (2005) established several experimental standardized spots for soil loss collection on the side slopes of the Xiaogan–Xiang fan freeway under construction and installed an on-the-spot auto-recorder of rainfall. The data collected were used for the revision of the main parameters R (rainfall and runoff) and $K$ (erodibility of soil) of the USLE, which is widely applied to forecast soil loss quantity in plowlands and predict the soil loss quantities of different types of soil on side slopes disturbed by engineering treatment (Wang et al., 2005). It can not only be applied to the prediction of disturbed soil loss in expressway construction, but also improve the prediction accuracy. It provides scientific support for relevant units or personnel to take reasonable preventive measures.

**Comment 6:** Paragraph starting on line 57. Many of the sentences need a citation to existing peer review or grey literature.

**Response 6:** Thank you for your comments! We have followed your advice to adjust it. Details are in following paragraph and manuscript.

The slope is the most fragile part of an expressway in a mountain area (Yuan et al., 2017; Mori et al., 2017). During the rainy season, soil erosion is easily caused by rainwash and leads to a worrisome extent of damage (Figure 2). At present, China's highway industry is still in a period of rapid development. By the end of 2014, the total mileages of highway network exceeds 4,400,000 kilometers, and the expressway's mileage is 112,000 kilometers (Zhou et al., 2016). According to statistics, with the development of highway construction in China, slope areas reach 200–300 million $m^2$ each year. In the next 20–30 years, expressways in China will measure more than 40,000 km. For every kilometer of a highway, the corresponding bare slope area formed measures 50,000–70,000 $m^2$. The annual amount of soil erosion is 9,000 $g/m^2$, which causes 450 t of soil loss every year (Chen 2010). Compared with the soil and water loss on forestlands and farmlands, which on subgrade slopes is special. Forestlands and farmlands are generally formed after years of evolution and belong to the native landscape. Most slopes are gentle and stable (Kateb et al., 2013). Traditional soil and water conservation research focuses on slopes with 20% grade or below, but the highway subgrade slope of steep slopes is generally greater than 30% (Zhou 2010). Soil erosion on subgrade side slopes affects not only soil and water loss along the highway but also road operation safety (Gong and Yang 2016; Jiang et al., 2017). Therefore, soil erosion on the

side slopes of mountain expressways must be studied to control soil erosion, improve the ecological environment of expressways, and realize sustainable land utilization (Wang et al., 2005; Yang and Wang 2006).

**Reference:**

Yuan, C., Yu, Q. H., You, Y. H., Guo, L.: Deformation mechanism of an expressway embankment in warm and high ice content permafrost regions. Applied Thermal Engineering 121: 1032-1039, 2017.

Mori, A., Subramanian, S. S., Ishikawa, T., Komatsu, M. A Case Study of a Cut Slope Failure Influenced by Snowmelt and Rainfall. Procedia Engineering, 189: 533-538, 2017.

Zhou, R. G., Zhong, L. D., Zhao, N. L., Fang, J., Chai, H., Jian, Z., Wei, L., Li. B.: The Development and Practice of China Highway Capacity Research. Transportation Research Procedia, 15: 14-25, 2016.

Kateb, H. E., Zhang, H. F., Zhang, P. C., Mosandl, R. Soil erosion and surface runoff on different vegetation covers and slope gradients: A field experiment in Southern Shaanxi Province, China. Catena, 105(5): 1-10, 2013.

**Comment 7:** Line 62. 50 - 70 thousand should be changed to 50,000-70,000 m$^2$.

**Response 7:** Thank you for your patience and careful work. We are grateful to the reviewer for pointing out this comment, we have made correction according to your comments. Details are in following paragraph and MS.

For every kilometer of a highway, the corresponding bare slope area formed measures 50,000–70,000 m$^2$.

**Comment 8:** General-please ensure numeric units are described consistently and using SI units.

**Response 8:** Thank you for your careful work. We have carefully corrected these mistakes according to your comment. Details are in the MS.

**Comment 9:** General-please only use author's last name for citations. E.g., line 73 should be Tresch et al. (date), not Tresch S et al (date).

**Response 9:** Thank you for your comments! We have followed your advise to adjusted it. Details are in following paragraph and MS.

Tresch et al. (1995) believed that the topographical factor *LS* is one of the main factors for soil erosion modeling within the RUSLE environment.

**Comment 10:** Line 80 and 81, please ensure variables such as *LS* and *S* are defined.

**Response 10:** Thank you for your comments! We have followed your advice to adjust and explained it. Details are in following paragraph and MS.

**Explain:** *L* is the slope length factor. *S* is the steepness factor. *LS* is slope length/slope

steepness factor (dimensionless).

Tresch et al. (1995) believed that the slope length/slope steepness factor $LS$ is one of the main factors for soil erosion modeling within the RUSLE environment. Various steepness factors ($S$) exist for the most used soil erosion modeling environment and significantly influence calculated erosion values.

**Comment 11:** Line 84. This sentence needs restructuring as when you say "this study", it sounds as though you are talking about primary research.

**Response 11:** We greatly appreciate your valuable suggestion concerning improvement to this paper. We have followed your advice to adjust it. Details are in following paragraph and MS.

Eighteen plot measurements on transects along slopes ranging from 20–90% in steepness were used in this study to qualitatively assess the most suitable S factors for steep subalpine slopes.

**Comment 12:** Paragraph starting on line 73 is very long (in excess of a page), and should be broken down into shorter paragraphs.

**Response 12:** Thank you for your careful reading of our manuscript and point out this shortcoming, we have followed your advice to adjust it. Details are in following paragraph and MS.

The use of revised universal soil loss equation (RUSLE) models as predictive tools for the quantitative estimation of soil erosion has been maturing for a long time. The range of application of these models involves nearly every aspect of soil erosion. In addition, many scholars have explored the process of using RUSLE models and combined research objects to correct the parameter values in these models, thus improving simulation accuracy:

Tresch et al. (1995) believed that the slope length/slope steepness factor $LS$ is one of the main factors for soil erosion modeling within the RUSLE environment. Various steepness factors ($S$) exist for the most used soil erosion modeling environment and significantly influence calculated erosion values. All existing S factors are derived only from gentle slope inclinations of up to 32%. Many cultivated areas, particularly in Switzerland, are steeper than this critical value. Eighteen plot measurements on transects along slopes ranging from 20–90% in steepness were used in this study to qualitatively assess the most suitable S factors for steep subalpine slopes. Results showed that a first selection of an S factor was possible for slopes above the critical 25% steepness. Rick D (2001) found that using universal soil loss equation (USLE) and RUSLE soil erosion models at regional landscape scales is limited by the difficulty of obtaining an $LS$ factor grid suitable for use in geographic information system (GIS) applications. Therefore, he described the modifications applied to the previous arc macro language (AML) code to produce a RUSLE-based version of the $LS$ factor grid. These alterations included replacing the USLE algorithms with their RUSLE counterparts and redefining assumptions on slope characteristics. Finally, in areas of western USA where the models were tested, the RUSLE-based AML program produced $LS$ values that were

roughly comparable to those listed in the RUSLE handbook guidelines. Silburn's (2011) research showed that estimating $K$ from soil properties (derived from cultivated soils) provided a reasonable estimate of $K$ for the main duplex soils at the study site as long as the correction for the undisturbed soil was used to derive $K$ from the measured data and to apply the USLE model.

However, methods used to fit the parameters affected the results, and minimizing the sum of the squares of errors in the soil losses provided better results than fitting an exponential equation did. Yang (2014) found that the $C$ factor value can be determined as a function of fractional bare soil and ground cover derived from MODIS data at regional or catchment scales. The method offers a meaningful estimate of the $C$ factor, thus indicating ground cover impact on soil loss and erosion hazard areas. The method is better than the commonly used techniques, which are based only on green vegetation (e.g., normalized difference vegetation index, NDVI). Thus, the study provided an appropriate approach to estimating the $C$ factor in hillslope erosion modeling in New South Wales, Australia, using emerging fractional vegetation cover products. This approach simply and effectively mapped the spatial and temporal distribution of the RUSLE cover factor and hillslope erosion hazard in a large area. The methods and results described in this article are valuable for understanding the spatial and temporal dynamics of hillslope erosion and ground cover. According to a study by Kinnell PIA (2014), runoff production, which is spatially uniform, is often inappropriate under natural conditions, where infiltration is spatially variable. The use of an upslope slope length that varies with the ratio of the upslope runoff coefficient to the runoff coefficient for the area down to the downslope boundary of the segment in modifications of the RUSLE approach produces only minor variations in soil loss compared with those predicted using the standard RUSLE approach when the runoff is spatially variable and the number of segments increases. On the contrary, the USLE-M approach provides predictions of soil loss that are influenced strongly by runoff when runoff varies in space and time. Therefore, an increase in the runoff through a segment causes an increase in soil loss, whereas a decrease in the runoff through a segment or cell results in a decrease in soil loss.

**Comment 13:** Line 96. Please state what "$K$" is.

**Response 13:** Thank you for your comments! According to your comments, we explained it. Details are in following paragraph and MS.

Explain: $K$ is soil erodibility factor.

Silburn's (2011) research showed that estimating soil erodiblity factor ($K$) from soil properties (derived from cultivated soils) provided a reasonable estimate of $K$ for the main duplex soils at the study site as long as the correction for the undisturbed soil was used to derive $K$ from the measured data and to apply the USLE model.

**Comment 14:** Line 101. Please state what "$C$" is.

**Response 14:** Thank you for your comments! According to your comments, we explained it. Details are in following paragraph and MS.

**Explain:** *C* is cover-management factor (dimensionless)

Yang (2014) found that the cover-management factor (*C*) value can be determined as a function of fractional bare soil and ground cover derived from MODIS data at regional or catchment scales.

**Comment 15:** Paragraph on line 122 needs more citations to existing literature. E.g., the sentence on line 125 should have a citation.

**Response 15:** We greatly appreciate your valuable suggestion concerning improvement to this paper. We have followed your advice to adjust it. Details are in following paragraph and MS.

In general, these studies are mainly limited to sloping fields (Tresch S and others 1995; Rick D and others 2001; Silburn 2011; Yang 2014; Kinnell 2014).

However, the accumulation degree of soil and water loss in highways cannot satisfy the requirements of model development (Xu et al., 2009; Bakr et al., 2012).

**References:**

Xu, X. L., Liu, W., Kong, Y. P., Zhang, K. L., Yu, B. F., Chen, J. D.: Runoff and water erosion on road side-slopes: Effects of rainfall characteristics and slope length. Transportation Research Part D: Transport and Environment, 14(7): 497-501, 2009.

Bakr, N., Weindorf, D. C., Zhu, Y. D., Arceneaux, A. E., Selim, H. M.: Evaluation of compost/mulch as highway embankment erosion control in louisiana at the plotscale. Journal of Hydrology, s 468-469(6): 257-267, 2012.

**Comment 16:** Line 142, please define what *C* and *P* factors are. This is confusing to talk about multiple parameters in the introduction without describing what they are.

**Response 16:** Thank you very much! According to your comment, we have followed your advice to adjust it. Details are in following paragraph and MS.

**Explain:** *C* is cover-management factor (dimensionless); and *P* is the support practice factor (dimensionless).

In using the RUSLE model, most of the research on the cover-management (*C*) and support practice (*P*) factors was conducted by referring to previous research results and data accuracy is often poor.

**Comment 17:** Section 2 - please add references throughout this paragraph. Many of the statements made should have a citation e.g., about the seasonal regime of the area.

**Response 17:** We are grateful to the reviewer for pointing out this comment. According to your comment, we have followed your advice to adjust it. Details are in following paragraph and MS.

Xinhe Expressway is in the southern margin of the Yunnan–Guizhou Plateau, which is in southeast Yunnan Province, Honghe Prefecture, and Hekou County. This highway was the first in

Yunnan to cross the border, thereby becoming an important communication channel between China and Vietnam and obtaining important strategic and economic value. The highway is at longitude 103° 33′ 45″–103° 58′ 32″ and latitude 22° 31′ 19″–22° 51′ 48″. The expressway stretches roughly from northwest to southeast, and the total length is 56.30 km. The climate type belongs to subtropical mountain, seasonal monsoon forest, and humid heat climate categories. Between May and the middle of October, the area experiences wet season characterized by abundant rainfall, concentrated precipitation, and increased rain at night (Fei et al., 2017; Zhang et al., 2017).

**References:**

Fei, X. H., Song, Q. H., Zhang, Y. P., Liu, Y. T., Sha, L. Q., Yu, G. R., Zhang, L. M., Duan, C. Q., Deng, Y., Wu, C. S., Lu, Z. Y., Luo, K., Chen, A. G., Xu, K., Liu, W. W., Huang, H., Jin, Y. Q., Zhou, R. W., Grace, J.: Carbon exchanges and their responses to temperature and precipitation in forest ecosystems in Yunnan, Southwest China. Science of The Total Environment, 616: 824-840, 2017.

Zhang, H., Liao, X. L., Zhai, T. L.: Evaluation of ecosystem service based on scenario simulation of land use in Yunnan Province. Physics and Chemistry of the Earth, Parts A/B/C. 2017.

**Comment 18:** Section 3 - please use the Harvard reference system to cite each dataset.

**Response 18:** We greatly appreciate your valuable suggestion concerning improvement to this paper. We have followed your advise to adjusted it. Details are in following paragraph and MS.

**2.1.1 Meteorological data**

Rainfall data from 2014 were obtained from Hekou Yao Autonomous County, Pingbian Miao Autonomous County, Jinping Miao Yao Autonomous County, and the meteorological department of Mengzi. The rainfall data type was in 5 min format. Meanwhile, two automatic weather stations were established along Xinhe Expressway to gather weather data during the 2014 experiment. Meteorological data was provided by the China Meteorological Data Network covered the period of 1959–2015 (http://data.cma.cn/site/index.html).

**2.1.4 Image data**

The remote sensing images used in this study were derived from 8m hyperspectral images produced by GF-1 satellite (http://www.rscloudmart.com/).

**Comment 19:** Section 3.1.2 Is the "S"-shaped sampling method already established in the literature? If so, please cite the literature. If not, please give more detail on this method.

**Response 19:** Thank you for your comments. We have followed your advise to adjusted it. Details are in following paragraph and MS.

Five mixed soil samples were obtained from one slope using the "S"-shaped sampling method (Shu et al., 2017).

**Reference:**

Shu, Z. Y., Wang, J. Y., Gong, W., Lv, X. N., Yan, S Y., Cai, Y., Zhao, C. P.: Effects of compound management in citrus orchard on soil micro-aggregate fractal features and soil physical and chemical properties. Journal of Nanjing Forestry University (Natural Sciences Edition), 41(5): 92-98, 2017.

**Comment 20:** Section 3.1.4 Is the imagery pixel size 8 m x 8m? If so, please state it in this way.

**Response 20:** Thank you for your comment! According to your comment, we explained it, details are in following paragraph.

It refers to the image of a multi spectral resolution of 8 meters.

**Comment 21:** Section 2.2 give a citation for the RULSE equation.

**Response 21:** Thank you for your comments. We have followed your advise to adjusted it. Details are in following paragraph and MS.

The RUSLE equation (Renard et al., 1997) was used to predict the soil and water loss on the side slopes of Xinhe Expressway. The RUSLE equation considers natural and anthropogenic factors that cause soil erosion to produce comprehensive results. Various parameters are easy to calculate, and the calculation method is relatively mature. The RUSLE model is suitable for soil erosion prediction in areas where the physical model is not needed. See Formula (1).

**Comment 22:** Section 3 general - the sub-section numbering switches from 3.X.X to 2.X.X - please check the numbering is in order.

**Response 22:** Thank you for your patience and careful work. We are grateful to the reviewer for pointing out this comment. We have followed your advise to adjusted it. Details are in the MS.

**Comment 23:** Section 4.1 the sentence referring to ArcGIS software needs further explanation. Why was it necessary to vectorise the data? What does the vectorisation have to do with soil erosion prediction?

**Response 23:** Thank you for your comment! According to your comment, we explained it, details are in following paragraph.

Vector data is a data organization way to represent the spatial distribution of geographic entities by using Euclidean geometry in points, lines, surfaces, and their combinations. The vector data model of ArcGIS is a layer. The amount of soil erosion is obtained by the superposition operation of the related layers in the ArcGIS.

**Comment 24:** Section 4.1 sentence starting "The natural and artificial slope catchment

watershed…" -the final statement "such as property" needs clarification.

**Response 24:** Thank you for your comments. We have followed your advise to adjusted it. Details are in following paragraph and MS.

The natural and artificial slope catchment watershed was divided into uniform prediction units on the basis of the extracted graphical units of the artificial and natural slope catchments and according to the differences in aspect, slope, land use, and water conservation measures, such as property.

**Explain:** such as property refers to aspect, slope, land use, and water conservation measures.

**Comment 25:** Section 4.1.1 please provide a citation for the ArcGIS tool。

**Response 25:** Thank you for your comments. We have followed your advice to adjust it. Details are in following paragraph and MS.

The catchment unit of the slope was constructed using the structural plane tools of ArcGIS combined with ridge and valley lines and artificial slope and highway boundaries (Zerihun et al., 2018).

**Reference:**

Zerihun, M., Mohammedyasin, M. S., Sewnet, D., Adem, A. A., & Lakew, M.: Assessment of soil erosion using RUSLE, GIS and remote sensing in NW Ethiopia. Geoderma Regional,12: 83-90, 2018.

**Comment 26:** Section 4.1 general - some of this information feels like it belongs in the methodology rather than results。

**Response 26:** We are grateful to the reviewer for pointing out this comment. According to your comment, we have followed your advice to adjust it. Details are in the MS.

**Comment 27:** The results section of the paper is highly technical, and currently would be more suited to an engineering type journal. Please consider the title of the journal and the likely audience. Try to tell a more logical story of why certain methods have been used, and think about what information is useful to the reader. Some information could possibly go in supplementary material.

**Response 27:** Thank you for your valuable and thoughtful comments! We have made a detailed revision of the manuscript and the adjustment of the structure, and we hope that the new manuscript will satisfy you. At the same time, we also explain the meaning of this article to the reader, details are in following paragraph.

We believe that the four new concepts presented in this manuscript may be of considerable interest to the usual readers of this journal.

**First, in terms of technical methods:** In this study, the revised universal soil loss equation (RUSLE) is used as a prediction model for soil and water loss on slopes, combined with GIS and remote sensing technology. The methods of field survey and runoff observation are used, on the

basis of fully considering the differences between the model parameters of the artificial slopes and the natural slopes of the expressway. The catchment area is considered a prediction unit. The prediction units of the artificial and natural slopes are classified, and the soil and water loss of each slope is predicted in real time, thus reflecting the soil and water loss of expressway slope accurately. This study not only provides technical experience and reference for the prediction of soil erosion but also helps promote the study of water and soil loss in mountain highways in the world.

**Second, in terms of data contribution:** In view of the fact that mountain areas have scattered populations, towns, and farmlands, traffic and economy move backward. The topography is complex, and the climate types vary. In addition, the accumulation of highway soil erosion research in the world cannot satisfy the requirements of model development. To date, no mature model of soil erosion for highway is available, thereby resulting in the loss of soil and water in highways in mountain areas. In addition, the relevant data reserves are weak. This study provides a large amount of data on such variables as mountain precipitation and soil erodibility factors. The work provides reference that allows international counterparts to study the ecological environmental problems in mountain areas, recognize and explore the laws of soil and water loss in mountain highways, and alleviate the scarcity of data in mountain areas to a certain extent.

**Third, in terms of model parameter improvement:** This research aims to characterize soil and water loss in mountain expressways by improving the method of slope element division. In the process of determining the model parameters, the interpolation method is used to obtain rainfall data values. The uneven spatial distribution and heterogeneity problems related to mountain rainfall are solved. The factor of soil and water conservation of artificial slope ($P$) is corrected by the area ratio method. The soil on the side slope of the expressway is different from the arable soil in the general sense, and the side slope type is also varied. Therefore, the soil erodibility factor ($K$) is corrected on the basis of the improvement of the slope unit partition and the field investigation. In addition, the other factors are corrected by the experimental data. This study is not only significant in improve the accuracy of the RUSLE model in predicting soil erosion but also provides an updated understanding and inspiration for international counterparts in related fields.

**Fourth, in terms of results and understanding:** This study fully considers the characteristics of expressways in mountain areas. The catchment area is considered a prediction unit. The method of slope division is improved, and a method of improving the parameters in the model is proposed. Comparison and analysis with actual observation data show that the method of soil and water loss prediction adopted in this paper has less error and higher prediction accuracy than other models and can satisfy prediction requirements. The risk grades of soil and water loss under 20- and one-year rainfall return periods show that the percentage of units with high and extremely high risk of soil erosion is 7.11% and mainly distributed in the K109+500–K110+500 and K133–K139+800 sections. Relevant departments should therefore reinforce the disaster

prevention and mitigation efforts and the corresponding soil and water conservation work in these areas.

**Comment 28:** Section 4.2.1 Please state what the interpolation calculations are

**Response 28:** Thank you for your comments! According to your comments, we explained it, details are in following paragraph and MS.

On the basis of determining the factors affecting precipitation in different temporal and spatial heterogeneity zones, in the interpolation process, the factors affecting precipitation are introduced, and the cokriging method is selected for interpolated count.

**Comment 29:** Table 1 and Table 3 Please make clearer what the section of expressway names actually refer to. I.e., how would I find "K83+500~K84+900" on a map?

**Response 29:** Thank you for your comments! According to your comments, we explained it, details are in following paragraph and MS.

In China, the pile number of the freeway is the meaning of mileage. In general, we can express the position on the road according to the number of the pile. When we enter this section of the highway, we can see the number of piles, but we can't find the corresponding number on the map.

**Comment 30:** General. When referring to equations, do not refer to the authors by name and then cite. Also, check the spelling of author names is consistent and do not use author first names. For example in section 4.2.3, a sentence reads: If the slope is less than 18, then the formula proposed by McCool et al (Mccol et al., 1987) was used. This should be rewritten as something like: If the slope is less than 18, then the formula of McCool et al. (1987) is used

**Response 30:** Thank you for your patience and careful work. We have followed your advice to adjust it. Details are in following paragraph and MS.

The *S* factor was calculated as follows. If the slope was less than 18°, then the formula of McCool et al. (1987) was used. If the slope was greater than 18°, then the formula of Liu et al. (1994) was adopted. See Formula (8).

The natural slope catchment slope was divided into less than 1°, 1°–3°, 3°–5°, and greater than or equal to 5° using the Reclassify tool in ArcGIS. The operation formula adopted the *L* factor algorithm of Moore and Burch (1986), as shown by Formulas (5) and (6).

**Comment 31:** Section 4.2.5 How was NDVI calculated? What data was used?

**Response 31:** Thank you for your comments! According to your comments, we explained it, details are in following paragraph and MS.

The *C*-factor after topography is an important factor that controls soil loss risk. In the RUSLE model, the *C*-factor has been used to reflect the effects of vegetation cover and management practices on the soil erosion rate ((Vander-Knijff et al., 2000; Prasannakumar et al., 2011;

Alkharabsheh et al., 2013). It is defined as the loss ratio of soils from land cropped under specific conditions to the corresponding loss from clean-tilled and continuous fallow (Wischmeier and Smith, 1978).   Due to the variety of land cover patterns with severe spatial and temporal variation, mainly in the watershed scale, data sets from satellite remote sensing were used to assess the C-factor (Vander-Knijff et al., 2000; Li et al., 2010; Chen et al., 2011; Alexakis et al., 2013). The algorithm used in this paper is a method to calculate the C factor proposed by Cai et al. (2000), it is related to vegetation and crop coverage. The formula is shown as (11). Then, the vegetation coverage data were corrected by selecting a sample plot every 2 km along the study area for investigation. The algorithm for calculating f is referred to Tan et al (2005). The formula is shown as (12). Finally, accurate vegetation coverage data were obtained (Figure 9). The *C* factor map of the soil erosion prediction unit in the slope catchment area is shown in Figure 10.

$$C = \begin{cases} 1 & 0 \le f < 0.1\% \\ 0.6508 - 0.3436 \times lg(f) & 0.1\% \le f < 78.3\% \\ 0 & f \ge 78.3\% \end{cases} \quad (11)$$

$$f = \frac{NDVI - NDVI_{min}}{NDVI_{max} - NDVI_{min}} \quad (12)$$

In the formula: *f* is the vegetation coverage, NDVI is the normalized differential vegetation index, $NDVI_{max}$ and $NDVI_{min}$ are the minimum and maximum value of NDVI in the study region, respectively.

[Figure]

Figure 9 Vegetation coverage along Xinhe Expressway

[Figure]

Figure 10 Spatial distribution map of cover and management practice factor

**References:**

Tan, B. X., Li, Z. Y., Wang, Y .H., Yu, P. T., Liu, L. B.: Estimation of Vegetation Coverage and Analysis of Soil Erosion Using Remote Sensing Data for Guishuihe Drainage Basin. Remote sensing technology and application. 20 (2): 215-220, 2005.

Cai, C. F., Ding, S. W., Shi, Z. H., Huang, L., Zhang, G. Y.: Study of Applying USLE and Geographical Information System IDRISI to Predict Soil Erosion in Small Watershed. Journal of Soil and Water Conservation. 14(2): 19-24, 2000.

Vander-Knijff, J.M., Jones, R.J.A., Montanarella, L.: Soil Erosion Risk Assessment in Europe EUR 19044 EN. Office for Official Publications of the European Communities, Luxembourg. 34, 2000.

Wischmeier, W.H., Smith, D. D.: Predicting rainfall erosion losses: a guide to conservation planning. In: USDA, Agriculture Handbook No. 537, Washington, DC, 1978.

Prasannakumar, R., Shiny, N., Geetha, H., Vijith, H.: Spatial prediction of soil erosion risk by remote sensing, GIS and RUSLE approach: a case study of Siruvani river watershed in Attapady valley, Kerala, India. Environmental Earth Science, 965-972, 2011.

Alkharabsheh, M.M., Alexandridis, T.K., Bilasb, G., Misopolinos, N.: Impact of land cover change on soil erosion hazard in northern Jordan using remote sensing and GIS. Four decades of progress in monitoring and modeling of processes in the soil-plant-atmosphere system: applications and challenges. Procedia Environmental Science, 19, 912-921, 2013.

Li, H., Chen, X. L., Kyoung, J. L., Cai, X. B., Myung S.: Assessment of soil erosion and sediment yield in Liao watershed, Jiangxi Province, China, using USLE, GIS, and RS. Journal of Earth

Science 2 (6), 941-953, 2010

Alexakis, D., Diofantos, G., Hadjimitsis, A.: Integrated use of remote sensing, GIS and precipitation data for the assessment of soil erosion rate in the catchment area of "Yialias" in Cyprus. Atmospheric Research, 131, 108-124, 2013.

Chen, T., Niu, R. Q., Li, P. X., Zhang, L. P., Du, B.: Regional soil erosion risk mapping using RUSLE, GIS, and remote sensing: a case study in miyun watershed, north china. Environmental Earth Sciences, 63(3), 533-541, 2011.

**Comment 32:** Section 4.3 How were the field measurements taken in 2014?

**Response 32:** Thank you for your comments! According to your comments, we explained it, details are in following paragraph and MS.

Runoff plots play an important role in soil erosion monitoring. In general, we need to observe the situation of precipitation (precipitation, precipitation intensity), and through artificial sampling method to obtain the volume of runoff and sediment.

**Comment 33:** Section 4.3 Please state why the RMSE is within an acceptable range.

**Response 33:** Thank you for your comments! According to your comments, we explained it, details are in following paragraph and MS.

The root mean square error is a commonly used measure of the difference between the measured values. The value of the root mean square error is often the amount of the model predicted or the estimated estimate. The acceptable range of RMSE in this study is to be judged by actual value and practical experience.

**Comment 34:** End of Section 4.3 The idea that the model may be defective needs further discussion. What might the uncertainties be? Why are the difference between monitoring and analogue spatially variable?

**Response 34:** Thank you for your comments! According to your comments, we explained it, details are in following paragraph and MS.

① End of Section 4.3 The idea that the model may be defective needs further discussion. What might the uncertainties be?

Response: The main problem of RUSLE is not whether it can be applied, but whether the calculation factors can be properly valued. Through the study of this paper, it is found that it is feasible to apply the modified universal soil loss equation to the prediction of soil erosion of the expressway slope. However, because the observation period is relatively short, and the time series of the sample data is not long enough, there is a certain error in the calculation of some factors, which may lead to some research results may not be accurate enough. This needs to be improved and corrected by long-term observation and more abundant field investigation and experiments, so as to further improve the prediction accuracy.

② Why are the difference between monitoring and analogue spatially variable?

Response: This is mainly due to the spatial variability of rainfall.

**Comment 35:** Can you provide any recommendations for how soil and water conservation measures could be rationally adjusted?

**Response 35:** Thank you for your comments! According to your comments, we explained it, details are in following paragraph and MS.

We may consider slowing down the roadbed slope to keep the slope stable, then the ecological slope protection technology can be adopted. Such as the spraying and planting technology of bolt hanging net, it can build a layer of planting matrix which can grow and develop on the weathered rock slope, and can resist the porous and stable structure of the scouring. Finally, it can achieve the purpose of preventing and controlling soil erosion, beautifying the landscape environment of the road area and ensuring the safety of road traffic.

**Comment 36:** Discussion - please comment on how meaningful these methods and results might be in other locations? Is this a site-specific study or does it have wider relevance elsewhere?

**Response 36:** Thank you for your comments! According to your comments, we explained it, details are in following paragraph.

**First, in terms of technical methods:** This study not only provides technical experience and reference for the prediction of soil erosion but also helps promote the study of water and soil loss in mountain highways in the world. **Second, in terms of data contribution:** The work provides reference that allows international counterparts to study the ecological environmental problems in mountain areas, recognize and explore the laws of soil and water loss in mountain highways, and alleviate the scarcity of data in mountain areas to a certain extent. **Third, in terms of model parameter improvement:** This study is not only significant in improve the accuracy of the RUSLE model in predicting soil erosion but also provides an updated understanding and inspiration for international counterparts in related fields.

**Comment 37:** Conclusion. Please write this in paragraphs, not numbered sections. Please define variables again.

**Response 37:** We greatly appreciate your valuable suggestion concerning improvement to this paper. We have followed your advice to adjust it. Details are in following paragraph and MS.

**4 Conclusions**

This study fully considered the differences between the model parameters of the artificial and natural slopes of mountain expressways. Each catchment area was considered a unit. The artificial and natural slope prediction units were then divided, thus producing 422 artificial slope prediction units and 814 natural slope catchment prediction units. The soil and water loss of each slope was predicted in real time, thus making the prediction of soil erosion accurate. The $R$ factor used the

space interpolation method and the $P$ factor of the artificial slope was corrected by the area ratio method in determining the parameters of model prediction. The other factors were corrected by the experimental data. Error analysis of the actual observation data revealed that the overall average absolute error of each monitoring area was 38.65 t·km$^{-2}$·a$^{-1}$, the average relative error was 31.18%, the root mean square error was between 20.95 and 65.64, and the Nash efficiency coefficient was 0.67. The method of soil and water loss prediction adopted in this work generally has less error and higher prediction accuracy than other models and can satisfy prediction requirements. The risk grades of the soil and water loss along the slope of Xinhe Expressway were divided into 20- and 1-year rainfall on the basis of the simulated prediction. The results showed that the percentage of slope areas with high and extremely high risks was 7.11%. These areas were mainly in the K109+500–K110+500 and K133–K139+800 sections. Therefore, relevant departments should strengthen disaster prevention and reduction efforts and corresponding water and soil conservation initiatives in these areas.

---

## Author Comment (AC4) · 15 Mar 2018

The comment was uploaded in the form of a supplement:
https://www.nat-hazards-earth-syst-sci-discuss.net/nhess-2017-406/nhess-2017-406-AC4-supplement.pdf

[Figure]

K83+500

Xinjie

K88

K93

K98

K103

K109

K115

K121

K127

K133

K139+800

Hekou

103°37'0"E  103°45'30"E  103°54'0"E

22°46'0"N

22°37'30"N

22°29'0"N

N
W        E
S

**Legend**

Starting point

Terminal point

Xinhe expressway

Stake number

0    2    4    8
km

DEM

High : 3032

Low : 17

**Fig. 1.** Overview of study region

**After rainfall, soil erosion has occurred on the slope**

**Fig. 2.** Soil erosion produced by rainwash on slope

**Natural slope**

Topography

Ridge line ✚ Valley line

Agrotype

Gray cinnamon soil
Lateritic red soil
Yellow brown earth
Leached cinnamon soil
Gray forest soil

Land use types

Farmland
woodland
Construction land
Barren land

Soil and water conservation measures

Horizontal step
Terraced fields
Contour strip cropping etc.;

**814 prediction units**

**Natural slope catchment area**

**Artificial slope**

The type of slope

Subgrade slope
Cutting slope

slope

The slope is 1:1.5
The slope is 1:1.0

Slope protection measures

Cement frame
+
plant slope

Six arris brick revetment

Plant species

macrophanerophytes
shrub
grass and trees
shrub and herb
arbor-shrub-grass compound land

**422 prediction units**

**Artificial slope catchment area**

**Fig. 3.** Prediction unit division

**Fig. 4.** Division results of prediction units (K133–K139)

**sub-rainfall (mm)**

| | |
|---|---|
| 33.15-34.39 | 37.36-38.03 |
| 34.39-35.48 | 38.03-38.77 |
| 35.48-36.53 | 38.77-39.69 |
| 36.53-37.36 | 39.69- 41.29 |

Xinjie
K88
K93
K98
K103
K109
K115
K121
K127
K133
Hekou

**Legend**

- Stake number
- Xinhe expressway
- Starting point
- Terminal point

0  2.5  5  10
km

103°37'0"E   103°45'30"E   103°54'0"E

22°46'0"N   22°46'0"N
22°37'30"N   22°37'30"N

**Fig. 5.** Interpolation results of secondary rainfall for June 5, 2014

**Fig. 6.** Interpolation results of I30 for June 5, 2014

**Legend**

- ⊙ **Stake number**
- ⚑ **Terminal point**
- ── **Xinhe expressway**

**R ( artificial slope)**
- 138.98-146.92
- 146.92-158.76
- 158.76-172.07
- 172.07-187.69
- 187.69- 205.53

**R (natural slope)**
- 131.49-146.31
- 146.31-157.65
- 157.65-171.11
- 171.11-187.55
- 187.55-204.14

**Fig. 7.** Spatial distribution map of rainfall erosivity factors (K127–K139+800)

Legend

- 🟢 Stake number
- —— Xinhe expressway

**LS (natural slope)**

| | |
|---|---|
| 🟥 | 1.93- 9.84 |
| 🟧 | 9.84-16.11 |
| 🟨 | 16.11-22.48 |
| 🟩 | 22.48- 29.95 |
| 🟩 | 29.95-42.23 |

**LS (Artificial slope)**

| | |
|---|---|
| 🟥 | 0.05 |
| 🟩 | 0.05-6.15 |
| 🟪 | 6.15-6.49 |

**Fig. 8.** Spatial distribution map of topographic factors (K134–K139)

**Fig. 9.** Vegetation coverage along Xinhe Expressway

Fig. 10. Spatial distribution map of cover and management practice factor

**Rainfall (mm)**

| | | | | | | |
|---|---|---|---|---|---|---|
| 🟥 | 108.09-113.25 | 🟨 | 118.59-123.66 | 🟦 | 128.74-133.08 | 🟢 Stake number |
| 🟧 | 113.25-118.59 | 🟩 | 123.66-128.74 | 🟦 | 133.08-138.69 | 〰 Xinhe expressway |

**Fig. 11.** Rainfall interpolation results under 20-year return

**K83+500**
**K88**
**K93**
**K98**
**K103**
**K109**
**K115**
**K121**
**K127**
**K133**
**K139+800**

103°37'0"E    103°45'30"E    103°54'0"E

22°46'0"N                22°46'0"N

22°37'30"N                22°37'30"N

103°37'0"E    103°45'30"E    103°54'0"E

N
W      E
S

0   2.5   5        10
km

| $I_{30}$/(mm·h$^{-1}$) | 46.01-50.42 | 54.68-58.84 | ● Stake number |
|---|---|---|---|
| 42.10-46.01 | 50.42-54.68 | 58.84-62.00 | Xinhe expressway |

**Fig. 12.** Rainfall intensity interpolation results under 20-year return

**Fig. 13.** Risk analysis of soil and water loss under 20-year rainfall conditions

Natural slope
- moderate
- nothing
- slight

Artificial slope
- moderate
- nothing
- slight

Legend
- Stake number
- Xinhe expressway

0   5   10   20
km

**Fig. 14.** Risk analysis of soil and water loss under 1-year rainfall amount

K83+500
K93
K103
K88
K98
K109
K115
K121
K133
K127
K139+800

**Supplement:**

**Response to Decision Letter**

**Dear Dr. Paolo Tarolli and referee,**

We are very pleased to learn from your letter about revision for our manuscript which entitled _"Dangerous degree forecast of soil and water loss on highway slopes in mountainous areas using RUSLE model"._

We greatly appreciate reviewer's thoughtful suggestions concerning improvement to our paper. These comments are all valuable and very helpful for revising and improving our paper, as well as the important guiding significance to our researches.

Thank you for your consideration!

Sincerely yours,

*Corresponding Author: Shi Qi

P.S.

**Response to review's comments for nhess-2017-406-RC2**

**Reviewer's #2 comments:**

**Comment 1:** Highway is booming in China in recent twenty years. Hence how to control soil erosion from the subgrade slope is an important issue of the road safety. From this point, the topic is valuable.

**Response 1:** Thank you for your careful reading of our manuscript. Thank you for the affirmation and support of this topic!

**Comment 2:** The manuscript is also readable although still many language errors exist. Nevertheless, the structure of the paper is poor.

**Response 2:** We are sorry for many language mistakes. Furthermore, we polished the manuscript with a professional assistance in writing, conscientiously. We hope that the new manuscript can meet your request. The revised part of the manuscript was marked in red, details in the manuscript.

**Comment 3:** Instead of a research paper, the manuscript looks like a scientific report. Too many original data are shown in the manuscript but short of in-depth discussions. I strongly suggest the authors rewrite the manuscript.

**Response 3:** Thank you for your valuable and thoughtful comments. According to your helpful advice, we have made a detailed revision and adjustment of the manuscript,

the revised part of the manuscript was marked in red, details in the manuscript.

**Comment 4:** Abstract. Lines 29-30: The word "show" should be replaced with the word "shows". There are many grammar errors throughout the manuscript. I think the paper may be polished by a soil scientist whose native language is English.

**Response 4:** Thank you for your patience and careful work! we have made correction according to your comments. Details are in following paragraph and manuscript.

Many high and steep slopes are formed by special topographic and geomorphic types and mining activities during the construction of mountain expressways. Severe soil erosion may occur under heavy rainfall conditions. Therefore, predicting soil and water loss on highway slopes is important in protecting infrastructure and human life. This work studies Xinhe Expressway, which is in the southern edge of Yunnan–Guizhou Plateau, as the research area. The revised universal soil loss equation is selected as the prediction model of the soil and water loss on the slopes. Moreover, geographic information system, remote sensing technology, field survey, runoff plot observation testing, cluster analysis, and cokriging are adopted. The partition of the prediction units of the soil and water loss on the expressway slope in the mountain area and the spatial distribution model of the linear highway rainfall are studied. In view of the particularity of the expressway slope in the mountain area, the model parameter factor is modified and the risk of soil and water loss along the mountain expressway is simulated and predicted under 20- and 1-year rainfall return periods. The results are as follows. (1) Considering natural watershed as the prediction unit of slope soil erosion can represent the actual situation of the soil and water loss of each slope. The spatial location of the soil erosion unit is realized. (2) An analysis of the actual observation data shows that the overall average absolute error of the monitoring area is 33.24 $t \cdot km^{-2} \cdot a^{-1}$, the overall average relative error is 33.96%, and the overall root mean square error is between 20.95 and 65.64, all of which are within acceptable limits. The Nash efficiency coefficient is 0.67, which shows that the prediction accuracy of the model satisfies the requirements. (3) Under the condition of 1-year rainfall, we find through risk classification that the percentage of prediction units with no risk of erosion is 78%. Results show that soil erosion risk is low and therefore does not affect road traffic safety. Under the 20-year rainfall condition, the percentage of units with high risk and extremely high risk is 7.11% and mainly distributed on the K109+500–K110+500 and K133–K139+800 sections. The prediction results can help adjust the layout of the water and soil conservation measures in these units.

**Comment 5:** Lines 28-30: Is the error of the erosion rate per year? I think the unit may be $t \cdot km^{-2} \cdot a^{-1}$. So are the units of the erosion rates appear through the manuscript.

**Response 5:** Thank you for your comments! We have followed your advice to adjust it. Details are in following paragraph and manuscript.

An analysis of the actual observation data shows that the overall average absolute error of the monitoring area is 33.24 $t \cdot km^{-2} \cdot a^{-1}$, the overall average relative error is 33.96%, and the overall root mean square error is between 20.95 and 65.64, all of which are within acceptable limits.

**Comment 6:** Lines 30-31: The unit of the root mean square error of the soil loss is same to the unit of soil loss. Here the unit is t·km$^{-2}$·a$^{-1}$

**Response 6:** Thank you for your patience and careful work! We have made correction according to your comments. Details are in the manuscript.

**Comment 7:** Lines 33-37: You said "the percentage of prediction units with no risk of erosion is 78% and that with mild soil erosion risk is 15.92%". Do you mean the high and extremely high risk is 1-78%-15.92%=6%? If my deduction is right, I think no large difference exists between this risk and that described in line 37, 7.11%.

**Response 7:** For the first question, we agreed your comment after all the authors discuss.

For the second question, we think this is under different rainstorm frequencies. They are: Under the condition of 1-year rainfall, we find through risk classification that the percentage of prediction units with no risk of erosion is 78%. Results show that soil erosion risk is low and therefore does not affect road traffic safety. Under the 20-year rainfall condition, the percentage of units with high risk and extremely high risk is 7.11% and mainly distributed on the K109+500–K110+500 and K133–K139+800 sections.

**Comment 8:** Introduction. Lines 60-61: "In the next 20-30 years, expressways in China will measure more than 40 thousand km". However I find in the network that "since 2016, the total length of the highway in China is about 131 thousand km". I do not know which length is correct?

**Response 8:** Thank you for your patience and careful work! We have made correction and modification this sentence by consulting the reference, the revised sentence and reference are as follows:

At present, China's highway industry is still in a period of rapid development. By the end of 2014, the total mileages of highway network exceeds 4,400,000 kilometers, and the expressway's mileage is 112,000 kilometers (Zhou et al., 2016). According to statistics, with the development of highway construction in China, slope areas reach 200–300 million m$^2$ each year. In the next 20–30 years, expressways in China will measure more than 40,000 km. For every kilometer of a highway, the corresponding bare slope area formed measures 50,000–70,000 m$^2$. The annual amount of soil erosion is 9,000 g/m$^2$, which causes 450 t of soil loss every year (Chen 2010).

**References:**

Yuan, C., Yu, Q. H., You, Y. H., Guo, L.: Deformation mechanism of an expressway embankment in warm and high ice content permafrost regions. Applied Thermal Engineering 121: 1032-1039, 2017.

Mori, A., Subramanian, S. S., Ishikawa, T., Komatsu, M. A Case Study of a Cut Slope Failure Influenced by Snowmelt and Rainfall. Procedia Engineering, 189: 533-538, 2017.

Zhou, R. G., Zhong, L. D., Zhao, N. L., Fang, J., Chai, H., Jian, Z., Wei, L., Li. B.: The Development and Practice of China Highway Capacity Research. Transportation Research Procedia, 15: 14-25, 2016.

Kateb, H. E., Zhang, H. F., Zhang, P. C., Mosandl, R. Soil erosion and surface runoff on different vegetation covers and slope gradients: A field experiment in Southern Shaanxi Province, China. Catena, 105(5): 1-10, 2013.

**Comment 9:** Line 131: "... the main type of soil erosion in the study area was gully erosion". Could USLE or RUSLE be applied in the gully erosion?

**Response 9:** Thank you for your comments! According to your comments, we explained it. Details are in following paragraph.

Some researchers have adjusted the model parameters of the RUSLE model based on the actual situation in each region, making this model applicable to gully erosion.

**Comment 10:** Lines 62-63: I am lost! Might you kindly let me you the area attacked by soil erosion in this sentence?

**Response 10:** Thank you for your comments! According to your comments, we explained it. Details are in following paragraph.

Chen's (2010) research is based on data collected during the expressway construction period, so as to estimate the area of the bare slope and the amount of soil erosion that can form in the next 20-30 years. It does not mean the amount of soil erosion in one place.

**Comment 11:** Line 68: The word "affect" should be "affects" or "influences".

**Response 11:** Thank you for your careful reading of our manuscript. we have made correction according to the your comments. Details are in following paragraph and MS.

Soil erosion on subgrade side slopes affects not only soil and water loss along the highway but also road operation safety (Gong and Yang 2016; Jiang et al., 2017).

**Comment 12:** Line 78: What's the meaning of the following sentence: "...have explored the process of using the RUSLE model"?

**Response 12:** Thank you for your comments! According to your comments, we adjusted it. Details are in following paragraph and manuscript.

In addition, many scholars have explored the process of using RUSLE models and combined research objects to correct the parameter values in these models

**Comment 13:** Lines 81-82: I am lost.

**Response 13:** Thank you for your comments! According to your comments, we adjusted it. Details are in following paragraph and manuscript.

Tresch et al. (1995) believed that the slope length/slope steepness factor *LS* is one of the main factors for soil erosion modeling within the RUSLE environment. Various steepness factors (*S*) exist for the most used soil erosion modeling environment and significantly influence calculated erosion values.

**Comment 14:** Line 85: 20%-90% should be revised as follows: 20-90%.

**Response 14:** Thank you for your patience and careful work. We are grateful to you for pointing out this comment. We have followed your advise to adjusted it. Details are in following paragraph and MS.

Eighteen plot measurements on transects along slopes ranging from 20–90% in steepness were used in this study to qualitatively assess the most suitable *S* factors for steep subalpine slopes. Results showed that a first selection of an *S* factor was possible for slopes above the critical 25% steepness (Tresch et al., 1995).

**Comment 15:** Lines 125-126: I cannot understand.

**Response 15:** Thank you for your comments! According to your comments, we adjusted it. Details are in following paragraph and manuscript.

However, the accumulation degree of soil and water loss in highways cannot satisfy the requirements of model development (Xu et al., 2009; Bakr et al., 2012). To date, no mature model of soil erosion in highways is available.

**Comment 16:** Line 141: "According to studies at home and abroad". Please remember you are writing a paper for the international journal instead of a scientific report in Chinese!

**Response 16:** Thank you very much for your reminding and advice! We have followed your advice to adjust it. Details are in following paragraph and MS.

According to the literature, the study of soil and water loss in highways has the following problems.

**Comment 17:** Line 158: "In this study," may be added in front of the sentence "A suitable..."

**Response 17:** We greatly appreciate your valuable suggestion concerning improvement to this paper. We have followed your advice to adjust it. Details are in following paragraph and MS.

In this study, a suitable prediction model of soil and water loss is established, the parameters of the model are revised, and the risk of soil and water loss under different rainfall scenarios is simulated and predicted.

**Comment 18:** Lines 161-162: I am lost.

**Response 18:** Thank you for your comments! According to your comments, we adjusted it. Details are in following paragraph and manuscript.

This study scientifically not only predicts the amount of soil erosion caused by highway construction in mountain areas for the rational layout of facilities, which reduces damage to the original topography and effectively prevents and controls new soil erosion, but also provides scientific and technical bases and reference methods.

**Comment 19:** Study area. Line 193: Figure 1 may be merged to Figure 4 in page 11.

**Response 19:** Thank you for your comments! According to your comments, we explained it, details are in following paragraph.

Figure 1 refers to the overview of the study region, figure 4 refers to the division results of prediction units. After all the authors discuss, we believed that these two graphs express independent content, and the content expressed in Figure 4 is not closely related to chapter 2 (Study area).

**Comment 20:** Materials and method. All of the subtitle of the part, including 3.1, 3.1.1-3.3.4, and 3.2, may be erased. Attention please, some of the subtitles in the manuscript are wrong.

**Response 20:** We are sorry for this mistake. We have carefully corrected it throughout the manuscript according to your comment. Details are in the paragraph.

**Comment 21:** Results and analysis. From page 9 to page 22: The part looks like a scientific report instead of a research paper. Except the original experimental data, hardly any in-depth discussion exists.

**Response 21:** Thank you for your instructive suggestions. According to your helpful advice, we have rewritten these parts. Details are in the paragraph. Thank you again!

**Comment 22:** Page 23: The calculated results have not been compared with the results described in other references. Also I do not know why the errors emerge.

**Response 22:** According to your comments, we explained it. Details are in following paragraph.

According to the literature, we found that the related research mainly focused on the estimation of soil erosion under the soil and water conservation measures of different types of slope protection. However, the research content of this paper is to use the RUSLE equation as the prediction model for soil and water loss on slopes with GIS technology as support in view of the characteristics of soil and water loss in mountain expressways. The soil erodibility factor ($K$), slope length factor ($LS$), and soil and water conservation measure factor ($P$) are revised to improve the method of dividing slope units. In determining the predictive parameters of the model, the $R$ factor is obtained by spatial interpolation.

The use of this technique addresses the shortage of rainfall data in mountain areas, the difficulty of representing the rainfall data of an entire expressway with those from a single meteorological station, and the uneven spatial distribution and strong heterogeneity of rainfall in mountain areas. In this study, a suitable prediction model of soil and water loss is established, the parameters of the model are revised, and the risk of soil and water loss under different rainfall scenarios is simulated and predicted.

**Comment 23:** Some of other problems are shown as follows: Where is the line number?

**Response 23:** We are sorry for this mistake. We have carefully corrected it throughout the manuscript according to your comment. Details are in the paragraph.

**Comment 24:** Segment 4.1: The segment is a method, and it may be simplified and removed to Part 3 Materials and method.

**Response 24:** Thank you for your instructive suggestions. According to your comments, we adjusted it. Details are in the manuscript.

**Comment 25:** Segment 4.2.5 in page 21: Why were the landform factors calculated according to the natural slopes (4.2.3) and the artificial slope (4.2.4), but was the vegetation cover calculated only in one situation (4.2.5)?

**Response 25:** Thank you for your comments! According to your comments, we explained it. Details are in following paragraph.

The C-factor after topography is an important factor that controls soil loss risk. In the RUSLE model, the C-factor has been used to reflect the effects of vegetation cover and management practices on the soil erosion rate ((Vander-Knijff et al., 2000; Prasannakumar et al., 2011; Alkharabsheh et al., 2013). It is defined as the loss ratio of soils from land cropped under specific conditions to the corresponding loss from clean-tilled and continuous fallow (Wischmeier and Smith, 1978). Due to the variety of land cover patterns with severe spatial and temporal variation, mainly in the watershed scale, data sets from satellite remote sensing were used to assess the C-factor (Vander-Knijff et al., 2000; Li et al., 2010; Chen et al., 2011; Alexakis et al., 2013). The algorithm used in this paper is a method to calculate the C factor proposed by Cai et al. (2000), it is related to vegetation and crop coverage. The formula is shown as (11). The algorithm for calculating f is referred to Tan et al (2005). The formula is shown as (12):

$$C = \begin{cases} 1 & 0 \le f < 0.1\% \\ 0.6508 - 0.3436 \times lg(f) & 0.1\% \le f < 78.3\% \\ 0 & f \ge 78.3\% \end{cases} \tag{11}$$

$$f = \frac{NDVI - NDVI_{min}}{NDVI_{max} - NDVI_{min}} \tag{12}$$

In the formula: $f$ is the vegetation coverage, NDVI is the normalized differential vegetation index, $NDVI_{max}$ and $NDVI_{min}$ are the minimum and maximum value of NDVI in the study region, respectively.

**References:**

Tan, B. X., Li, Z. Y., Wang, Y .H., Yu, P. T., Liu, L. B.: Estimation of Vegetation Coverage and Analysis of Soil Erosion Using Remote Sensing Data for Guishuihe Drainage Basin. Remote sensing technology and application. 20 (2): 215-220, 2005.

Cai, C. F., Ding, S. W., Shi, Z. H., Huang, L., Zhang, G. Y.: Study of Applying USLE and Geographical Information System IDRISI to Predict Soil Erosion in Small Watershed. Journal of Soil and Water Conservation. 14(2): 19-24, 2000.

Vander-Knijff, J.M., Jones, R.J.A., Montanarella, L.: Soil Erosion Risk Assessment in Europe EUR 19044 EN. Office for Official Publications of the European Communities, Luxembourg. 34, 2000.

Wischmeier, W.H., Smith, D.D.: Predicting rainfall erosion losses: a guide to conservation planning. In: USDA, Agriculture Handbook No. 537, Washington, DC, 1978.

Prasannakumar, R., Shiny, N., Geetha, H., Vijith, H.: Spatial prediction of soil erosion risk by remote sensing, GIS and RUSLE approach: a case study of Siruvani river watershed in Attapady valley, Kerala, India. Environmental Earth Science, 965-972, 2011.

Alkharabsheh, M.M., Alexandridis, T.K., Bilasb, G., Misopolinos, N.: Impact of land cover change on soil erosion hazard in northern Jordan using remote sensing and GIS. Four decades of progress in monitoring and modeling of processes in the soil-plant-atmosphere system: applications and challenges. Procedia Environmental Science, 19, 912-921, 2013.

Li, H., Chen, X. L., Kyoung, J. L., Cai, X. B., Myung S.: Assessment of soil erosion and sediment yield in Liao watershed, Jiangxi Province, China, using USLE, GIS, and RS. Journal of Earth Science 2 (6), 941-953, 2010

Alexakis, D., Diofantos, G., Hadjimitsis, A.: Integrated use of remote sensing, GIS and precipitation data for the assessment of soil erosion rate in the catchment area of "Yialias" in Cyprus. Atmospheric Research, 131, 108-124, 2013.

Chen, T., Niu, R. Q., Li, P. X., Zhang, L. P., Du, B.: Regional soil erosion risk mapping using RUSLE, GIS, and remote sensing: a case study in miyun watershed, north china. Environmental Earth Sciences, 63(3), 533-541, 2011.

**Comment 26:** Line 4 of page 28: "A cement box should be added in the soil a year": Is this the only conservation practice we should adopt?

**Response 26:** Thank you for your comments! According to your comments, we explained it. Details are in following paragraph.

Some other conservation practices: The technology of mortar rubble retaining wall and retaining wall (slope); the technology of honeycomb mesh grass protection; the technology of hydraulic seeding grass protection.

**Comment 27:** References. Page 29: "Liu, X. Y.: Study on the slope stability and its rheological influence in Mountain highway. Central South University, 2013": Is it a paper in Chinese journal or international journal, or an academic dissertation in Chinese?

**Response 27:** Thank you for your comments! According to your comments, we explained it. Details are in following paragraph.

This article is a graduation thesis in Chinese.

**Comment 28:** Page 30: "Bosco, C., De Rigo, D., Dewitte, O., Poesen, J., and..." should be removed forward as the second reference.

**Response 28:** Thank you for your careful reading of our manuscript. We have followed your advice to adjust it. Details are in the manuscript.

**Comment 29:** Page30: "Wang, H. J., Yang, Y., and Wang, W. J.: Prediction of Soil Loss Quantity...": Is it a paper in Chinese journal?

**Response 29:** Thank you for your careful reading of our manuscript. we explained it. Details are in following paragraph.

This article is an academic dissertation in Chinese.

---

## Referee Report (RR1)

The manuscript has been improved. However, it's regretful that most of my advices have not been accepted. I think a major revision may be needed.

(1) Most of my advices have been neglected although the authors have written a very long response, e.g., Responses 12, 13, 15, 19, 21, 22. Especially, I hope to emphasize that a comparison of the results with other references and a discussion in depth may be added in the manuscript, otherwise the manuscript will not look like a scientific paper.

(2) I still do not think the Revised Universal Soil Loss Equation (RUSLE) is suitable for the gully erosion although the authors have given an explanation in Response 9. The gully erosion with mass failures is different to the water erosion on the gentle slope.

(3) In Response 29, now that you have agreed to my advices, why do not you revise the manuscript according to my comment? Besides, is the reference list referred in Response 27 up to the demand of the journal?

(4) There are so many language errors in the revised version shown in the authors' response. It is not imaginable that the last sentences in responses 18 and 26 have been polished by a scientist whose native language is English.

---

## Author Response (AR2)

**Response to Decision Letter**

**Dear Dr. Paolo Tarolli and referee,**

We are very pleased to learn from your letter about revision for our manuscript which entitled _"Dangerous degree forecast of soil and water loss on highway slopes in mountainous areas using RUSLE model"._

We greatly appreciate reviewer's thoughtful suggestions concerning improvement to our paper. These comments are all valuable and very helpful for revising and improving our paper, as well as the important guiding significance to our researches.

Thank you for your consideration!

Sincerely yours,

*Corresponding Author: Shi Qi

P.S.

==========================================================================

**Response to review's comments for nhess-2017-406-RC2**

**Reviewer's #2 comments:**

**Comment 1:** This manuscript has improved considerably since the original submission, however two major issues remain (plus some minor issues outlined below). These are (a) it is not clear what the significance of this work is beyond a comprehensive case study. Please emphasise how this work advances our knowledge, and how others could use it beyond this location. (b) There are still issues with communication. Primarily, the lack of explanation of what the RULSE model is in the introduction RUSLE, and the number of large tables which are difficult to quickly draw any conclusions from The language is significantly improved, but in places remains a little dense.

**Response 1:** (a) This study analysis the characteristics of soil erosion in the process of expressway construction, and improves the following aspects on the basis of previous research: (1) In order to be closer to the actual situation, we divide the highway slope into natural unit and artificial unit, and calculate the amount of soil loss from the slope surface to the pavement by the slope surface catchment unit, which is more in line with the actual situation, and this idea can be popularized; (2) We consider the spatial heterogeneity of linear engineering of expressway, then the rainfall factor is spatially interpolated, it has made up for the defects of rainfall data usually used by rainfall stations in previous studies. (3) We modify the parameters of the artificial slope by means of actual survey, runoff plot observation and other means, and the parameters of the artificial slope are corrected by referring to the form of the project and the materials used.

(b) RUSLE is a set of mathematical equations that estimate average annual soil loss and

sediment yield resulting from interrill and rill erosion (Renard et al., 1997; Foster et al., 1999; Zerihun et al., 2018; Toy et al., 2002). It is derived from the theory of erosion processes, more than 10,000 plot-years of data from natural rainfall plots, and numerous rainfall-simulation plots. RUSLE is an exceptionally well-validated and documented equation. A strength of RUSLE is that it was developed by a group of nationally-recognized scientists and soil conservationists who had considerable experience with erosional processes. (Soil and Water Conservation Society, 1993).

Besides, we have perfected the language of the manuscript, hoping to meet the requirements of reviewers and editor.

**Comment 2:** There are still many statements made without any reference to the literature. For example, line 55 "According to statistics, with the development of highway construction in China, slope areas reach 200-300 million $m^2$ 56 each year". Please check throughout for statements made without supporting reference to the literature.

**Response 2:** Thank you for your patience and careful work! We have made correction according to your comments. Details are in following paragraph and manuscript.

According to statistics, with the development of highway construction in China, slope areas reach 200–300 million $m^2$ each year (Dong and Zeng 2003).

**Reference:**

Dong, H., Zeng, H.: Discussion on the current situation and the future of China highway construction. Technology & Economy in Areas of Communication (TEAC), 5(3):17-18, 2003 (in Chinese).

**Comment 3:** Line 59 "Compared with the soil and water loss on forestlands and farmlands, which on subgrade slopes is special". This sentence is not clear.

**Response 3:** Thank you for your comments. We have followed your advice to revise it. Details are in following paragraph and manuscript.

The soil and water loss of roadbed slope is different from that of soil and water in woodland and farmland.

**Comment 4:** Line 69 "The use of revised universal soil loss equation (RUSLE) models as predictive tools for the quantitative estimation of soil erosion has been maturing for a long time" add references.

**Response 4:** Thank you for your comments. We have followed your advice to add references. Details are in following paragraph and manuscript.

The use of revised universal soil loss equation (RUSLE) models as predictive tools for the quantitative estimation of soil erosion has been maturing for a long time (Panagos et al., 2018; Cunha et al., 2017; Taye et al., 2017; Renard 1997)

References:

Panagos, P., Standardi, G., Borrelli, P., Lugato, E., Montanarella, L., Bosello, F.: Cost of

agricultural productivity loss due to soil erosion in the European union: from direct cost evaluation approaches to the use of macroeconomic models. Land Degradation & Development. 2018.

Cunha, E. R. D., Bacani, V. M., Panachuki, E.: Modeling soil erosion using RUSLE and GIS in a watershed occupied by rural settlement in the Brazilian Cerrado. Natural Hazards, 85, 1-18, 2017.

Taye, G., Vanmaercke, M., Poesen, J., Wesemael, B. V., Tesfaye, S., Teka, D., et al.: Determining RUSLE P-and C-factors for stone bunds and trenches in rangeland and cropland, North Ethiopia. Land Degradation & Development, 29(5), 2017.

Renard, K. G.: Predicting soil erosion by water: A guide to conservation planning with the Revised Universal Soil Loss equation (RUSLE), 1997.

**Comment 5:** Line 72 onwards. You immediately go into detail about the RULSE model parameters/evolution without giving any background to what the RULSE models/equations are. First introduce what the model/equation is before talking about specific parts (e.g., LS).

**Response 5:** We are grateful to the reviewer for pointing out this comment, we have made supplement according to your comments. Details are in following paragraph and manuscript.

RUSLE is a set of mathematical equations that estimate average annual soil loss and sediment yield resulting from interrill and rill erosion (Renard et al., 1997; Foster et al., 1999; Zerihun et al., 2018; Toy et al., 2002). It is derived from the theory of erosion processes, more than 10,000 plot-years of data from natural rainfall plots, and numerous rainfall-simulation plots. RUSLE is an exceptionally well-validated and documented equation. A strength of RUSLE is that it was developed by a group of nationally-recognized scientists and soil conservationists who had considerable experience with erosional processes. (Soil and Water Conservation Society, 1993).

The use of revised universal soil loss equation (RUSLE) models as predictive tools for the quantitative estimation of soil erosion has been maturing for a long time...

**References:**

Renard, K. G., Foster, G. R., Weesies, G. A., McCool, D. K., Yoder, D. C.: Predicting soil erosion by water-a guide to conservation planning with the Revised Universal Soil Loss Equation (RUSLE). United States Department of Agriculture, Agricultural Research Service (USDA-ARS) Handbook No.703. United States Government Printing Office: Washington, DC. 1997.

Foster, G. R., Weesies, G. A., McCool, D. K., Joder, D. C., Renard, K. G.: Revised Universal Soil Loss Equation User's Manual. Gov. Print. Office, Washington D.C. (48p), 1999.

Zerihun, M., Mohammedyasin, M. S., Sewnet, D., Adem, A. A., Lakew, M.: Assessment of soil erosion using RUSLE, GIS and remote sensing in NW Ethiopia. Geoderma Regional, 12, 83-90, 2018.

Toy, T. J., Foster, G. R., Renard, K. G.: Soil Erosion: Processes, Prediction, Measurement, and Control. 2002

Soil and Water Conservation Society. RUSLE user's guide. Soil and Water Cons. Soc. Ankeny, IA. 164pp, 1993.

**Comment 6:** Line 107 use gender neutral pronouns (i.e., use "they", not "he").

**Response 6:** Thank you for your careful work! We have carefully corrected these mistakes according to your comment. Details are in the following paragraph and manuscript.

Zhang (2016) investigated the spatio-temporal distribution of soil erosion in ring expressway before and after construction process, they used land use/cover map of Ningbo City in 2010, topographic map, map of North Ring expressway and field survey data was collected to derive digital elevation model (DEM).

**Comment 7:** Line 177 state what the R factor is. Is this rainfall and runoff? If so, these should have separate parameter notations.

**Response 7:** Thank you for your comments! We have followed your advice to explain and revise it. Details are in the following paragraph and manuscript.

The $R$ factor in this sentence refers to the rainfall.

In order not to cause ambiguity, we revised the sentence as follows:

In determining the predictive parameters of the model, the rainfall is obtained by spatial interpolation.

**Comment 8:** Line 198 you state the difference in precipitation but it is not clear what the values correspond to in terms of difference.

**Response 8:** Thank you for your comments! We have followed your advice to revise it. Details are in the following paragraph and manuscript.

**Revised:**

Between May and the middle of October, the area experiences wet season characterized by abundant rainfall, concentrated precipitation, and increased rain at night, the variation on precipitation is from 400 to 2000mm, and most of the regions are between 800 to 1800mm (Fei et al., 2017; Zhang et al., 2017).

**Comment 9:** Line 203 "The terminal is across the river from the old street of Vietnam". This is unclear.

**Response 9:** Thank you very much! We discussed it on the basis of your comments, we agreed to delete the sentence, but this will not affect the completeness and accuracy of this part. Details are in the manuscript.

**Comment 10:** Line 206. You state here that most of the slope is below 30 degrees, yet in the introduction, you state that most highway slopes are above 30 degrees. This makes the justification for this case study somewhat weak if it is not representative of the slopes of other

road areas. Please clarify.

**Response 10:** Thank you for your comments! We have followed your advice to explain and revise it. Details are in the following paragraph and manuscript.

**Study Area:**

"The slopes on both sides rise and fall, and most of the valleys constitute "V"- and "U"-shaped sections. The natural slopes on both sides are mostly below 30°". The marked part means that the slope of mountain.

**Introduction:**

"Traditional soil and water conservation research focuses on slopes with 20% grade or below, but the highway subgrade slope of steep slopes is generally greater than 30% (Zhou 2010)." The mark part refers to subgrade slope.

**Revised:**

Traditional soil and water conservation research focus on slopes with 20% grade or below, but the roadbed slope of the highway is generally greater than 30% (Zhou 2010). The marked part means that article slope.

In order to distinguish these two types of slopes, the schematic diagram is as follows:

[Figure]

**Comment 11:** Figure 1 part C give units in the legend (I assume this is metres above sea level).

**Response 11:** Thank you for your comments! We have followed your advice to revise it.

Details are in the following paragraph and manuscript.

[Figure]

**Comment 12:** Figure 2 is mentioned in text before figure 1. Re-order the figures and include the new figure 1 (formerly figure 2) in the introduction.

**Response 12:** Thank you for your careful work! We have carefully corrected these mistakes according to your comment. Details are in the following paragraph and manuscript.

[Figure]

**Figure 1.** Soil erosion produced by rainwash on slope

[Figure]

**Figure 2.** Overview of study region

**Comment 13:** Line 239 state what the quartering method is and give a reference for this.

**Response 13:** Thank you for your comments! We have followed your advice to explain and revise it. Details are in the following paragraph and manuscript.

**Method of coning and quartering:** It means that each pile is made into a uniform conical, and compressed into a cone and then divided into four equal parts by a cross frame. The schematic diagram is as follows:

[Figure]

Divide the sample into quarters.     Take "**2**", "**4**" parts of the sample     Agian, divide the sample into quarters.     Take "**2'**", "**4'**" parts of the sample

Reference:

Oyekunle, J. A. O., Ogunfowokan, A. O., Torto, N., Akanni, M. S.: Determination of organochlorine pesticides in the agricultural soil of Oke-Osun farm settlement, Osogbo, Nigeria. Environmental Monitoring & Assessment, 177(1-4), 51, 2011.

**Comment 14:** Line 254 this description of the RULSE model is good, and should be included much earlier on in the introduction, even if this results in some repetition.

**Response 14:** We greatly appreciate your valuable suggestion concerning improvement to this paper. We have followed your advice to adjust it. Details are in following paragraph and manuscript.

The contents added in the introduction are as follows:

RUSLE is a set of mathematical equations that estimate average annual soil loss and sediment yield resulting from interrill and rill erosion (Renard et al., 1997; Foster et al., 1999; Zerihun et al., 2018; Toy et al., 2002). It is derived from the theory of erosion processes, more than 10,000 plot-years of data from natural rainfall plots, and numerous rainfall-simulation plots. RUSLE is an exceptionally well-validated and documented equation. A strength of RUSLE is that it was developed by a group of nationally-recognized scientists and soil conservationists who had considerable experience with erosional processes. (Soil and Water Conservation Society, 1993).

**Comment 15:** Line 261 please give more detail on what the units MJ·mm/(hm²·h) means and ensure this is in SI notation.

**Response 15:** Thank you for your comments! We have followed your advice to explain it. Details are in the following paragraph and manuscript.

$$A = R \cdot K \cdot L \cdot S \cdot C \cdot P, \qquad (1)$$

where $A$ is the average soil loss per unit area by erosion (t/hm²), R is the rainfall erosivity factor (MJ·mm / (hm²·h)), $K$ is the soil erodibility factor (t·hm²·h / (hm²·MJ·mm)), $L$ is the slope length

factor, S is the steepness factor, *C* is the cover and management practice factor, and *P* is the conservation support practice factor. The values of *L*, *S*, *C*, and *P* are dimensionless.

The *R* factor is an expression of the erosivity of rainfall and runoff at a particular location. Analyses of data indicated that when factors other than rainfall are held constant, soil loss is directly proportional to a rainfall factor composed of total storm kinetic energy (E) times the maximum 30-min intensity ($I_{30}$) (Wischmeier and Smith, 1958).

By definition, the value of EI for a given rainstorm equals the product, total storm energy (E) times the maximum 30-min intensity ($I_{30}$), where E is in is in hundreds of foot-fons per acre and $I_{30}$ is in inches per hour (in/h). El is an abbreviation for energy-times-intensity, and the term should not be considered simply an energy parameter. The data show that rainfall energy, itself, is not a good indicator of erosive potential. The storm energy indicates the volume of rainfall and runoff, but a long, slow rain may have the same E value as a shorter rain at much higher intensity. Raindrop erosion increases with intensity. The $I_{30}$ component indicates the prolonged-peak rates of detachment and runoff. The product term, EI, is a statistical interaction term that reflects how total energy and peak intensity are combined in each particular storm. Technically, it indicates how particle detachment is combined with transport capacity.

The energy of a rainstorm is a function of the amount of rain and of all the storm's component intensities. Median raindrop size increases with rain intensity (Wischmeier and Smith 1958), and terminal velocities of free-falling waterdrops increase with increased drop-size (Gunn and Kinzer 1949). Since the energy of a given mass in motion is proportional to velocity-squared, rainfall energy is directly related to rain intensity.

**Comment 16:** Line 272 This sentence is very long and unclear. Please break up into shorter statements.

**Response 16:** We greatly appreciate your valuable suggestion concerning improvement to this paper. We have followed your advice to revised it. Details are in following paragraph and manuscript.

Line 272: The natural and artificial slope catchment watershed was divided into uniform prediction units on the basis of the extracted graphical units of the artificial and natural slope catchments and according to the differences in aspect, slope, land use, and water conservation measures, such as property.

On the basis of the extracted graphical units of the artificial and natural slope, the natural and artificial slope was divided into a uniform prediction unit according to the aspect, slope, land use,

water conservation measures.

**Comment 17:** Figure 3. This figure caption needs much more detail to explain how this works. Figure captions should be standalone so that the reader could fairly easily look at the figure and understand it without having to read the body text.

**Response 17:** We greatly appreciate your valuable suggestion concerning improvement to this paper. We have followed your advice to adjust it. Details are in following paragraph and manuscript.

Figure 3 contains the contents of 2.3 and 3.1, which is a summary of the two parts. Therefore, in the original manuscript, Figure 3 as part of the content of 2.3 is one-sided. In order not to cause misunderstanding between readers and peer reviewers, at the same time, it is also conducive to the accurate interpretation of the manuscript, after discussion, all authors believe that Figure 3 should be deleted.

**Comment 18:** Line 298 incorrect Harvard reference (McCool).

**Response 18:** We are grateful to the reviewer for pointing out this mistake. We have followed your advice to revise it. Details are in following paragraph and manuscript.

Mccool (1987) stated that slope length varies within a 10 m range and has only a small effect on results.

**Comment 19:** Figure 4 does not look like it shows 1236 units-are we looking at a subset here? If so, state this in the figure caption.

**Response 19:** We are in complete agreement with your comment. The Figure 4 expresses a subset. So, we have followed your advise to revise it. Details are in following paragraph and manuscript.

[Figure]

**Figure 4.** Division results of prediction units (A subset-6.8 km)

**Comment 20:** Line 324 state method of interpolation.

**Response 20:** Thank you for your comments! We have followed your advice to explain and revised it. Details are in the following paragraph and manuscript.

The method here refers to the co-kriging.

The $P$ and $I_{30}$ values along the highway were obtained by the method of co-kriging calculations.

In the process of co-kriging calculations, combined with environmental characteristics, a number of factors that affect precipitation are introduced.

**Comment 21:** Line 329 (and more generally throughout) you often state what method you used with no justification as to why this is an appropriate method. Please give further information why mean standard error is an appropriate criteria.

**Response 21:** Thank you for your comments! We have followed your advice to explain it. Details are in the following paragraph.

The mean standard error reflects the variation of the sample mean to the population means, thus reflecting the size of sampling error, it is an indicator of precision of measurement results.

This is the basic knowledge of statistics. Such a method is often used for analysis of similar studies (Yang et al., 2008; Liu 2014)

We have done several error analysis, the result shows that there is little difference between

the results of the analysis. Considering the structure of the whole article without lengthy text, so it is expressed only by mean standard error.

**Comment 22:** Figure 5 and 6 I suggest combining these into a two part figure.

**Response 22:** Thank you for your comments! We have followed your advice to adjust it. Details are in following paragraph and manuscript.

[Figure]

**Figure 5(a).** Interpolation results of secondary rainfall for June 5, 2014;

**Figure 5(b).** Interpolation results of $I_{30}$ for June 5, 2014

**Comment 23:** Figure 7 it is very difficult to see any variation in R due to the units being very small. I suggest showing a sub figure with zoomed sections.

**Response 23:** We greatly appreciate your valuable suggestion concerning improvement to this paper. We have followed your advice to adjust it. We added a subgraph, hoping to reflect the variation of R value as far as possible. Details are in following paragraph and manuscript.

[Figure]

**Figure 7.** Spatial distribution map of rainfall erosivity factors (K127–K139+800)

[Figure]

**Figure 7(a).** The subgraph of Figure 7 with zoomed sections.

**Comment 24:** Line 352 incorrect Harvard reference

**Response 24:** Thank you for your comments! We have followed your advice to adjust it. Details are in following paragraph and manuscript.

The calculation method of the *K* value was adopted by Formula 4 to obtain the soil erodibility factor values of each slope (Sharply and Williams 1990 ), as shown in Tables 3 and 4.

**Comment 25:** Table 3 and 4. Are these necessary for the main body of your text? I suggest these should be supplementary material/appendices.

**Response 25:** We are in complete agreement with your comment. We take the Table 3 and 4 as the supplementary material/appendices.

**Comment 26:** Equation 6 some issue with font/layout. Not possible to understand currently.

**Response 26:** Thank you for your comments! We have followed your advice to revised it. Details are in following paragraph and manuscript.

$$\lambda = flowacc \cdot cellsize \qquad (6)$$

**Comment 27:** Section 3.2.5 state what remote sensing imagery and method was used.

**Response 27:** Thank you for your comments! We have followed your advice to explain and revise it. Details are in the following paragraph and manuscript.

The remote sensing images used in this study were derived from 8m hyperspectral images produced by GF-1 satellite (http://www.rscloudmart.com/).

**Method:**

Vegetation coverage and management factor (*C*): *C* is defined as the ratio of soil loss from land cropped under specific conditions to the corresponding loss from clean-tilled, continuous fallow. Taking full advantage of the Normalized Difference Vegetation Index (NDVI) data, *C* is calculated according to the equation of Gutman and Ignatov (1998).

$$C = 1 - \frac{NDVI - NDVI_{min}}{NDVI_{max} - NDVI_{min}}$$

**Comment 28:** Table 9 convert to a figure. It is hard to quickly look at a table and spot trends/differences.

**Response 28:** Thank you for your comments! We have followed your advice to revise it. Details are in the following paragraph and manuscript.

**K83+550**

[Figure]

The time of the second rainfall

**Figure 9.** Comparison of model prediction and monitoring results ( K83+550)

**K93+550**

[Figure]

The time of the second rainfall

**Figure 10.** Comparison of model prediction and monitoring results ( K93+550)

[Figure]

**Figure 11.** Comparison of model prediction and monitoring results ( K133+550)

**Comment 29:** Tables 10-12. There are too many tables in this paper. Consider moving to supplementary material, summarising or converting to a figure.

**Response 29:** We are in complete agreement with your comment. So, we have followed your advise to adjust it. We take the Table 10-12 as the supplementary material/appendices. Details are in the manuscript.

**Comment 30:** This work presents an interesting case study, which clearly required much research to be undertaken. It is still not clear how transferrable this would be to other settings, or what the value of the work is beyond this case study. In the discussion, try to emphasise the bigger picture in terms of how others can now build upon your research.

**Response 30:** We greatly appreciate your valuable suggestion concerning improvement to this paper. We have followed your advice to revised it. We have added some content (**Discussion**), hoping to meet the requirements of the reviewer. Details are in following paragraph and manuscript.

[revised manuscript text omitted]

**Response to Decision Letter**

**Dear Dr. Paolo Tarolli and referee,**

We are very pleased to learn from your letter about revision for our manuscript which entitled _"Dangerous degree forecast of soil and water loss on highway slopes in mountainous areas using RUSLE model"._

We greatly appreciate reviewer's thoughtful suggestions concerning improvement to our paper. These comments are all valuable and very helpful for revising and improving our paper, as well as the important guiding significance to our researches.

Thank you for your consideration!

Sincerely yours,

*Corresponding Author: Shi Qi

P.S.

=======================================================================

**Response to review's comments for nhess-2017-406-RC1**

**Reviewer's #1 comments:e**

**Comment 1:** The manuscript has been improved. However, it's regretful that most of my advices have not been accepted. I think a major revision may be needed.

**Response 1:** First of all, all the authors thank you for your affirmation of the last revision, though it still needs some improvement. We hope that this revision will meet your requirements!

**Comment 2:** Most of my advices have been neglected although the authors have written a very long response, e.g., Responses 12, 13, 15, 19, 21, 22.

**Response 2:** Thank you for your patience and careful work! We will further improve and modify these response according to your comments. Details are in following paragraph and manuscript.

=======================================================================

**Comment 12:** Line 78: What's the meaning of the following sentence: "...have explored the process of using the RUSLE model"?

**Response 12*:** Thank you for your comments! We have revised this sentence again, hoping to express it more clearly. Details are in following paragraph and manuscript.

Moreover, many scholars have made many useful explorations in modifying the model parameter values and improving the simulation accuracy.

**Comment 13:** Lines 81-82: I am lost.

**Response 13\*:** We have revised this sentence again, hoping to express it more clearly. Details are in following paragraph and manuscript.

Tresch S et al (1995) thought that the topographical factor LS are one of the main factors for soil erosion modelling approaches within the RUSLE environment. There exists a large variety of different S-factors, for the most used soil erosion modelling environment, which have highly significant influences on the calculated erosion values.

**Revised:**

Tresch et al. (1995) believed that the slope length ($L$) or slope steepness factor ($S$) is one of the main factor for soil erosion prediction, and significantly influence the erosion values calculated by the RUSLE.

**Comment 15:** Lines 125-126: I cannot understand.

**Response 15\*:** Thank you for your comments! We have revised this sentence again, hoping to express it more clearly. Details are in following paragraph and manuscript.

However, the accumulation degree of soil and water loss in highways cannot satisfy the requirements of model development (Xu et al., 2009; Bakr et al., 2012). To date, no mature model of soil erosion in highways is available.

**Revised:**

However, the research progress on soil and water loss of highway hardly meet the requirements of the practical work. (Xu et al., 2009; Bakr et al., 2012). So far, we still need to do a lot of work on the prediction of soil erosion in highway slopes.

**Comment 19:** Study area. Line 193: Figure 1 may be merged to Figure 4 in page 11.

**Response 19:** Thank you for your comments! According to your comments, we explained it, details are in following paragraph.

Figure 1 refers to the overview of the study region, figure 4 refers to the division results of prediction units. After all the authors discuss, we believed that these two graphs express independent content, and the content expressed in Figure 4 is not closely related to chapter 2 (Study area). So, our final decision is not to merge Figure 1 to Figure 4.

**Comment 20:** Results and analysis. From page 9 to page 22: The part looks like a scientific report instead of a research paper. Except the original experimental data, hardly any in-depth discussion exists. Page 23: The calculated results have not been compared with the results described in other references. Also I do not know why the errors emerge. Especially, I hope to emphasize that a comparison of the results with other references and a discussion in depth may be added in the manuscript, otherwise the manuscript will not look like a scientific paper.

**Response 20:** We greatly appreciate your valuable suggestion concerning improvement to this paper. We have followed your advice to revised it. We have added some content (**Discussion**), hoping to meet the requirements of the reviewer. Details are in following paragraph and

manuscript.

**Discussion**

[revised manuscript text omitted]

=======================================================================

**Comment 4:** I still do not think the Revised Universal Soil Loss Equation (RUSLE) is suitable for the gully erosion although the authors have given an explanation in Response 9. The gully erosion with mass failures is different to the water erosion on the gentle slope.

**Response 4:** Thank you for your comment! We agree with your comment after discussing and consulting the references.

After careful checking, we think that the relevant parts of the manuscript is independent of the RUSLE model. Therefore, it is incompatible with the core idea of this paragraph. So, we decided to delete this sentence for the sake of the accuracy of the manuscript.

**Comment 5:** In Response 29, now that you have agreed to my advice, why do not you revise the manuscript according to my comment? Besides, is the reference list referred in Response 27 up to the demand of the journal?

**Response 5:** Thank you for your comments! We have revised it again, details are in following paragraph and manuscript.

========================================================================

**Comment 29:** Page30: "Wang, H. J., Yang, Y., and Wang, W. J.: Prediction of Soil Loss Quantity...": Is it a paper in Chinese journal?

**Response 29\*:** We have revised this sentence again, hoping to express it more clearly. Details are in following paragraph and manuscript.

Wang, H. J., Yang, Y., and Wang, W. J.: Prediction of Soil Loss Quantity on Side Slope of Freeway Construction: Amendments to Main Parameters of USLE. Journal of Wuhan University of Technology (Transporatation Science & Engineering), 29(1):12-15, 2005 (in Chinese).

========================================================================

**Comment 6:** Besides, is the reference list referred in Response 27 up to the demand of the journal?

**Response 6:** We have revised this reference, details are in following paragraph and manuscript.

Liu, X. Y.: Study on the slope stability and its rheological influence in Mountain highway. Central South University, 2013 (in Chinese).

**Comment 7:** There are so many language errors in the revised version shown in the authors' response. It is not imaginable that the last sentences in responses 18 and 26 have been polished by a scientist whose native language is English.

**Response 7:** Thank you for your instructive suggestions. According to your helpful advice, we have rewritten these responses. Details are in the paragraph and manuscript. Thank you again!

========================================================================

**Comment 18:** Lines 161-162: I am lost.

**Response 18:** Thank you for your comments! We have revised this sentence again, hoping to

express it more clearly. Details are in following paragraph and manuscript.

This study scientifically not only predicts the amount of soil erosion caused by highway construction in mountain areas for the rational layout of facilities, which reduces damage to the original topography and effectively prevents and controls new soil erosion, but also provides scientific and technical bases and reference methods.

**Revised:**

This study not only scientifically predicts the amount of soil erosion caused by the highway construction in mountain areas, but also provides a scientific basis for the prevention and control of soil erosion, and the rational allocation of prevention and control measures.

**Comment 26:** Line 4 of page 28: "A cement box should be added in the soil a year": Is this the only conservation practice we should adopt?

**Response 26:** Thank you for your comments! According to your comments, we explained it. Details are in following paragraph.

Some other conservation practices: The technology of mortar rubble retaining wall and retaining wall (slope); the technology of honeycomb mesh grass protection; the technology of hydraulic seeding grass protection.

**Revised:**

Some other conservation practices: Technology of masonry retaining wall; Technology of honeycomb grid revetment protection; Technology of hydraulic seeding grass protection.

==========================================================================

---

## Author Response (AR3)

**Response to Decision Letter**

**Dear Dr. Paolo Tarolli and referee,**

We are very pleased to learn from your letter about revision for our manuscript which entitled _"Dangerous degree forecast of soil and water loss on highway slopes in mountainous areas using Revised Universal Soil Loss Equation model"._

We greatly appreciate reviewer's thoughtful suggestions concerning improvement to our paper. These comments are all valuable and very helpful for revising and improving our paper, as well as the important guiding significance to our researches.

Thank you for your consideration!

Sincerely yours,

*Corresponding Author: Shi Qi

P.S.

========================================================================

**Response to review's comments for nhess-2017-406-Report 1**

**Reviewer's #1 comments**

**Comment 1:** In page 23 of the Response to Decision Letter, you said "After careful checking, we think that the relevant parts of the manuscript is (are) independent of the RUSLE model. Therefore, it is incompatible with the core idea of this paragraph. So, we decided to delete this sentence for the sake of the accuracy of the manuscript." Nevertheless, the title of your paper is as follows: Dangerous degree forecast of ⋯. USING RUSLE MODEL.

**Response 1:** Thank you for your patience and careful work! We have followed your advice to explain it. Details are in the following paragraph.

In this paper, the RUSLE model is used to predict the soil erosion of the expressway slope in mountainous area. But in the original manuscript, the study of Yang (2001) mentioned in the introduction was about gully erosion, so we decided to delete the following contents:

Yang (2001) investigated the behavior of soil erosion on the slope of a railway embankment

during construction by comparing artificial and natural rainfalls on the special Qinhuangdao-Shenyang line of passenger trains. The results showed that the main type of soil erosion in the study area was gully erosion, which caused more soil erosion than surface erosion did.

**Comment 2:** In depth discussions are still needed. Yes, you have added two paragraphs in the section Discussion. However, the second paragraph, lines 567-576 in page 27, is not a discussion. I think to put the paragraph in the end of the Section Introduction is the best.

**Response 2:** We greatly appreciate your valuable suggestion concerning improvement to this paper. We have followed your advice to adjust it. Details are in following paragraph and manuscript.

In this study, we analyse the following characteristics of soil erosion to improve certain aspects expressway construction based on previous research: (1) We divide the highway slope into natural and artificial units and calculate the amount of soil loss from the slope surface to the pavement based on the slope surface catchment unit. Considering that this approach is in line with the actual situation, the findings can thus can be popularised. (2) Previous studies have shown that the spatial interpolation method of precipitation has been stagnant in the study of the spatial and temporal distribution of precipitation in mountain areas (Liu and Zhang, 2006). At the same time, the problem involves two aspects. From the timescale perspective, the characteristics of rainfall distribution and the influencing factors are not fully considered. From the spatial scale perspective, the spatial heterogeneity of the region is ignored. Many studies have limited the factors affecting precipitation to altitude factors, thereby leading to low interpolation accuracy (Zhao et al., 2011; Liu et al., 2010). Thus, we consider the spatial heterogeneity of linear engineering of the expressway. Then, the rainfall factor is spatially interpolated to compensate the following limitations: shortage of rainfall data of mountain areas; difficulty of representing the rainfall data of an entire expressway by using those data from a single meteorological station; and uneven spatial distribution and strong heterogeneity of rainfall in mountain areas (Li et al., 2017). (3) We modify the parameters of the artificial slope by actual survey, runoff plot observation and other methods, and the parameters of the artificial slope are corrected by referring to the form of the project and the utilised materials. In this study, we not only scientifically predict the amount of

soil erosion caused by highway construction in mountain areas but also provide a scientific basis for the prevention and control of soil erosion and the rational allocation of prevention and control measures. Moreover, the safe operation of highways and the virtuous cycle of the ecological environment should be ensured to promote the sustainable development of the local economy.

Reference:

Li, Y., Qi S., Cheng, B. H., Ma, J. M., Ma, C., Qiu, Y. D., Chen, Q. Y.: A Study on Factors of Space-time Distributions of Precipitation in Ailao Mountain Area and Comparison of Interpolation Methods. EARTH AND ENVIRONMENT, 45(6): 600-610 (in Chinese)

Liu, J. T., Zhang, J. B.: Interpolation analysis of the spatial distribution of precipitation in mountain area. Journal of Irrigation & Drainage, 25:34-38, 2006 (in Chinese)

Zhao, C. C., Ding, Y. J., Ye, B. S., Zhao, Q. D.: Spatial distribution of precipitation in Tianshan Mountains and its estimation. Advance in water science, 22:315-322, 2011 (in Chinese)

Liu, Z. Y., Zhang, X., Fang, R. H. Analysis of spatial interpolation methods to precipitation in Yulin based on DEM. Journal of Northwest A&F University (Nat. Sci. Ed.), 38:227-234, 2010 (in Chinese)

**Comment 3:** There are so many language errors in the latest version of the manuscript. For example, in lines 81-83, the word factor may be revised as factors, and the word influence may be revised as influences. In line 567, the word analysis may be revised as analyzes.

**Response 3:** We are grateful to the reviewer for pointing out these mistake. We have followed your advice to revise it. Details are in following paragraph and manuscript.

Line 81-83:

Tresch et al. (1995), in a study of Switzerland, argued that the slope length ($L$) or slope steepness factor ($S$) is one of the main factors of soil erosion prediction, and these parameters significantly influenced the erosion values calculated by RUSLE.

Line 567:

In this study, we analyse the following characteristics of soil erosion to improve certain aspects expressway construction based on previous research.

**Comment 4:** I am sorry in the latest manuscript I could not find what you said in the last page of the Response to Decision Letter, namely Technology of masonry retaining wall; (maybe comma is better here) Technology of honeycomb grid revetment protection...

**Response 4:** Thank you for your comments! We have followed your advice to revised it. Details are in following paragraph and manuscript.

We may consider slowing down the roadbed slope to keep the slope stable, and ecological slope protection technologies can be adopted. For example, the spraying and planting technology of bolt hanging net can be used to build a layer of planting matrix that can grow and develop on the weathered rock slope, as they can resist the porous and stable structure of the scouring. Technologies of masonry retaining wall and honeycomb grid revetment protection can also be used. Various technologies can be adopted for the purpose of preventing and controlling soil erosion, and they can beautify the landscape environment of the road area whilst ensuring road traffic safety.

**Response to Decision Letter**

**Dear Dr. Paolo Tarolli and referee,**

We are very pleased to learn from your letter about revision for our manuscript which entitled *"Dangerous degree forecast of soil and water loss on highway slopes in mountainous areas using RUSLE model".*

We greatly appreciate reviewer's thoughtful suggestions concerning improvement to our paper. These comments are all valuable and very helpful for revising and improving our paper, as well as the important guiding significance to our researches.

Thank you for your consideration!

Sincerely yours,

*Corresponding Author: Shi Qi

P.S.

========================================================================

**Response to review's comments for nhess-2017-406-Report 2**

**Reviewer's #2 comments:**

**Comment 1:** General: This article needs a final read by a proofreader for the English language. There are numerous small errors with use of singulars and plurals and general phrasing.

**Response 1:** Thank you for your patience and careful work! We have revised the manuscript again, we hope to meet your requirements. Details are in the manuscript.

**Comment 2:** Line 19: Geographic information system correct to 'systems' (plural), and the same for field survey (should be plural).

**Response 2:** Thank you for your comments. We have followed your advice to revise it. Details are in following paragraph and manuscript.

Moreover, geographic information systems, remote sensing technology, field surveys, runoff plot observation testing, cluster analysis and co-kriging calculations are adopted.

**Comment 3:** Line 29. Unclear what the unit 'a' is. I am guessing that t is tonnes but this is also not immediately clear.

**Response 3:** Thank you for your comments! We have followed your advice to explain it. Details are in the following paragraph.

'a' means 'anniversary'.

**Comment 4:** Line 30 should RMSE have the same units as the absolute error?

**Response 4:** Thank you for your comments! We have followed your advice to explain it. Details are in the following paragraph.

Absolute error is the absolute magnitude of the deviation of the measured value from the true value, so its unit is the same as that of the measured value.

The root mean square error is the deviation between the observed value and its true value, or the observed value and its simulated value. It is dimensionless.

**Comment 5:** Line 36. It will be unclear to the reader what these sections are at this point. Consider removing this information or making it clearer to the reader. E.g., 'specific sections located near to the town of⋯'.

**Response 5:** Thank you for your comments! We have followed your advice to delete it. Details are in following paragraph and manuscript.

Under the 20-year rainfall condition, the percentage of units with high risk and extremely high risk is 7.11%. The prediction results can help adjust the design of water and soil conservation measures for these units.

**Comment 6:** Paragraph starting line 43 has no references and should be supported by citation to the literature.

**Response 5:** Thank you for your comments! We have followed your advice to revise it. Details are in following paragraph and manuscript.

China has gradually accelerated the construction of highways in recent years and subsequently improved transportation networks and driven rapid economic development (Jia et al., 2005)

Reference:

Jia, Y. H., Dai, D. C., Liu, Y.: Performance Analyse and Evaluation of Freeway in China. JOURNAL OF BEIJING JIAOTONG UNIVERSITY, 29(6):1-5, 2005 (in Chinese)

**Comment 7:** Line 55 sentence starting 'According to statistics, with the development of highway..' is somewhat unclear and needs rephrasing. Do you mean something like 'on an annual basis, highways are constructed through 200-300 km2 per year'. Square kilometres rather than square meters also seem more appropriate for numbers here.

**Response 7:** Thank you for your comments! We have followed your advice to revise it. Details are in following paragraph and manuscript.

Statistics further indicate that in the next 20–30 years, the expressways in China will measure more than 40,000 km. For every kilometre of highway, the corresponding bare slope area is expected to measure 50,000–70,000 $m^2$ (Wang, 2006).

Reference:

Wang, C. J.: Regional Impaction and Evolution of Express Way Networks in China. PROGRESS IN GEOGRAPHY, 25(6):126-137, 2006 (in Chinese)

**Comment 8:** Line 82 influence should be influences or influenced-depending on whether you are using past tense for 'believed'.

**Response 8:** Thank you for your comments! We have followed your advice to revise it. Details are in following paragraph and manuscript.

Tresch et al. (1995), in a study of Switzerland, argued that the slope length ($L$) or slope steepness factor ($S$) is one of the main factors of soil erosion prediction, and these parameters significantly influenced the erosion values calculated by RUSLE.

**Comment 9:** Line 84, why do you say 'particularly in Switzerland'? This seems a little strange as up to now the work has focussed on China. If Tresch focuses on Switzerland then mention this earlier on, e.g. 'Tresch et al. (1995) in a study of Swiss..'

**Response 9:** We greatly appreciate your valuable suggestion concerning improvement to this

paper. We have followed your advice to revised it. Details are in following paragraph and manuscript.

Tresch et al. (1995), in a study of Switzerland, argued that the slope length (*L*) or slope steepness factor (*S*) is one of the main factors of soil erosion prediction, and these parameters significantly influenced the erosion values calculated by RUSLE. All existing S factors can be derived only from gentle slope inclinations of up to 32%; however, many cultivated areas are steeper than this critical value.

**Comment 10:** Line 88 citation should not have initials. This seems to be a problem throughout.

**Response 10:** Thank you for your patience and careful work! We have revised this sentence again, details are in following paragraph and manuscript.

Rick (2001) found that using universal soil loss equation (USLE) and RUSLE soil erosion models at regional landscape scales was limited by the difficulty of obtaining an *LS* factor grid suitable for geographic information system (GIS) applications.

**Comment 11:** Line 97 Define the parameter K

**Response 11:** Thank you for your comments! We have followed your advice to revise it. Details are in following paragraph and manuscript.

Silburn (2011) showed that estimating soil erodibility factor (*K*) from soil properties (derived from cultivated soils) provided a reasonable estimate of *K* for the main duplex soils at the study site as long as the correction for the undisturbed soil was used to derive K from the measured data prior application to the USLE model (Silburn, 2011).

**Comment 12:** Sentence starting at the end of line 103 goes over five lines and is quite difficult to read. Please split into smaller sentences.

**Response 12:** Thank you for your comments! We have followed your advice to revise it, we hope to meet your requirement. Details are in following paragraph and manuscript.

Chen (2010), who initially considered the terrain characteristics of roadbed side slopes and conducted concrete analysis of the terrain factor calculation method in RUSLE, appraised the

compatible terrain factor computational method of roadbed side slope and proposed a revised method based on the measured data of soil erosion in the subgrade side slope of Hurongxi Expressway (from Enshi to Lichuan) in Hubei Province.

**Comment 13:** Line 131 Define what the C factor is.

**Response 13:** Thank you for your comments! We have followed your advice to revise it. Details are in following paragraph and manuscript.

The method was better than the commonly used techniques based on green vegetation (e.g. normalised difference vegetation index (NDVI)) only, and it was appropriate for estimating the vegetation cover management factor ($C$) in the model led hillslope erosion in New South Wales, Australia by using emerging fractional vegetation cover products.

**Comment 14:** Line 181 mention the method of rainfall interpolation and add references here. I suggest commenting on some of the uncertainties of interpolation in regions with steep topography as at present this is presented as somewhat of a perfect solution.

**Response 14:** We greatly appreciate your valuable suggestion concerning improvement to this paper. We have followed your advice to revised it. Details are in following paragraph and manuscript.

In this study, we analyse the following characteristics of soil erosion to improve certain aspects expressway construction based on previous research: (1) We divide the highway slope into natural and artificial units and calculate the amount of soil loss from the slope surface to the pavement based on the slope surface catchment unit. Considering that this approach is in line with the actual situation, the findings can thus can be popularised. (2) Previous studies have shown that the spatial interpolation method of precipitation has been stagnant in the study of the spatial and temporal distribution of precipitation in mountain areas (Liu and Zhang, 2006). At the same time, the problem involves two aspects. From the timescale perspective, the characteristics of rainfall distribution and the influencing factors are not fully considered. From the spatial scale perspective, the spatial heterogeneity of the region is ignored. Many studies have limited the factors affecting precipitation to altitude factors, thereby leading to low interpolation accuracy (Zhao et al., 2011;

Liu et al., 2010). Thus, we consider the spatial heterogeneity of linear engineering of the expressway. Then, the rainfall factor is spatially interpolated to compensate the following limitations: shortage of rainfall data of mountain areas; difficulty of representing the rainfall data of an entire expressway by using those data from a single meteorological station; and uneven spatial distribution and strong heterogeneity of rainfall in mountain areas (Li et al., 2017). (3) We modify the parameters of the artificial slope by actual survey, runoff plot observation and other methods, and the parameters of the artificial slope are corrected by referring to the form of the project and the utilised materials. In this study, we not only scientifically predict the amount of soil erosion caused by highway construction in mountain areas but also provide a scientific basis for the prevention and control of soil erosion and the rational allocation of prevention and control measures. Moreover, the safe operation of highways and the virtuous cycle of the ecological environment should be ensured to promote the sustainable development of the local economy.

Reference:

Li, Y., Qi S., Cheng, B. H., Ma, J. M., Ma, C., Qiu, Y. D., Chen, Q. Y.: A Study on Factors of Space-time Distributions of Precipitation in Ailao Mountain Area and Comparison of Interpolation Methods. EARTH AND ENVIRONMENT, 45(6): 600-610 (in Chinese)

Liu, J. T., Zhang, J. B.: Interpolation analysis of the spatial distribution of precipitation in mountain area. Journal of Irrigation & Drainage, 25:34-38, 2006 (in Chinese)

Zhao, C. C., Ding, Y. J., Ye, B. S., Zhao, Q. D.: Spatial distribution of precipitation in Tianshan Mountains and its estimation. Advance in water science, 22:315-322, 2011 (in Chinese)

Liu, Z. Y., Zhang, X., Fang, R. H. Analysis of spatial interpolation methods to precipitation in Yulin based on DEM. Journal of Northwest A&F University (Nat. Sci. Ed.), 38:227-234, 2010 (in Chinese)

**Comment 15:** Sentence on line 203 English language needs some work.

**Response 15:** Thank you for your comments! We have followed your advice to revise it. Details are in following paragraph and manuscript.

Between May and the middle of October, the area experiences wet season characterised by

abundant rainfall, concentrated precipitation and increased rain at night time; the variation of precipitation is 400–2000 mm, whilst most regions have 800–1800 mm (Fei et al., 2017; Zhang et al., 2017).

**Comment 16:** Equation 4. There is some variability in the number of decimal places used for different variables. These should be consistent and reflect the precision of the original data. Does e represent Euler's number here, or is it a parameter? Please state.

**Response 16:** Thank you for your comments! We have followed your advice to explain it. Details are in the following paragraph.

In equation 4, the decimal is an empirical parameter whose value is fixed, and it has no variability. 'e' is the base number of natural logarithm and an infinite non recurring decimal.

**Comment 17:** Figure 6. As with figure 5, it is very difficult to distinguish the slope units with the colour scheme you have chosen, and this will not work in greyscale.

**Response 17:** Thank you for your comments! We have followed your advice to revise it. Details are in following paragraph and manuscript.

[Figure]

**Figure 5.** Spatial distribution map of rainfall erosivity factors (K127–K139+800)

[Figure]

**Figure 5(a).** The subgraph of Figure 5 with zoomed sections

[Figure]

**Figure 6.** Spatial distribution map of topographic factors (K134–K139)

**Comment 18:** Table 5 and 6 would be better presented as figures (e.g., bar or line charts) as it is very hard to visually discern any trends or differences in the raw numbers. If needed, you could put the corresponding table into an appendix.

**Response 18:** We are in complete agreement with your comment. The author did not analyze and compare the data in Table 5 and table 6, the two tables are mainly a demonstration of the result of calculation. Therefore, we decided to take the two tables as appendices. Details are in the manuscript.

**Comment 19:** Line 411. 'The research method of Chen Zongwei' is this a reference? Please put in an appropriate format.

**Response 19:** Thank you for your patience and careful work! We have followed your advice to revise it. Details are in the following paragraph and manuscript.

The method of Chen Zongwei (2010) was adopted for the calculation of the *LS* factor of the artificial slopes

Reference:

Chen, Z. W., He, F., and Wang, J. J.: Revises of Terrain Factors of Roadbed Side Slope in Universal Soil Loss Equation. HIGHWAY, 12:180-185, 2010 (in Chinese).

**Comment 20:** Line 435 double bracket should be removed.

**Response 20:** Thank you for your patience and careful work! We have followed your advice to revise it. Details are in the following paragraph and manuscript.

The *C*-factor after topographic analysis is considered an important factor soil loss risk control. In the RUSLE model, the C-factor is used to depict the effects of vegetation cover and management practices on soil erosion rate (Vander-Knijff et al., 2000; Prasannakumar et al., 2011; Alkharabsheh et al., 2013).

**Comment 21:** Figure 14 is good (where you show several segments of road in 'strips') and I suggest using a similar format for figures 5 and 6. Figure 14 and 15 please put a description of the variables in the legend caption. I.e., this is Risk within a natural slope, not just 'natural slope'.

**Response 21:** Thank you for your comments! We have followed your advice to revise it. Details are in following paragraph and manuscript.

[Figure]

**Figure 14.** Risk analysis of soil and water loss under 20-year rainfall conditions

[Figure]

**Figure 15.** Risk analysis of soil and water loss for the 1-year rainfall amount

**Comment 22:** Figure 13 please put a description of the variable I within the legend caption.

**Response 21:** Thank you for your comments! We have followed your advice to revise it.

So that you can interpret the meaning of this Figure more clearly. Details are in following paragraph and manuscript.

[Figure]

**Legend  Rainfall (mm)**

| | | |
|---|---|---|
| 108.09-113.25 | 118.59-123.66 | 128.74-133.08 |
| 113.25-118.59 | 123.66-128.74 | 133.08-138.69 |

**Figure 12.** Rainfall interpolation results under 20-year return

[Figure]

**Legend  I$_{30}$/(mm·h$^{-1}$)**

| | | |
|---|---|---|
| | 46.01-50.42 | 54.68-58.84 |
| 42.10-46.01 | 50.42-54.68 | 58.84-62.00 |

**Figure 13.** Rainfall intensity interpolation results under 20-year return

**Comment 23:** Sentence starting on line 550 is 8 lines long and hard to read. Please split into shorter sentences.

**Response 23:** Thank you for your patience and careful work! We have followed your advice to revise it. Details are in the following paragraph and manuscript.

The RUSLE model can also be used to predict soil erosion of natural slopes. On the premise that rainfall erosivity variations have not been considered, we find that the methods of model parameter acquisition are consistent in the literature analysis and comparison for areas of the same type (Yang 1999; Yang 2002; Peng et al., 2007; Zhao et al., 2007; Chen et al., 2014; Zhu et al., 2016). Moreover, after comparing the monitoring data with runoff plots, we find that the error between the predicted value and the monitoring value calculated by the RUSLE model is negligible (Yang 1999; Yang 2002; Li et al., 2004). These findings indicate that the prediction results of the model are reliable.

**Comment 24:** The discussion section is good in that it brings the findings back into the literature. However, the English needs some work here.

**Response 24:** Thank you for your comment! We have followed your advice to revise it. We hope to meet your requirement. Details are in the following paragraph and manuscript.

**4   Discussion**

Slope is the main factor of soil and water loss caused by highways. Thus, slope is very important for prediction and early warning systems. A highway slope can be divided into natural and engineering (artificial) slopes. The RUSLE model can also be used to predict soil erosion of natural slopes. On the premise that rainfall erosivity variations have not been considered, we find that the methods of model parameter acquisition are consistent in the literature analysis and comparison for areas of the same type (Yang 1999; Yang 2002; Peng et al., 2007; Zhao et al., 2007; Chen et al., 2014; Zhu et al., 2016). Moreover, after comparing the monitoring data with runoff plots, we find that the error between the predicted value and the monitoring value calculated by the RUSLE model is negligible (Yang 1999; Yang 2002; Li et al., 2004). These findings indicate that the prediction results of the model are reliable. In the prediction of slope erosion of engineering (artificial) slopes, previous studies have emphasised surface disturbance during construction (He, 2004; Liu et al., 2011; He, 2008; Hu, 2016; Zhang et al., 2016; Song et al., 2007) but did not consider soil erosion as a result of the construction. In the process of predicting soil and water loss in engineering slopes by using the RUSLE model, the correction of the

conservation support practice factor (i.e. cement block and hexagonal brick) is often ignored (Zhang, 2011; Morschel et al., 2004; Correa and Cruz, 2010). In addition, most cases use RUSLE modelling to predict the soil erosion of highway slopes. Remote sensing is usually based on grid data but does not consider catchment units (IsIam et al., 2018; Villarreal et al., 2016; Wu and Yan 2014; Chen et al., 2010).

---

## Author Response (AR4)

**Response to Decision Letter**

**Dear Dr. Paolo Tarolli,**

Thank you for your letter and for the reviewer's comments concerning our manuscript entitled _"Dangerous degree forecast of soil and water loss on highway slopes in mountainous areas using Revised Universal Soil Loss Equation model"_ (ID: nhess-2017-406). Those comments are all valuable and very helpful for revising and improving our paper, as well as the important guiding significance to our researches. We have studied comments carefully and have made correction which we hope meet with approval. Revised portion are marked in red in the manuscript.

Thank you for your consideration!

Sincerely yours,

*Corresponding Author: Shi Qi

P.S.

=========================================================================

**Response to review's comments for nhess-2017-406-Editor**

**Editor's comments**

**Comment 1:** at this stage of the review process, only minor suggestions have been proposed. Please check carefully the feedback of the referee #2. In addition, there are also few comments from my side, mainly related to improvement of the figures.

**Response 1:** Firstly, all authors wish to express our sincere thanks to you for your positive comments, and thank you for your patience and careful work! Then, we have followed your advice to revised it. Details are in following paragraph and manuscript.

**Comment 2:** fig. 1 please avoid to write into the figure, simply add a yellow arrow, and then describe it in the caption;

**Response 2:** Thank you for your comments! We have followed your advice to revise it. Details are in following paragraph and manuscript.

[Figure]

**Figure 1.** Soil erosion produced by rainwash on a slope after rainfall

**Comment 3:** for the figures related to rainfall interpolation (fig. 4a, fig.12, fig. 13), please use a gradient of blue color, that is more logic when one is looking at water;

**Response 3:** Thank you for your comments! We have followed your advice to revise it. Details are in following paragraph and manuscript.

[Figure]

**Figure 6(a).** Interpolation rainfall results of June 5, 2014
**Figure 6(b).** Interpolation $I_{30}$ results of June 5, 2014

[Figure]

**Figure 12.** Rainfall interpolation results under 20-year return period

**Figure 13.** Rainfall intensity interpolation results under 20-year return period

**Comment 4:** fig. 7 is very difficult to read, since you overwrite some legend and word on the map, can you find a different solution? Maybe also a different range of colors?

**Response 4:** Thank you for your comments! We have followed your advice to revise it. Details are in following paragraph and manuscript.

[Figure]

**Figure 7.** Vegetation coverage along Xinhe Expressway

**Comment 5:** can you merge fig. 12 and 13 in one figure? In addition, what do you mean for "rainfall" and "rainfall intensity"? Please clarify. What do you mean for "20-year return"? Do you mean "return period"?

**Response 5:** We greatly appreciate your valuable suggestion concerning improvement to this paper. We have followed your advice to revised and explained it. Details are in following paragraph and manuscript.

[Figure]

**Figure 15.** Rainfall interpolation results under 20-year return period

**Figure 16.** Rainfall intensity interpolation results under 20-year return period

**Explain:**

➢ Rainfall refer to a weather phenomenon in which condensed water vapor in the atmosphere descends to the earth's surface in different ways.

➢ Rainfall intensity refer to the average amount of rainfall over a period of time. It can be expressed by the depth of rainfall per unit time or by the volume of rainfall per unit area of time.

**Comment 6:** fig. 14 and fig. 15: are necessary the maps of elevations? The investigated road sections are already showed in the figure 1. Maybe you can improve the figure 1, making the topographic map larger, and the symbol/number of the investigated sections bigger

**Response 6:** Thank you for your comments! We have followed your advice to revise it. Details are in following paragraph and manuscript.

[Figure]

**Figure 14.** Risk analysis of soil and water loss under 20-year rainfall conditions

[Figure]

**Figure 15.** Risk analysis of soil and water loss for the 1-year rainfall amount

**Comment 7:** a general comment for all the figures: can you select a different north arrow? Maybe one very simple? Delete the word "legend" in all the figures, since everyone knows what is the legend.

**Response 7:** Thank you for your patience and careful work! We have followed your advice to revise it. Details are in the manuscript.

**Response to Decision Letter**

**Dear Dr. Paolo Tarolli and referee,**

We are very pleased to learn from your letter about revision for our manuscript which entitled _"Dangerous degree forecast of soil and water loss on highway slopes in mountainous areas using RUSLE model"._

We greatly appreciate reviewer's thoughtful suggestions concerning improvement to our paper. These comments are all valuable and very helpful for revising and improving our paper, as well as the important guiding significance to our researches.

Thank you for your consideration!

Sincerely yours,

*Corresponding Author: Shi Qi

P.S.

==================================================================================

**Response to review's comments for nhess-2017-406-Reviewer**

**Reviewer's #2 comments**

**Comment1:** Some of the descriptions or discussions in the manuscript may be simplified. For examples: Point (2), Lines 180-190. You may give a topic sentence with the general idea of the point at first, and then give a concise explanation of the idea. Presently the description here is too complicated and chaotic.

**Response 1:** Thank you for your comment! We have followed your advice to revise it. Details are in following paragraph and manuscript.

Previous studies have shown that the spatial interpolation method of precipitation is unsuitable for the study of the spatiotemporal distribution of precipitation in mountain areas (Liu and Zhang, 2006). The problem involves two aspects. From the timescale perspective, the characteristics of rainfall distribution and the influencing factors are not fully considered. From the spatial scale perspective, the spatial heterogeneity of the region is ignored. Furthermore, many studies have limited the factors that affect precipitation to altitude factors, leading to low interpolation accuracy

(Zhao et al., 2011; Liu et al., 2010). Thus, in this study, we consider the spatial heterogeneity of linear engineering of the expressway. The rainfall factor is spatially interpolated to compensate for the following limitations: shortage of rainfall data on mountain areas, difficulty of representing the rainfall data of an entire expressway by using data from a single meteorological station, and uneven spatial distribution and strong heterogeneity of rainfall in mountain areas (Li et al., 2017).

**Comment 2:** Lines 230-256. All of the titles of level 3, e.g., 2.1.1 Meteorological data, may be erase, and the last paragraph from lines 255 to 256 may be merged to the above paragraph.

**Response 2:** Thank you for your comment! We have followed your advice to revise it. Details are in following paragraph and manuscript.

According to your comment, we deleted all of the titles of level 3, e.g., 2.1.1. We merged line 255-256 to the above paragraph.

**2 Materials and methods**

**2.1 Data sources**

Rainfall data from 2014 were obtained from Hekou Yao Autonomous County, Pingbian Miao Autonomous County, Jinping Miao Yao Autonomous County and the meteorological department of Mengzi. The rainfall data were obtained at 5 min intervals. Meanwhile, two automatic weather stations were established along Xinhe Expressway to gather weather data during the 2014 experiment. Meteorological data, which were provided by the China Meteorological Data Network, covered the period of 1959–2015 (http://data.cma.cn/site/index.html).

Data on soil types were provided by Yunnan Traffic Planning and Design Institute. Data on soil texture and organic matter were obtained via field surveys, data sampling and processing methods. Soil samples were initially collected at each 1 km range of the artificial and natural slopes on both sides of the highway. Five mixed soil samples were obtained from one slope by using the 'S'-shaped sampling method (Shu et al., 2017). Then, the method of coning and quartering was adopted (Oyekunle et al., 2011), and half of the mixed soil samples were brought to the laboratory for analysis. Finally, 186 soil samples were obtained. After the soil samples were dried and sieved, soil texture and organic carbon content were measured via specific gravity speed measurement and potassium dichromate external heating, respectively.

The topographic map and design drawings of Xinhe Expressway were provided by the Traffic Planning and Design Institute of Yunnan Province. The 1:2000 scale of the topographic map coordinate system was based on the 2000 GeKaiMeng urban coordinate system, the elevation system for 1985 national height data and the format for the CAD map in DWG. The remote sensing

images used in this study were derived from 8 m hyperspectral images produced by the GF-1 satellite (http://www.rscloudmart.com/).

**Comment 3:** 5 Conclusions: The sentences from lines 564 to 571 may be erased because they all discuss methods instead of conclusions.

**Response 3:** We are in complete agreement with your comment. We have followed your advice to revise it. Details are in the following paragraph and manuscript.

The error analysis of the actual observation data showed that the overall average absolute error of each monitoring area was 38.65 t·km−2·a−1, the average relative error was 31.18%, the root mean square error was between 20.95 and 65.64 and the Nash efficiency coefficient was 0.67. The method of soil and water loss prediction adopted in this work generally has a smaller error and higher prediction accuracy than other models, and it can satisfy prediction requirements. The risk grades of soil and water loss along the slope of Xinhe Expressway were divided into 20- and 1-year rainfall conditions based on simulated predictions. The results showed that the percentage of slope areas with high and extremely high risks was 7.11%. These areas are mainly located in the K109+500–K110+500 and K133–K139+800 sections. Therefore, relevant departments should strengthen disaster prevention and reduction efforts and corresponding water and soil conservation initiatives in these areas.

**Comment 4:** There are many language errors throughout the manuscript. Numerous sentences are strange and unreadable. Some long terms are composed a series of nouns. I am afraid they are translated from the Chinese-English dictionary words word.

**Response 4:** We greatly appreciate your valuable suggestion concerning improvement to this paper. We have followed your advice to revised it. Details are in the manuscript.

**Comment 5:** Line 22. What is the meaning "the linear highway rainfall" ?

**Response 5:** Thank you for your comment! We have followed your advice to revise it. Details are in following paragraph and manuscript.

The partition of the prediction units of soil and water loss on the expressway slope in the mountainous area and the spatial distribution of rainfall on a linear highway are studied.

**Comment 6:** Line 23: "…the model parameter factor". You may erase any of the words,

parameter or factor.

**Response 6:** Thank you for your comment! We have followed your advice to revise it. Details are in following paragraph and manuscript.

the model parameter is modified, and the risk of soil and water loss along the mountain expressway is simulated and predicted under 20- and 1-year rainfall return periods.

**Comment 7:** Lines 55-56. What' the meaning of the word measure? Do you want to say that in the next 20-30 years, the expressways in China will BE INCREASED BY 40, 000 km?

**Response 7:** We are in complete agreement with your comment. We have followed your advice to revise it. Details are in the following paragraph and manuscript.

Statistics further indicate that in the next 20-30 years, the expressways in China will have a total length of more than 40,000 km.

**Comment 8:** Line 56. The unit of the erosion rate is normally tons/km2.a.

**Response 8:** Thank you for your comment! We checked the relevant literature carefully and then revised it. Details are in following paragraph and manuscript.

The annual amount of soil erosion is 9,000 g/m$^3$, which can cause 450 t of soil loss annually (Chen, 2010).

**Comment 9:** Line 70. The word applicable may be replaced with the word applied.

**Response 9:** We are in complete agreement with your comment. We have followed your advice to revise it. Details are in the following paragraph and manuscript.

RUSLE was derived from the theory of erosion processes and has been applied to more than 10,000 plot-years of data from natural rainfall plots and numerous rainfall-simulation plots.

**Comment 10:** Line 78. The word scientists is preferable than the word scholars.

**Response 10:** We are in complete agreement with your comment. We have followed your advice to revise it. Details are in the following paragraph and manuscript.

many scientists have conducted useful explorations to modify the model's parametric values and improve its simulation accuracy.

**Comment 11:** Line 235. The term rainfall data type is strange.

**Response 11:** Thank you for your comment! We have followed your advice to revise it. Details are in following paragraph and manuscript.

The rainfall data were obtained at 5 min intervals.

**Comment 12:** Line 241. The sentence is strange and unreadable.

**Response 12:** Thank you for your comment! We have followed your advice to revise it. Details are in the following paragraph and manuscript.

Data on soil texture and organic matter were obtained via field surveys, data sampling and processing methods.

**Comment 13:** The definite article THE is always absent when it is needed in the manuscript.

**Response 13:** Thank you for your patience and careful work! We have followed your advice to revise it. Details are in the manuscript.

**Comment 14:** The font size of the text in the figure should be clear and unified. The texts in Figure 2 are too small. So do Figures 4a and b. Some sequence numbers of the figure titles are not right, e. g. Figures 5 and 5a. The representation methods of the titles of Figures 4a and b are not right.

**Response 14:** Thank you for your patience and careful work! We greatly appreciate your valuable suggestion concerning improvement to this paper. We have followed your advice to revised it. Details are in the following paragraph and manuscript.

➢ We revised and typesetted Figure 2 to make it look clearer.

[Figure]

**Figure.2** The location of study areas in China

[Figure]

**Figure.3** The location of the study areas in Yunnan Province

[Figure]

**Figure.4** Overview of the study region

> The layout style of Figure 4 was revised according to the requirements of the editor. At the same time, we have revised the representation methods of the titles of Figures 4a and b.

[Figure]

**Figure 6(a).** Interpolation rainfall results of June 5, 2014

**Figure 6(b).** Interpolation $I_{30}$ results of June 5, 2014

[Figure]

**Figure 7.** Spatial distribution map of rainfall erosivity factors (K127–K139+800)

[Figure]

**Figure 8.** Spatial distribution of rainfall erosion factor in typical a section of a highway

**Comment 15:** Still there are many other minor errors which have not been written here. I strongly suggest you carefully check the whole paper sentence by sentence before you submit the revised version again.

**Response 15:** Thank you for your patience and careful work! We greatly appreciate your valuable suggestion concerning improvement to this paper. We have followed your advice to revised it. Details are in the manuscript.

---

## Author Response (AR5)

**Response to Decision Letter**

**Dear Dr. Paolo Tarolli,**

Thank you for your comments concerning our manuscript entitled *"Dangerous degree forecast of soil and water loss on highway slopes in mountainous areas using Revised Universal Soil Loss Equation model"* (ID: nhess-2017-406). Those comments are all valuable and very helpful for revising and improving our paper, as well as the important guiding significance to our researches. We have studied your comments carefully and have made revision which marked in red in the paper. We have tried our best to revise our manuscript according to the comments which we hope meet with approval.

Thank you very much for your consideration.

Best regards!

Yours sincerely,

*Corresponding Author: Shi Qi

P.S.

=======================================================================

**Response to editor's comments for nhess-2017-406**

**Comment 1:**

After my check on the revised version of the manuscript and on your responses provided at the last review stage, I found a few further issues to be fixed before the final publication. Few of these seem typos, few are some necessary changes, and several are technical corrections on figures; please check carefully all these issues, since they can be critical; the paper cannot meet the high-scientific standards without clarification on these points.

**Response 1:**

Thanks very much for your kind work and consideration on publication of our paper! We have studied your comments carefully and have made revision which marked in red in the paper.

**Comment 2:**

Minor corrections/typos in the text:

In the captions of figures 15 and 16 you corrected with "return period", but in revised text (line 487, line 500, line 504, line 505, line 563, line 564) still you are repeating "20-year rainfall" and "1-year rainfall"; the question is: referring to the text (not to the captions) which kind of rainfall is this? Return period? Cumulated rainfalls? Others? Please correct;

**Response 2:**

Thank you for your patience and careful work! We have followed your advice to revise it. Details are in the following paragraph and manuscript.

The total soil erosion amount of each prediction unit for the 20-year return period rainfall data was obtained by simulation according to the classification standards of soil erosion intensity.

The risk of soil erosion is high in these units. For example, from K134+500 to K135+500 (1000 m), the average soil erosion amount on both sides of the slope for the 20-year return period rainfall amount reached 1757 $t \cdot km^{-2} \cdot a^{-1}$.

Similarly, the risk of soil erosion was analysed according to the grading standard of soil and water loss risk under the 20-year return period rainfall condition. This analysis was performed by simulating the soil erosion amount of each prediction unit for the 1-year return period rainfall amount (Figure 16).

The risk grades of soil and water loss along the slope of Xinhe Expressway were divided into 20- and 1-year return period rainfall conditions based on simulated predictions.

**Comment 3:**

Equation (1): maybe there are few typos here, please also check in literature, but what is "t/hm2"? Is the "time" unit missed? Usually it is written as (t ha−1 yr−1); in general I recommend to check all the unit of measures written in this work, in order to be consistent with the RUSLE literature (and SI system) ("ha" is for hectare, "y" for years); see also fig. 12-13-14: what is "a" in the t km-2 a-1?

**Response 13:**

Thank you for your patience and careful work! We have followed your advice to revise it. Details

are in the following paragraph and manuscript.

where $A$ is the average soil loss per unit area by erosion (t/ha$^2$·y), $R$ is the rainfall erosivity factor (MJ·mm / (ha$^2$·h·y)), $K$ is the soil erodibility factor (t·ha$^2$·h / (ha$^2$·MJ·mm))

"a" refers to anniversary

**Comment 4:**

in the equation (7) and (8) for the unit of the slope you wrote (°), however, in the literature, several scientists referred it as (%) or a-dimensional; which unit is this?

**Response 4:**

Thank you for your comment! We checked the relevant literature carefully and ensured that this unit is correct.

References:

Mccool, D. K., Brown, L. C., Foster, G. R., Mutchler, C. K., and Meyer, L. D.: Revised slope steepness factor for the universal soil loss equation. Transactions of the ASAE-American Society of Agricultural Engineers (USA), 30(5): 1387-1396, 1987.

Liu, B. Y., Nearing, M. A., Shi, P. J., and Jia, Z. W.: Slope length effects on soil loss for steep slopes. Soil Science Society of America Journal 64(5): 1759-1763, 2000.

**Comment 5:**

Some small changes in the English and structure of the sentences need to be provided at the beginning of the chapter 3.2.3. Here my suggestions: line 356 (rephrase saying "slope length factor"), line 364-365 (rephrase with "flowacc is the total number of upstream/contributing pixels for each pixel"), line 365 (rephrase with "…and cell size refers to the DEM resolution (0.5 m). m is a variable length-slope exponent")

**Response 5:**

Special thanks to you for your good comments! We have made correction according to your comments.

On the basis of the topographic map (1:2000 scale) and highway design of Xinhe Expressway,

the slope length factor of the slope catchment was calculated by using DEM data with 0.5 m spatial resolution generated by ArcGIS.

$\lambda$ is the slope length, *flowacc* is the total number of contributing pixels for each pixel that is higher than the pixel and cell size refers to the DEM resolution (0.5m). *m* is a variable length-slope exponent.

**Comment 6:**

In the title and in the text: you can delete "water loss", here you are discussing soil loss;

**Response 6:**

We are in complete agreement with your comment. We have followed your advice to revise it. Details are in the following paragraph and manuscript.

1. Therefore, predicting soil loss on highway slopes is important in protecting infrastructure and human life.

2. The partition of the prediction units of soil loss on the expressway slope in the mountainous area and the spatial distribution of rainfall on a linear highway are studied.

3. Given the particularity of the expressway slope in the mountainous area, the model parameter is modified, and the risk of soil loss along the mountain expressway is simulated and predicted under 20- and 1-year rainfall return periods.

4. (1) Natural watersheds can be considered for the prediction of slope soil erosion to represent the actual situation of soil loss on each slope.

5. Keywords: Soil loss; highway slopes; mountainous areas; RUSLE; dangerous degree forecast

6. The soil loss of roadbed slopes differs from the soil loss in woodlands and farmlands.

7. Soil erosion on roadbed side slopes affects not only soil loss along highways but also road operation safety (Gong and Yang, 2016; Jiang et al., 2017).

8. However, the research on soil loss of highways hardly meets the requirements of practical work (Xu et al., 2009; Bakr et al., 2012).

9. Related literature indicates that research on soil loss in highways has the following limitations.

10. After estimating the historical soil loss of each slope prediction unit, the results were compared with data from the three monitoring plots along the side slope of Xinhe Expressway (Figures 10–12).

11. Figure 15(a)(b). Risk analysis of soil loss under 20-year return period rainfall conditions

12. The grading results showed that the percentage of prediction units classified as having low and mild risks of soil loss was 88.60%.

13. Similarly, the risk of soil erosion was analysed according to the grading standard of soil loss risk under the 20-year return period rainfall condition.

14. Figure 16(a)(b). Risk analysis of soil loss for the 1-year return period rainfall amount

15. Slope is the main factor of the soil and water loss caused by highways. Thus, slope is crucial for prediction and early warning systems.

16. In the process of predicting soil loss in engineering slopes by using the RUSLE model, the correction of the conservation support factor (i.e. cement block and hexagonal brick) is often ignored (Zhang, 2011; Morschel et al., 2004; Correa and Cruz, 2010).

17. The method of soil loss prediction adopted in this work generally has a smaller error and higher prediction accuracy than other models, and it can satisfy prediction requirements.

18. The risk grades of soil loss along the slope of Xinhe Expressway were divided into 20- and 1-year return period rainfall conditions based on simulated predictions.

**Comment 7:**

I would finally suggest a further improvement of the conclusions: provide at the beginning a sentence that describes the purpose of the work and a sentence that describe the study area.

**Response 7:**

We greatly appreciate your valuable suggestion concerning improvement to this paper. We have followed your advice to revised it. Details are in the manuscript.

In this study, we used the revised universal soil loss equation as the prediction model for soil loss on slopes, predicting the soil loss on highway slopes and simulating the risk of soil loss along the mountain expressway. We not only scientifically predict the amount of soil erosion caused by highway construction in mountain areas but also provide a scientific basis for the prevention and control of soil erosion and rational allocation of prevention and control measures. The error analysis of the actual observation data showed that the overall average absolute error of each monitoring area was 38.65 $t \cdot km^{-2} \cdot a^{-1}$, the average relative error was 31.18%, the root mean square error was between 20.95 and 65.64 and the Nash efficiency coefficient was 0.67…

**Comment 8:**

in the maps, when you are showing colours of a gradient (not in the case of rainfall), better to use also a gradient of these (e.g. from blue to red, where red is the maximum or high risk)

**Response 8:**

Thank you for your patience and careful work! We try to use dark legend to indicate high risk, but considering the color matching and visibility of the whole graph, we do not follow the rule of color gradient from blue to red completely. We sincerely hope to meet the publishing requirements!

**Comment 9:**

I would suggest avoiding to put the legend of colours (and also scale bars) into the figures when there is a background colour (it is very difficult to read it and to understand); this is the case of figure 6a and 6b, but also 15 and 16;

**Response 9:**

We are in complete agreement with your comment. We have followed your advice to adjust it. Details are in the following paragraph and manuscript.

[Figure]

**Figure 4(a).** Interpolation results of secondary rainfall for June 5, 2014

**Figure 4(b).** Interpolation results of $I_{30}$ for June 5, 2014

[Figure]

**Figure 13.** Rainfall interpolation results under 20-year return period

**Figure 14.** Rainfall intensity interpolation results under 20-year return period

**Comment 10:**

figure 2-3-4 can be merged in one figure since now they cover too much space; if you provide this, then you need to change then also the figure citation in the text;

**Response 10:**

Thank you for your comment. We have followed your advice to adjust it. Details are in the following paragraph and manuscript.

[Figure]

Figure 2. The location and the overview of the study region

**Comment 11:**

I printed figure 4 to check quality; it seems that you used a very coarse DEM where one can note several strips; surely this is not the DEM you used in your analysis, please re-style it (here, only for the visualization and purpose of the figure, you can just use the new global 30m DEM);

**Response 11:**

Thank you for your comment. We have followed your advice to adjust it. Details are in the following paragraph and manuscript.

[Figure]

**Comment 12:**

for figures 7,8,9,11, try to use a font size of the numbers and words smaller (maybe avoiding to use bold), to improve the visualization of the area (you did it well with 18 and 19);

**Response 12:**

Thank you for your comment. We have followed your advice to adjust it. Details are in the following paragraph and manuscript.

[Figure]

**Figure 5.** Spatial distribution map of rainfall erosivity factors (K127–K139+800)

[Figure]

**Figure 6.** Spatial distribution of rainfall erosion factor in typical a section of a highway

[Figure]

**Figure 7.** Spatial distribution map of topographic factors (K134–K139)

[Figure]

**Figure 9.** Spatial distribution map of the cover and management practice factor

**Comment 13:**

figure 17 and 18: delete in both the blue arrows, while adding (a) and (b), so they can be 17(a) and 17(b), 18(a) and 18(b), change the captions and in the text accordingly;

**Response 13:**

We are in complete agreement with your comment. We have followed your advice to revise it. Details are in the following paragraph and manuscript.

[Figure]

[Figure]

**Figure 17(a)(b).** Risk analysis of soil loss under 20-year return period rainfall conditions

[Figure]

[Figure]

**Figure 18(a)(b).** Risk analysis of soil and water loss for the 1-year return period rainfall amount